# Diffusion Flow Matching: Dimension-Improved KL Bounds and Wasserstein Guarantees

**Marta Gentiloni Silveri** [1]  **Giovanni Conforti** [2]  **Alain Durmus** [1]

## Abstract

Diffusion Flow Matching (DFM) has recently emerged as a versatile framework for generative modeling, yet its theoretical convergence properties remain only partially understood. In this work, we provide refined and novel convergence guarantees for Brownian motion based DFMs, focusing on the discretization error. Our analysis is conducted under the Kullback–Leibler (KL) divergence and the 2-Wasserstein distance. Under finite-moment conditions and a mild score integrability assumption, we derive KL convergence bounds with improved dimensional dependence compared to prior work, achieving, up to our knowledge, state-of-the-art scaling under minimal conditions. We further extend the analysis to the 2-Wasserstein distance: under an additional first-order score integrability assumption and a weak log-concavity condition, we obtain convergence guarantees with dimensional dependence consistent with the KL case.

## 1. Introduction

A central problem in machine learning and statistics is the generation of new samples from a target distribution that is only accessible through finite data. Generative modeling addresses this challenge by learning a mechanism that transforms a simple and tractable base distribution into the data distribution of interest. Deep generative models have been successfully applied in a wide range of domains such as image and video synthesis (Ho et al., 2020; Nichol & Dhariwal, 2021), speech and audio generation (Kong et al., 2020), molecular and material design (Liu & et al., 2023; Zang & Wang, 2020; Dunn & Koes, 2025), and medical imaging

(Luo et al., 2025), owing to their ability to learn and simulate complex data-generating processes from observations alone.

Among the broad class of generative approaches, continuous-time generative models have recently emerged as a particularly powerful and conceptually elegant paradigm. These models describe the transformation between probability distributions through differential equations or stochastic dynamics evolving over time, enabling fine-grained control over the generation process and providing a natural connection to tools from stochastic analysis and optimal transport. Examples include score-based generative models (SGMs) that leverage stochastic differential equations (Ho et al., 2020; Song & Ermon, 2020; 2019), and probability flow ordinary differential equations (Chen et al., 2023b; Rombach et al., 2022; Ramesh et al., 2022; Popov et al., 2021) that enable deterministic sampling.

Within this landscape, Flow Matching (FM) models have gained significant attention as a unifying framework for constructing continuous-time transports between probability distributions (Peluchetti, 2022; Lipman & et al., 2024; Albergo & Vanden-Eijnden, 2022; Albergo et al., 2023; Lipman et al., 2023; Liu, 2022; Liu et al., 2023). The key idea underlying FM is to define a coupling $\pi$ between a base distribution $\mu$ and a target distribution $\nu^\star$, and to introduce an interpolating process, referred to as an *interpolant*, that bridges the two distributions over a finite time horizon. The dynamics of this interpolant induce a time-dependent velocity field that can be learned from samples, thereby specifying a transport map from $\mu$ to $\nu^\star$.

When the interpolant follows deterministic dynamics, the resulting model corresponds to a deterministic FM formulation. Allowing for stochasticity in the interpolant leads instead to *Diffusion FM* (DFM) models. While FM and its stochastic formulation offer greater flexibility and robustness, they also introduce substantial technical challenges. In general, the interpolant does not define a Markov diffusion process and therefore cannot be directly characterized by a stochastic differential equation. To address this issue, DFM relies on the construction of a *Markovian projection*: a diffusion process that matches the marginal distributions of the interpolant at every time. The drift of this mimicking

[1]Ecole Polytechnique, Massy Palaiseau, France [2]Università degli Studi di Padova, Padua, Italy. Correspondence to: Marta Gentiloni Silveri <marta.gentiloni-silveri@polytechnique.edu>, Giovanni Conforti <giovanni.conforti@math.unipd.it>, Alain Durmus <alain.durmus@polytechnique.edu>.

*Proceedings of the 43rd International Conference on Machine Learning*, Seoul, South Korea. PMLR 306, 2026. Copyright 2026 by the author(s).

diffusion satisfies a regression identity and can be efficiently approximated using a neural network trained on samples from the interpolant. Once learned, the diffusion process can be simulated using standard numerical schemes, such as the Euler–Maruyama method, to generate samples from the learned distribution.

Despite promising empirical results (Albergo et al., 2023) and its conceptual appeal, the theoretical foundations of Diffusion Flow Matching remain comparatively underdeveloped. Existing analyses are largely confined to deterministic FM models, while rigorous convergence guarantees for DFMs, particularly in terms of quantitative error bounds and distributional metrics, are still scarce. This gap motivates a deeper theoretical investigation of DFM models and their convergence properties.

**Our contribution** In this work, we analyze DFM models based on $d$-dimensional Brownian bridge. We establish theoretical guarantees for convergence to the target distribution, both in Kullback–Leibler (KL) divergence and in Wasserstein-2 ($\mathscr{W}_2$) distance, under standard and mild assumptions on the data. Our results extend and improve upon prior work by sharpening the dimensional dependence of the KL guarantees and by establishing $\mathscr{W}_2$ bounds accounting for all the sources of error. Our main contributions are summarized as follows:

(1) KL **convergence bounds.**

• **Without early stopping and constant step-size.** We derive in Theorem 1 an improved explicit upper bound on the KL divergence between the target distribution $\nu^\star$ and the DFM output without early stopping, under standard assumptions. In particular, we suppose that the two marginals $\mu$ and $\nu^\star$ admit finite 8-th moment (**H**2) that the coupling $\pi$ admits a score with finite 8-th moment (**H**3), and standard $\mathrm{L}^2$ drift-approximation accuracy (**H**1) under the Markovian projection. The resulting bound achieves $\mathcal{O}(h)$ dependence on the time step size, while improving the dimensional scaling from $\mathcal{O}(d^4)$ to $\mathcal{O}(d^3)$ compared to prior works.

• **With early stopping and constant step-size.** By assuming the same moment conditions on $\mu$ and $\nu^\star$ and approximation error of the drift, replacing the condition **H**3 on $\pi$ by the mild condition that the score of the conditional coupling $\pi_{0|1}$ admits finite 8-th order moment **H**4, we obtain in Theorem 2 an explicit bound on the KL divergence between a smoothed target distribution and the early-stopped DFM output. Our bound preserves the $\mathcal{O}(h)$ dependence and the improved $\mathcal{O}(d^3)$ dimensional scaling of the non-early-stopped case. From this result, we deduce in Corollary 1 that choosing $\delta = \mathcal{O}(\varepsilon^2/d)$ and $h = \mathcal{O}(\varepsilon^{10}/d^7)$ yields a $\mathscr{W}_{2,\mathrm{FM}}^2$-error of order $\mathcal{O}(\varepsilon^2)$.

• **With early stopping and novel step-size schedule.** In Theorem 3, we establish faster KL convergence rates in the early stopping regime via a novel step-size schedule, while preserving the $\mathcal{O}(d^3)$ and $\mathcal{O}(h)$ dependences. The result relies only on the moment and drift approximation assumptions in **H**1 and 2, together with the integrability condition on the score of the conditional coupling $\pi_{0|1}$ (**H**4). As a consequence, Corollary 2 yields improved complexity bounds in the Fortet–Mourier metric: choosing $\delta = \mathcal{O}(\varepsilon^2/d)$, it suffices to take $h = \tilde{\mathcal{O}}(\varepsilon^2/d^3)$, where $\tilde{\mathcal{O}}$ hides logarithmic factors in $d$ and $1/\varepsilon$, to guarantee a $\mathscr{W}_{2,\mathrm{FM}}^2$-error of order $\mathcal{O}(\varepsilon^2)$.

(2) $\mathscr{W}_2$ **convergence bounds.**

• **Without early stopping and constant step-size.** We establish $\mathscr{W}_2$ convergence bounds for DFMs in the non-early-stopped regime in Theorem 4, applicable to a broad class of distributions. Under appropriate $\mathrm{L}^2$-approximation error for the drift **H**5, moment conditions **H**2 and 3, a weak log-concavity assumption on the coupling $\pi$ (**H**6) and an integrability condition on the Jacobian of the score associated with $\pi$ (**H**7), we derive bounds scaling as $\mathcal{O}(\sqrt{h})$ in the time step and $\mathcal{O}(\sqrt{d^3})$ in the dimension. These rates are consistent with the corresponding KL guarantees. Furthermore, in Corollary 3, we show that our results apply in the case where $\pi$ is the independent coupling, provided that the scores of the marginals $\mu$ and $\nu^\star$, together with their Jacobians, are integrable, and weakly concave (**H**8).

**Notation** Given a measurable space $(\mathsf{E}, \mathcal{E})$, we denote by $\mathcal{P}(\mathsf{E})$ the set of probability measures of $\mathsf{E}$. Also, given a topological space $(\mathsf{E}, \tau)$, we use $\mathcal{B}(\mathsf{E})$ to denote the Borel $\sigma$-algebra on $\mathsf{E}$. We denote by $\mathbb{W} = \mathrm{C}([0,1], \mathbb{R}^d)$ the set of continuous functions from $[0,1]$ to $\mathbb{R}^d$ and we refer to it as the Wiener space. We denote by $\mathrm{Leb}^d$ the Lebesgue measure on $\mathbb{R}^d$. Given two probability measures $\mu, \nu \in \mathcal{P}(\mathbb{R}^d)$, we denote by $\Pi(\mu, \nu)$ the set of couplings between $\mu$ and $\nu$, *i.e.*, $\xi \in \Pi(\mu, \nu)$ if and only if $\xi$ is a probability measure on $\mathbb{R}^d \times \mathbb{R}^d$ and $\xi(\mathsf{A} \times \mathbb{R}^d) = \mu(\mathsf{A})$ and $\xi(\mathbb{R}^d \times \mathsf{A}) = \nu(\mathsf{A})$ for all measurable $\mathsf{A} \subseteq \mathbb{R}^d$. The relative entropy (or KL-divergence) of $\mu$ with respect to $\nu$ is defined by $\mathrm{KL}(\mu|\nu) := \int \log(\mathrm{d}\mu/\mathrm{d}\nu)\mathrm{d}\mu$ if $\mu$ is absolutely continuous with respect to $\nu$, and $\mathrm{KL}(\mu|\nu) := +\infty$ otherwise. If $\mu$ and $\nu$ have finite second moment, the $2-$Wasserstein distance is defined by $\mathscr{W}_2^2(\mu, \nu) := \inf_{\xi \in \Pi(\mu,\nu)} \int \|x - y\|^2 \, \mathrm{d}\xi(x,y)$ and the Fortet-Mourier distance of order 2 is defined by $\mathscr{W}_{2,\mathrm{FM}}^2(\mu, \nu) = \inf_{\pi \in \Pi(\mu,\nu)} \int \min\{\|x - y\|^2, 1\}\mathrm{d}\pi(x,y)$. Given $\xi \in \mathcal{P}(\mathbb{R}^d)$ and $p \geq 1$, we denote by $\| \cdot \|_{\mathrm{L}^p(\xi)} := (\int \| \cdot \|^p\mathrm{d}\xi)^{1/p}$ the $\mathrm{L}^p$-norm with respect to $\xi$. Given two real numbers $u, v \in \mathbb{R}$, we write $u \lesssim v$ (resp. $u \gtrsim v$) to mean $u \leq Cv$ (resp. $u \geq Cv$) for a universal constant $C > 0$. Also, we denote by $\|x\|$ the Euclidean norm of $x \in \mathbb{R}^d$, by $\langle x, y \rangle$ the scalar product between $x, y \in \mathbb{R}^d$, and by $x^{\mathrm{T}}$ the transpose of $x$. Last, we use standard Big-$\mathcal{O}$ notation.

## 2. Diffusion Flow Matching

In this section, we provide a concise yet self-contained overview of Brownian motion based Diffusion Flow Matching (DFM) models, following the probabilistic formulation introduced in recent works. Recall that $\nu^\star \in \mathcal{P}(\mathbb{R}^d)$ denote the target (or data) distribution and $\mu \in \mathcal{P}(\mathbb{R}^d)$ denote the base (or prior) distribution.

As already highlighted, DFM is a procedure that designs a generative model able to produce new samples approximately distributed according to $\nu^\star$, by learning a stochastic transport from $\mu$ to $\nu^\star$ over a finite time horizon. To this end, DFM proceeds as follows:

1. **Markovian projection.** First, DFM constructs the Markovian projection $(X_t^{\mathrm{M}})_{t\in[0,1]}$ of an interpolated process $(X_t^{\mathrm{I}})_{t\in[0,1]}$, such that $X_0^{\mathrm{M}}$ and $X_1^{\mathrm{M}}$ have distributions $\mu$ and $\nu^\star$, respectively. The process $(X_t^{\mathrm{M}})_{t\in[0,1]}$ is defined as the solution to a stochastic differential equation (SDE).

2. **Approximation of the Markovian projection.** While $(X_t^{\mathrm{M}})_{t\in[0,1]}$ would yield an exact generative model mapping samples from $\mu$ to samples from $\nu^\star$ if the associated SDE could be solved exactly, this is generally infeasible in practice. Consequently, suitable numerical or modeling approximations must be introduced.

We detail these two crucial stages in what follows.

**Markovian projection.** The core objects at the basis of this construction is a coupling $\pi$ between $\mu$ and $\nu^\star$, *i.e.*, a probability measure on the product space $\mathbb{R}^{2d}$ such that, for any $\mathsf{A} \in \mathcal{B}(\mathbb{R}^d)$, $\pi(\mathsf{A} \times \mathbb{R}^d) = \mu(\mathsf{A})$ and $\pi(\mathbb{R}^d \times \mathsf{A}) = \nu^\star(\mathsf{A})$, and the transition of the Brownian bridge process that we now recall. It is well-known that if $(B_t)_{t\in[0,1]}$ is a Brownian motion starting from a distribution $\mu \in \mathcal{P}(\mathbb{R}^d)$, then it induces a Markov kernel $\mathrm{b}\mathbb{B}$ which corresponds to a regular version of the conditional distribution of the path $(B_t)_{t\in[0,1]}$ given $B_0$ and $B_1$, *i.e.*,

$$\mathbb{P}[(B_t)_{t\in[0,1]} \in \mathsf{A} | (B_0, B_1)] = \mathrm{b}\mathbb{B}((B_0, B_1), \mathsf{A}) ,$$

for any $\mathsf{A} \in \mathcal{B}(\mathbb{W})$. The existence of this kernel is guaranteed, for instance, by Theorem 8.37 in (Klenke, 2013). Then, by integrating these bridges against the coupling $\pi$, one defines the *stochastic interpolant*

$$\mathbb{I}(\pi, \mathbb{B})(\mathsf{A}) = \int \mathrm{b}\mathbb{B}((x_0, x_1), \mathsf{A})\, \pi(\mathrm{d}x_0, \mathrm{d}x_1) , \quad (1)$$

for any $\mathsf{A} \in \mathcal{B}(\mathbb{W})$. This construction corresponds to the law of a stochastic process $(X_t^{\mathrm{I}})_{t\in[0,1]}$, which defines a random path connecting the two marginals: by construction, the interpolant satisfies $X_0^{\mathrm{I}} \sim \mu$ and $X_1^{\mathrm{I}} \sim \nu^\star$.

While $(X_t^{\mathrm{I}})_{t\in[0,1]}$ provides a conceptually meaningful interpolation between $\mu$ and $\nu^\star$, it is generally *non-Markovian*. Indeed, its dynamics depend on the terminal value $X_1^{\mathrm{I}}$ and satisfies

$$\mathrm{d}X_t^{\mathrm{I}} = 2\nabla_x \log p_{1-t}(X_1^{\mathrm{I}} \mid X_t^{\mathrm{I}})\mathrm{d}t + \sqrt{2}\mathrm{d}W_t , \quad t \in [0,1] ,$$

with $X_0^{\mathrm{I}} \sim \mu$. Here $(W_t)_{t\in[0,1]}$ denotes a $d$-dimensional Brownian motion and $(y, x) \mapsto p_s(y|x)$ denotes the heat kernel, *i.e.*, for any $x, y \in \mathbb{R}^d$ and $0 < s \leq 1$

$$p_s(y|x) = \frac{1}{(4\pi s)^{d/2}} \exp\left( -\frac{\|y - x\|^2}{4s} \right) . \quad (2)$$

We refer to Section 2.1 of (Gentiloni Silveri et al., 2024) for details.

To obtain a tractable generative model, one constructs a Markov diffusion $(X_t^{\mathrm{M}})_{t\in[0,1]}$ that shares the same time marginals as the interpolant. This is achieved via the Markovian projection (Gyöngy, 1986; Krylov): $X_0^{\mathrm{M}} \sim \mu$ and

$$\mathrm{d}X_t^{\mathrm{M}} = \tilde{\beta}_t(X_t^{\mathrm{M}})\mathrm{d}t + \sqrt{2}\mathrm{d}W_t , \quad t \in [0,1] , \quad (3)$$

where the mimicking drift is given by the conditional expectation

$$\tilde{\beta}_t(x) = \mathbb{E}\big[2\nabla_x \log p_{1-t}(X_1^{\mathrm{I}} \mid X_t^{\mathrm{I}}) \,\big|\, X_t^{\mathrm{I}} = x\big] \quad (4)$$
$$= \mathbb{E}\left[ \frac{X_1^{\mathrm{I}} - X_t^{\mathrm{I}}}{1-t} \Big| X_t^{\mathrm{I}} = x \right] .$$

The resulting process satisfies

$$X_t^{\mathrm{I}} \stackrel{\mathrm{dist}}{=} X_t^{\mathrm{M}} , \quad (5)$$

for all $t \in [0,1]$ (Gentiloni Silveri et al., 2024; 2025), and therefore constitutes an ideal diffusion model transporting $\mu$ to $\nu^\star$.

While the Markovian projection provides an ideal generative model, it is not directly accessible. First, the mimicking drift defined in (4) is intractable and even using some approximation, the continuous time SDE (3) cannot be simulated exactly. As a result, turning this construction into a workable model requires overcoming these practical limitations.

**Approximation of the Markovian projection.** The first challenge in approximating the Markovian projection is that of approximating the mimicking drift $\tilde{\beta}_t$. Since, for any $t \in [0,1]$, $\tilde{\beta}_t$ writes as a conditional expectation (4), then, by Corollary 8.17 in (Klenke, 2013) and (5), $\tilde{\beta}_t$ solves the regression problem:

$$\arg\min_f \mathbb{E}\Big[ \big\| f(t, X_t^{\mathrm{M}}) - \tilde{\beta}_t(X_t^{\mathrm{M}}) \big\|^2 \Big] .$$

Therefore, for any $t \in [0, 1]$, $\tilde{\beta}_t$ can be estimated using a family of neural networks $\{x \mapsto s_\theta(t, x)\}_{\theta \in \Theta}$ and minimizing the $\mathrm{L}^2$ regression loss

$$\theta \mapsto \mathbb{E}\left[\left\| s_\theta(t, X_t^{\mathrm{M}}) - \tilde{\beta}_t(X_t^{\mathrm{M}}) \right\|^2 \right] .$$

The trained network $s_{\theta^\star}(t, \cdot)$ then serves as proxy of $\tilde{\beta}_t$. However, the resulting diffusion which reads:

$$\mathrm{d}X_t^{\mathrm{NN}} = s_{\theta^\star}(t, X_t^{\mathrm{NN}})\mathrm{d}t + \sqrt{2}\mathrm{d}W_t , \quad t \in [0, 1] ,$$

with $X_0^{\mathrm{NN}} \sim \mu$, cannot be solved in general and we have to rely on numerical discretization. We make use of the Euler-Maruyama scheme, *i.e.*, for a choice of sequence of step sizes $\{h_k\}_{k=1}^N$, $N \geqslant 1$, and the corresponding time discretization $t_k = \sum_{i=1}^k h_i$, such that $t_0 = 0$ and $t_N = 1$, we define the continuous process $(X_t^{\theta^\star})_{t \in [0,1]}$, recursively on the intervals $[t_k, t_{k+1}]$ by

$$\mathrm{d}X_t^{\theta^\star} = s_{\theta^\star}(t_k, X_{t_k}^{\theta^\star})\mathrm{d}t + \sqrt{2}\mathrm{d}W_t , \quad t \in [t_k, t_{k+1}] , \quad (6)$$

with $X_0^{\theta^\star} \sim \mu$. Finally, this dynamics is used to define the generative model associated with the DFM procedure. In particular, approximate samples from $\nu^\star$ are obtained by simulating trajectories of $(X_t^{\theta^\star})_{t \leq 1-\delta}$ with $\delta \geqslant 0$. When $\delta = 0$, the dynamics is run until its terminal time and no early stopping is performed. In contrast, choosing $\delta > 0$ results in an early stopping of the dynamics, which may be beneficial in practice to mitigate numerical instabilities near the terminal time.

In short, DFM generates samples by simulating an Euler–Maruyama discretization of a neural-network approximation of the Markovian projection associated with the stochastic interpolant (6). The simulation is run up to time $t = 1 - \delta$, for some $\delta \geq 0$.

## 3. Main Results

In this section, we present our main theoretical contributions: *non-asymptotic convergence guarantees* for Brownian-motion-based Diffusion DFM (6). We analyze convergence in two metrics: Kullback–Leibler divergence and Wasserstein-2 distance. Our results are established under mild moment and regularity conditions on the distributions $\mu$, $\nu^\star$, and the coupling $\pi$, together with standard $\mathrm{L}^2$ drift-approximation assumptions common in the literature on diffusion and score-based generative models (Chen et al., 2023c;a; Gentiloni Silveri et al., 2024; Gao et al., 2023).

### 3.1. Convergence in Kullback–Leibler Divergence

We first focus on convergence in KL divergence, which provides a strong notion of similarity between distributions

and is widely used in theoretical analyses of diffusion models (Chen et al., 2023a; Conforti et al., 2025; Gentiloni Silveri et al., 2025). We state next two assumptions that we will be made in all of our results.

As is standard, we first assume that the learned drift approximates the true drift with $\varepsilon^2$ accuracy in $\mathrm{L}^2$ norm.

**H1** (Drift approximation). *There exists $\theta^\star \in \Theta$ and $\varepsilon^2 > 0$ such that*

$$\sum_{k=0}^{N-1} h_{k+1}\mathbb{E}\big[\|s_{\theta^\star}(t_k, X_{t_k}^{\mathrm{M}}) - \tilde{\beta}_{t_k}(X_{t_k}^{\mathrm{M}})\|^2\big] \leq \varepsilon^2 .$$

Furthermore, we impose the following moment conditions. For $m \in \mathbb{N}$, $\zeta \in \mathcal{P}(\mathbb{R}^m)$, and $p \geq 1$, we denote by $\mathbf{m}_p[\zeta] = \int \|x\|^p \mathrm{d}\zeta(x)$ the $p$-th moment. When $\zeta$ has a density with respect to $\mathrm{Leb}^m$, we identify it with the density itself.

**H2** (Moment condition). *It holds $\mathbf{m}_8[\mu] + \mathbf{m}_8[\nu^\star] < +\infty$.*

**KL convergence without early stopping and constant step-size.** In the full regime setting, we also impose standard integrability conditions on the score of the coupling, which are routinely assumed in analyses of diffusion-based generative models (Conforti et al., 2025; Gentiloni Silveri et al., 2024).

**H3** (Score integrability of the coupling). *The coupling $\pi$ is absolutely continuous with respect to $\mathrm{Leb}^{2d}$ with positive density, $\log \pi$ is $\mathrm{C}^1$, and $\|\nabla \log \pi\|_{\mathrm{L}^8(\pi)} < +\infty$.*

Under these assumptions, we can quantify how close the law of the generated samples is to the target $\nu^\star$ in KL divergence.

**Theorem 1.** *Let $\{t_k\}_{k=0}^{N_h}$ be a uniform partition of $[0, 1]$ with step size $h = 1/N_h > 0$. Under **H** 1 to 3, if $\mathbf{m}_8[\mu], \mathbf{m}_8[\nu^\star] \lesssim d^4$, denoting by $\nu_1^{\theta^\star}$ the law of $X_1^{\theta^\star}$, we have*

$$\mathrm{KL}(\nu^\star | \nu_1^{\theta^\star}) \lesssim \varepsilon^2 + h(h^{1/8} + 1)\left(d^2 + \|\nabla \log \pi\|_{\mathrm{L}^8(\pi)}^4\right) d .$$

The bound provided in Theorem 1 has two additive contributions: the first term, $\varepsilon^2$, corresponds to the drift-approximation error; while the second term accounts for the discretization error and scales as $\mathcal{O}(h)$. Importantly, the bound scales cubically with the dimension $d$, improving upon previous $d^4$-scaling results (Gentiloni Silveri et al., 2024). We refer to Section 4 for a detailed literature comparison. Note that, to obtain a KL error of order $\mathcal{O}(\varepsilon^2)$ and a computational complexity $\mathcal{O}(\varepsilon^{-2})$, one needs

$$N_h = \frac{1}{\varepsilon^2}\left(d^2 + \|\nabla \log \pi\|_{\mathrm{L}^8(\pi)}^4\right) d ,$$

discretization points.

*Remark* 1. For simplicity, we set the time horizon to $T = 1$. For an arbitrary $T > 0$, the bound in Theorem 1 remains valid with a Brownian reference on $[0, T]$, up to a multiplicative factor $\max\{1, T^8\}$ in the discretization error.

**KL convergence with early stopping and constant step-size.** If the process is stopped before reaching $t = 1$, we can relax **H3** and refine the bound.

More precisely, let $\pi_{0|1}$ denote a regular conditional distribution of $X_0^{\mathrm{I}}$ given $X_1^{\mathrm{I}}$. We assume the following integrability condition.

**H4** (Score integrability of the conditional coupling). *The conditional coupling $\pi_{0|1}$ is absolutely continuous with respect to $\mathrm{Leb}^d$, with strictly positive density. Moreover, $\log \pi_{0|1}$ is $\mathrm{C}^1$ and satisfies $\|\nabla \log \pi_{0|1}\|_{\mathrm{L}^8(\pi_{0|1})} < +\infty$.*

We remark that **H4** is substantially weaker than **H3**. For instance, when the coupling is chosen to be independent, that is $\pi = \mu \otimes \nu^\star$, the conditional distribution $\pi_{0|1}$ coincides with the prior distribution $\mu$, and **H4** reduces to an integrability condition on the score of $\mu$: $\|\nabla \log \mu\|_{\mathrm{L}^8(\mu)} < +\infty$. Consequently, the only assumption involving the target distribution $\nu^\star$ is the moment condition **H2**.

Under this milder assumption, we are able to derive the following result.

**Theorem 2.** *Fix $0 < \delta < 1/2$. Let $\{t_k\}_{k=0}^{N_h}$ be a uniform partition of $[0, 1]$ with step size $h = 1/N_h > 0$. Under **H1, 2** and **4**, if $\mathbf{m}_8[\mu], \mathbf{m}_8[\nu^\star] \lesssim d^4$, then, for $\nu_{1-\delta}^\star$ and $\nu_{1-\delta}^{\theta^\star}$ denoting the laws of $X_{1-\delta}^{\mathrm{M}}$ and $X_{1-\delta}^{\theta^\star}$, we have*

$$\mathrm{KL}(\nu_{1-\delta}^\star | \nu_{1-\delta}^{\theta^\star}) \lesssim \varepsilon^2 + h(h^{1/8} + 1) \\ \cdot \left( \frac{d^2}{\delta^4} + \|\nabla \log \pi_{0|1}\|_{\mathrm{L}^8(\pi_{0|1})}^4 \right) d \ .$$

Consistently with Theorem 1, the dependence on the dimension remains cubic in $d$. As a consequence, our result improves upon Theorem 3 in (Gentiloni Silveri et al., 2024). In this setting, achieving a KL error of order $\mathcal{O}(\varepsilon^2)$ with computational complexity $\mathcal{O}(\varepsilon^{-2})$ is ensured by choosing

$$N_h = \frac{1}{\varepsilon^2} \left( \frac{d^2}{\delta^4} + \|\nabla \log \pi_{0|1}\|_{\mathrm{L}^8(\pi_{0|1})}^4 \right) d \ .$$

An extensive comparison with the related literature is provided in Section 4.

**Corollary 1.** *Fix $\delta = \mathcal{O}(\varepsilon^2/d)$ and $h = \mathcal{O}(\varepsilon^{10}/d^7)$. Let $\{t_k\}_{k=0}^{N_h}$ be a uniform partition of $[0, 1]$ with step size $h = 1/N_h > 0$. Under **H1, 2** and **4**, if $\mathbf{m}_8[\mu], \mathbf{m}_8[\nu^\star] \lesssim d^4$, then, for $\nu_{1-\delta}^{\theta^\star}$ denoting the law of $X_{1-\delta}^{\theta^\star}$, we have*

$$\mathscr{W}_{2,FM}^2(\nu^\star | \nu_{1-\delta}^{\theta^\star}) \lesssim \mathcal{O}(\varepsilon^2) \ .$$

This result ensures a computational complexity $\mathcal{O}(\varepsilon^{-10})$.

**KL convergence with early stopping and novel step-size schedule.** In the early stopping regime, we show that an appropriately designed step-size schedule yields an accelerated rate of convergence in KL. Our result holds under **H1, 2** and **4**.

**Theorem 3.** *Fix $0 < \delta < 1/2$. Let $\{t_k\}_{k=0}^{M_h+N}$ be a partition of $[0, 1]$ with step sizes $\{h_k\}_{k=1}^{M_h+N}$ such that $h_k = h = 1/(2M_h)$ for $k \leq M_h$ and $h_k = h \min\{t_k, 1 - t_k\}$ for $M_h < k \leq M_h + N$. Under **H1, 2** and **4**, denoting by $\nu_{1-\delta}^\star$ and $\nu_{1-\delta}^{\theta^\star}$ the laws of $X_{1-\delta}^{\mathrm{M}}$ and $X_{1-\delta}^{\theta^\star}$, we have*

$$\mathrm{KL}(\nu_{1-\delta}^\star | \nu_{1-\delta}^{\theta^\star}) \lesssim \varepsilon^2 + hd^3 \log \frac{1}{\delta} + h(h^{1/8} + 1) \\ \cdot \left( d^2 + \|\nabla \log \pi_{0|1}\|_{\mathrm{L}^8(\pi_{0|1})}^4 \right) d \ .$$

Note that, to obtain a KL error of order $\mathcal{O}(\varepsilon^2)$ and a computational complexity $\mathcal{O}(\varepsilon^{-2})$, one needs

$$M_h = \frac{1}{2\varepsilon^2} \left[ \left( d^2 + \|\nabla \log \pi_{0|1}\|_{\mathrm{L}^8(\pi_{0|1})}^4 \right) d + d^3 \log \frac{1}{\delta} \right],$$

and $N = 2M_h \log(1/\delta)$.

**Corollary 2.** *Fix $\delta = \mathcal{O}(\varepsilon^2/d)$ and $h = \tilde{\mathcal{O}}(\varepsilon^2/d^3)$, where $\tilde{\mathcal{O}}$ hides logarithmic factors in $d$ and $1/\varepsilon$. Let $\{t_k\}_{k=0}^{M_h+N}$ be a partition of $[0, 1]$ with step sizes $\{h_k\}_{k=1}^{M_h+N}$ such that $h_k = h$ for $k \leq M_h$ and $h_k = h \min\{t_k, 1 - t_k\}$ for $M_h < k \leq M_h + N$. Under **H1, 2** and **4**, if $\mathbf{m}_8[\mu], \mathbf{m}_8[\nu^\star] \lesssim d^4$, then for $\nu_{1-\delta}^{\theta^\star}$ denoting the law of $X_{1-\delta}^{\theta^\star}$, we have*

$$\mathscr{W}_{2,FM}^2(\nu^\star, \nu_{1-\delta}^{\theta^\star}) \lesssim \mathcal{O}(\varepsilon^2) \ .$$

This result ensures a computational complexity $\mathcal{O}(\varepsilon^{-2})$.

### 3.2. Convergence in Wasserstein-2 Distance

We now turn our attention to convergence in *Wasserstein-2 distance*, which is a natural metric for evaluating the quality of generated samples in generative modeling (Gentiloni Silveri & Ocello, 2025; Strasman et al., 2024; Strasman et al.).

In this section, we introduce a modified $\mathrm{L}^2$ drift approximation condition, tailored to the Wasserstein setting:

**H5** (Drift approximation for Wasserstein analysis). *There exist $\theta^\star$ and $\varepsilon > 0$ such that*

$$\sum_{k=0}^{N-1} h_{k+1} \mathbb{E}\left[ \|s_{\theta^\star}(t_k, X_{t_k}^{\theta^\star}) - \tilde{\beta}_{t_k}(X_{t_k}^{\theta^\star})\|^2 \right]^{1/2} \leq \varepsilon \ .$$

Assumptions of this form are standard in the analysis of SGMs under Wasserstein metrics (see, e.g., (Gao et al., 2023; Bruno et al., 2023; Strasman et al., 2024; Gentiloni Silveri & Ocello, 2025; Strasman et al.)). An important feature

of **H5** is that the expectation is taken over the trajectory of the generative process $(X_{t_k}^{\theta^\star})_{k=0}^N$.

In addition, we assume that $\pi$ satisfies a weak log-concavity condition in the sense of (Conforti, 2022), together with suitable integrability assumptions on the Jacobian of its associated score function.

Let $\beta : \mathbb{R}^d \to \mathbb{R}$ be a smooth potential function. Its *weak convexity profile* is defined as

$$\kappa_\beta(r) = \inf \left\{ \frac{\langle \nabla\beta(x) - \nabla\beta(y), x - y \rangle}{\|x - y\|^2} : \|x - y\| = r \right\}, \tag{7}$$

for any $r > 0$. This function quantifies non-uniform convexity lower bounds, allowing one to weaken classical convexity conditions. Furthermore, for any $M \geq 0$ and $r > 0$, define

$$f_M(r) = 2\sqrt{M} \tanh\left(r\sqrt{M}/2\right) .$$

The weak convexity profile (7) is widely used in the analyses of diffusion-based generative models. Recent results establish state-of-the-art KL and $\mathscr{W}_2$ convergence guarantees for these models under such weak regularity conditions (Gentiloni Silveri & Ocello, 2025; Bruno & Sabanis, 2025; Kremling et al., 2025; Gentiloni Silveri et al., 2025).

We assume that $\pi$ is weakly log-concave:

**H6** (Weak log-concavity of the coupling). *$\pi \in \mathcal{P}(\mathbb{R}^d)$ is absolutely continuous with respect to $\mathrm{Leb}^{2d}$, $\log\zeta$ is $\mathcal{C}^1$, and satisfies for some $\alpha_\pi > 0, M_\pi \geq 0$ and any $r > 0$,*

$$\kappa_{-\log\pi}(r) \geq \alpha_\pi - r^{-1} f_{M_\pi}(r) .$$

Strongly log-concave distributions satisfy **H6** as a special case choosing $M_\zeta = 0$. Furthermore, perturbations of strongly log-concave distributions are weakly log-concave: for instance, if $-\log\xi = V + W$ with $V$ strongly convex and $W$ smooth with Lipschitz gradient, then $\xi$ is weakly log-concave. This includes classical double-well potentials, which are widely used in physics and statistics. Finally, it is well-known that Gaussian mixtures satisfy **H6**.

Moreover, we assume that the Jacobian of the score function associated to $\pi$ is integrable:

**H7** (First-order score integrability of the coupling). *The coupling $\pi$ is absolutely continuous with respect to $\mathrm{Leb}^{2d}$ with positive density, $\log\pi$ is $\mathrm{C}^2$, and*

$$\|\nabla^2 \log\pi\|_{\mathrm{L}^2(\pi)} := \left( \int_{\mathbb{R}^{2d}} \left\| \nabla^2 \log\frac{\mathrm{d}\pi}{\mathrm{dLeb}^{2d}} \right\|_{\mathrm{op}}^2 \mathrm{d}\pi \right)^{1/2}$$
$$< +\infty .$$

Note that a Lipschitz score function implies **H7**.

**$\mathscr{W}_2$ convergence bound without early stopping and constant step-size.** Under the conditions above, we can establish precise quantitative bounds on the $\mathscr{W}_2$ distance between the target distribution $\nu^\star$ and the distribution of the generated samples.

**Theorem 4.** *Let $\{t_k\}_{k=0}^{N_h}$ be a uniform partition of $[0,1]$ with step size $h = 1/N_h > 0$. Under **H2**, **3** and **5** to **7**, if $\mathrm{m}_8[\mu], \mathrm{m}_8[\nu^\star] \lesssim d^4$, then, denoting by $\nu_1^{\theta^\star}$ the law of $X_1^{\theta^\star}$, we have*

$$\mathscr{W}_2(\nu^\star, \nu_1^{\theta^\star}) \tag{8}$$
$$\lesssim \mathrm{C} \left( \varepsilon + \sqrt{h}(h^{\frac{1}{16}} + 1)\sqrt{\left(d^2 + \|\nabla\log\pi\|_{\mathrm{L}^8(\pi)}^4\right)d} \right) .$$

*where*

$$\mathrm{C} = \exp\left( \frac{8\sqrt{2}}{\sqrt{\alpha_\pi}} \exp\left( \frac{M_\pi}{\alpha_\pi} \right) \|\nabla^2 \log\pi\|_{\mathrm{L}^2(\pi)} \right) .$$

The bound provided in Theorem 4 naturally decomposes into two additive contributions: one coming from the drift-approximation error, $\varepsilon$, and one from the discretization error, which, consistently with the KL setting, exhibits a $\mathcal{O}(\sqrt{h})$ dependence on the time step and a $\mathcal{O}(\sqrt{d^3})$ dependence on the space dimension. Furthermore, an implication of Theorem 4 is that choosing the number of discretization steps $N_h = d^3/\varepsilon^2$ guarantees a $\mathscr{W}_2$ error of order $\mathcal{O}(\varepsilon)$, with overall computational complexity scaling as $\mathcal{O}(\varepsilon^{-1})$.

We can specify our results to the case where $\pi = \mu \otimes \nu^\star$. In this case, it is sufficient that the marginals $\mu$ and $\nu^\star$ are weakly log-concave in the sense of (Conforti, 2022), and that their scores, together with the corresponding Jacobians, are integrable, to guarantee convergence.

**H8.** *The marginals $\mu$ and $\nu^\star$ are absolutely continuous with respect to $\mathrm{Leb}^d$ with positive densities, $\log\mu, \log\nu^\star$ are $\mathrm{C}^2$, and*

$$\|\nabla\log\mu\|_{\mathrm{L}^8(\mu)} + \|\nabla\log\nu^\star\|_{\mathrm{L}^8(\nu^\star)} < +\infty ,$$
$$\|\nabla^2\log\mu\|_{\mathrm{L}^2(\mu)} + \|\nabla^2\log\nu^\star\|_{\mathrm{L}^2(\nu^\star)} < +\infty .$$

*Moreover, they satisfy for some $\alpha_\mu, \alpha_{\nu^\star} > 0, M_\mu, M_{\nu^\star} \geq 0$ and any $r \geq 0$,*

$$\kappa_{-\log\mu}(r) \geq \alpha_\mu - r^{-1} f_{M_\mu}(r) ,$$
$$\kappa_{-\log\nu^\star}(r) \geq \alpha_{\nu^\star} - r^{-1} f_{M_{\nu^\star}}(r) .$$

Under the above conditions, **H3**, **6** and **7** hold:

**Lemma 1.** *Under **H8**, **H3**, **6** and **7** hold true for $\pi = \mu \otimes \nu^\star$. In particular,*

$$\|\nabla\log\pi\|_{\mathrm{L}^8(\pi)} \lesssim \|\nabla\log\mu\|_{\mathrm{L}^8(\mu)} + \|\nabla\log\nu^\star\|_{\mathrm{L}^8(\nu^\star)}$$
$$\|\nabla^2\log\pi\|_{\mathrm{L}^2(\pi)} \lesssim \|\nabla^2\log\mu\|_{\mathrm{L}^2(\mu)} + \|\nabla^2\log\nu^\star\|_{\mathrm{L}^2(\nu^\star)} ,$$

*and $\pi$ is weakly log-concave with parameters*

$$\alpha_\pi = \min\{\alpha_\mu, \alpha_{\nu^\star}\}, \quad M_\pi = 2\max\{M_\mu, M_{\nu^\star}\}.$$

We then automatically deduce the following corollary.

**Corollary 3.** *Let $\{t_k\}_{k=0}^{N_h}$ be a uniform partition of $[0,1]$ with step size $h = 1/N_h > 0$. Under **H2**, 5 and 8, choosing $\pi = \mu \otimes \nu^\star$ and assuming $\mathbf{m}_8[\mu], \mathbf{m}_8[\nu^\star] \lesssim d^4$, the bound in Theorem 4 holds with $\|\nabla \log \pi\|_{\mathrm{L}^8(\pi)}$, $\|\nabla^2 \log \pi\|_{\mathrm{L}^2(\pi)}$, $\alpha_\pi$ and $M_\pi$ as in Lemma 1.*

# 4. Related works and comparison with existing literature

The theoretical understanding of DFMs remains limited, despite encouraging empirical results (Albergo et al., 2023). While SGMs have been extensively studied in terms of both practical performance and theoretical properties (Chen et al., 2023a; Conforti et al., 2025; Gentiloni Silveri & Ocello, 2025; Arsenyan et al., 2025), rigorous convergence guarantees for DFM models are relatively recent and still leave room for improvement.

**KL Convergence Bounds.** *Non-early stopping.* In this setting, DFM has been analyzed in (Gentiloni Silveri et al., 2024), where the authors derived non-asymptotic KL convergence bounds (Theorem 2). Their analysis accounts for all practical sources of error, including drift estimation and time discretization. The bounds require moment conditions on the marginals (Assumption H1), their scores (Assumption H2(i)), and the score of the coupling (Assumption H2(ii)). The drift estimator is assumed to approximate the true drift in $\mathrm{L}^2$ norm, without any further smoothness assumptions. The resulting rates scale with the dimension as $d^4$. These results provide the first explicit KL guarantees for non-early-stopped DFM that incorporate all sources of error. In Theorem 1, we improve upon Theorem 2 in (Gentiloni Silveri et al., 2024) by relaxing the integrability conditions on the scores of the marginals (Assumption H2(i)) and by reducing dimension dependence to $d^3$.

*Early stopping.* The same work (Gentiloni Silveri et al., 2024) also considers KL convergence under early stopping. In Theorem 3, the authors establish explicit bounds on the KL divergence between a smoothed target distribution and the output of an early-stopped DFM, assuming an independent coupling $\pi = \mu \otimes \nu^\star$, moment conditions on $\mu$ and $\nu^\star$ (Assumption H1), and moment assumptions on the score of $\mu$. The resulting bounds again scale as $d^4$ and, to the best of our knowledge, represent the only existing KL guarantees for early-stopped DFM. In Theorem 2, we consider a more general set of assumptions, which reduces to

those of Theorem 3 in (Gentiloni Silveri et al., 2024) when $\pi$ is chosen as the independent coupling, and we obtain a $\mathcal{O}(d^3)$ dimensional scaling. Consequently, Theorem 2 improves upon Theorem 3 of (Gentiloni Silveri et al., 2024) both in terms of assumptions and dimensional dependence. Moreover, using a tailored step-size schedule and under the same assumptions as Theorem 2, we obtain in Theorem 3 accelerated KL convergence in the early-stopping regime, while maintaining $\mathcal{O}(d^3)$ complexity. From this result, we derive in Corollary 2 improved complexity bounds in the Fortet–Mourier metric: $\delta = \mathcal{O}(\varepsilon^2/d)$ and $h = \tilde{\mathcal{O}}(\varepsilon^2/d^3)$ yield a $\mathcal{W}_{2,\mathrm{FM}}^2$-error of order $O(\varepsilon^2)$.

*Two-sided early stopping.* More recently, (Liu et al., 2025) studied KL convergence for DFMs built on a more general class of stochastic interpolants, allowing for bridge processes beyond the classical Brownian bridge. Their analysis assumes finite eighth-order moments for $\mu$ and $\nu^\star$, as well as bounded variation of the time derivative of the interpolation function (Assumption 4.1), together with a standard $\mathrm{L}^2$ approximation error for the learned drift. However, their convergence guarantees (Theorem 4.3) depend crucially on an early stopping procedure applied not only at the terminal time $t = 1$, but also at the initial time $t = 0$. Consequently, the KL bound decomposes into three additive terms. The first term, of order $\varepsilon^2$, corresponds to the drift estimation error. The second term, $\mathrm{KL}(\mathcal{L}(X_\delta^{\mathrm{M}}) \,|\, \hat{\mu})$, with $\delta > 0$ and $\hat{\mu} \in \mathcal{P}(\mathbb{R}^d)$ denoting the initialization point of the model (6), captures the initialization error: it is strictly positive, analytically intractable, and cannot be controlled in practice. The third term accounts for the discretization error and scales as $\mathcal{O}(d^3)$. Notably, the discretization error diverges as the process approaches the prior distribution. This divergence necessitates truncating the dynamics away from $t = 0$, which in turn induces the non-vanishing initialization error. When specializing their bounds to the Brownian bridge setting (Section 5), the same structural limitations persist: the initialization error remains non-zero and the discretization error diverges when approaching the prior. As a result, their method forfeits one of the main advantages of DFMs over SGMs, namely the ability to generate samples without any intrinsic initialization bias.

**Wasserstein-2 Convergence Bounds.** *Non-early stopping.* Convergence in the 2-Wasserstein distance has been investigated in (Boffi et al., 2025) and (Xiangjun & Zhongjia, 2025). In (Boffi et al., 2025), the authors establish 2-Wasserstein convergence guarantees for pre-trained diffusion models via Lagrangian and Eulerian distillation losses, thereby controlling the Wasserstein gap between the teacher and the student models. However, their analysis requires the velocity field to satisfy a one-sided Lipschitz condition in space with a time-dependent Lipschitz constant, and it does

not account for time-discretization errors. Theorem 3.15 in (Xiangjun & Zhongjia, 2025) provides bounds on the 2-Wasserstein distance between the generative and target distributions that explicitly incorporate all sources of error and scale as $\mathcal{O}(\sqrt{d})$. It is important to emphasize that these guarantees are obtained under the assumption of a Gaussian prior distribution, a setting in which such strong results are to be expected and which is therefore rather restrictive. Moreover, their analysis relies on Gaussian tail assumptions on the target distribution (Assumption 3.7) and on regularity conditions on the learned velocity field (Assumption 3.13). In contrast, in Theorem 4 we establish non-asymptotic and fully quantitative convergence guarantees in $\mathscr{W}_2$ distance that simultaneously capture both drift-approximation and time-discretization errors. These results hold for arbitrary prior distributions and rely only on mild moment and integrability conditions (**H**2, 3 and 7) on the coupling $\pi$, its associated score, and the Jacobian of the score, as well as weak convexity assumptions on the log-density of $\pi$ (**H** 6). Furthermore, our analysis does not require any *a priori* smoothness assumptions on either the drift or its estimator; instead, the necessary regularity is derived directly from **H**6 and 7. In Corollary 3, we further specialize our bounds to the case of the independent coupling, showing that the same guarantees are obtained by imposing analogous assumptions on the marginals, rather than on the joint coupling (**H**8).

*Early stopping.* The same work (Xiangjun & Zhongjia, 2025) also establishes $\mathscr{W}_2$-convergence guarantees under an early stopping procedure (Theorem 3.19). While these bounds account for all sources of error and scale as $\mathcal{O}(\sqrt{d})$, they remain valid only for Gaussian priors and compactly supported target distributions (Assumption 3.17). Moreover, the analysis continues to rely on regularity conditions on the estimator (Assumption 3.13). Therefore, the resulting $\mathscr{W}_2$ bounds exhibit the same structural limitations as in the non-early-stopped setting.

## 5. Methodology

**KL Convergence Bounds.** Once the standard decomposition of the KL divergence based on Girsanov's theorem is applied, and Itô's formula is used on the process $(\tilde{\beta}_t(X_t^{\mathrm{M}}))_{t\in[0,T]}$, the problem reduces to obtaining an upper bound on the time-integrated $\mathrm{L}^2$-norm of the mean acceleration field driving the evolution of the mimicking drift, namely $(\partial_t + \mathcal{L}_t^{\mathrm{M}})\tilde{\beta}_t$, where $\mathcal{L}_t^{\mathrm{M}}$ denotes the generator of $(X_t^{\mathrm{M}})_{t\in[0,1]}$. This quantity—often referred to as the reciprocal characteristic of the mimicking drift—admits an explicit expansion involving up to three logarithmic derivatives of conditional distributions. To control its integrated norm, we adapt a strategy inspired by (Gentiloni Silveri et al., 2024). The main novelty lies in the treatment of the most delicate contributions, namely the terms involving three logarithmic

derivatives. Here, we depart from the approach of (Gentiloni Silveri et al., 2024) and develop refined bounds. In (Gentiloni Silveri et al., 2024) these are handled via direct expansion, leading to $d^4$ scaling. Our key idea is to avoid this expansion and instead apply an integration-by-parts argument, which transfers one derivative onto the coupling. This effectively reduces the order of differentiation from three to two, lowering the combinatorial complexity and improving the scaling from $d^4$ to $d^3$.

**Wasserstein-2 Convergence Bounds** Our starting point is the synchronous coupling between the Markovian projection $(X_t^{\mathrm{M}})_{t\in[0,1]}$ of the stochastic interpolant and the continuous-time interpolation $(X_t^{\theta^\star})_{t\in[0,1]}$ of the generative model. The proof then proceeds via a recursive argument in time: for each discretization step $k$, we estimate $\|X_{t_{k+1}}^{\mathrm{M}} - X_{t_{k+1}}^{\theta^\star}\|_{\mathrm{L}^2}$ in terms of $\|X_{t_k}^{\mathrm{M}} - X_{t_k}^{\theta^\star}\|_{\mathrm{L}^2}$. and propagate the bound along the time grid.Such a recursion requires sufficient regularity of the drift, and establishing this regularity under minimal data assumptions is the main technical challenge. Such a recursion requires sufficient regularity of the drift, and establishing this regularity under minimal data assumptions is the main technical challenge. The key step is to obtain a bound on the Lipschitz constant of the mimicking drift. To this end, we exploit the fact that its Jacobian admits an explicit representation in terms of conditional expectations involving logarithmic derivatives of conditional densities. This representation allows us to leverage the structure of the stochastic interpolant: since it shares the same Brownian bridge, the regularity assumptions imposed on the data distribution can be transported to the associated conditional kernels and their marginals (see Lemma 7). Building on this, we control the resulting conditional expectations via log-Sobolev and Poincaré inequalities, which yields a quantitative bound on the spatial Lipschitz constant of the mimicking drift (see Proposition 4). Once this regularity estimate is established, the recursive scheme closes and leads to the desired bound. We refer to the proof of Theorem 4 in Appendix C for more details.

## 6. Conclusion

In this work, we conduct a thorough theoretical analysis of a DFM model built upon a $d$-dimensional Brownian bridge. We establish dimension-improved KL convergence bounds and provide $\mathscr{W}_2$ convergence guarantees, under mild and standard assumptions. Despite these advances, several avenues remain open for improvement. In particular, it would be highly valuable to relax the integrability requirements on the score functions further; to achieve sharper dependence on the space dimension and to undertake a statistical analysis of DFMs.

## Acknowledgments and Disclosure of Funding

The work of Marta Gentiloni-Silveri has been supported by the Paris Ile-de-France Région in the framework of DIM AI4IDF. Alain Durmus has received funding from the Fondation de l'École polytechnique as part of its "Servir la science" campaign. The work of Alain Durmus is supported by the France 2030 program with the reference ANR-25-PEIA-0001 (THEOREM project); by Hi! Paris and Agence Nationale de la Recherche (Grant 11-LABX-0047) and by the European Union (ERC-2022-SYG-OCEAN-101071601). Views and opinions expressed are however those of the author(s) only and do not necessarily reflect those of the European Union or the European Research Council Executive Agency. Neither the European Union nor the granting authority can be held responsible for them.

## Impact Statement

This paper presents work whose goal is to advance the field of machine learning. There are many potential societal consequences of our work, none of which we feel must be specifically highlighted here.

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

# A. Preliminaries

**Extra Notation** Given a matrix $\mathbf{A} \in \mathbb{R}^{n \times s}$, we denote by $\|\mathbf{A}\|_{\mathrm{op}}$ the operator norm of $\mathbf{A}$. For $f : [0,1] \times \mathbb{R}^d \to \mathbb{R}$ regular enough, we denote by $\nabla_x f(t,x), \nabla_x^2 f(t,x)$ and $\Delta_x f(t,x)$ respectively the gradient, hessian and laplacian of $f$, defined for $t, x \in [0,1] \times \mathbb{R}^d$ by $\nabla_x f(t,x) := (\partial_{x_i} f(t,x))_i$, $\nabla_x^2 f(t,x) := (\partial_{x_i} \partial_{x_j} f(t,x))_{i,j}$, $\Delta f(t,x) := \sum_{i=1}^d \partial_{x_i}^2 f(t,x)$, where $\partial_{x_j}$ denotes the partial derivative with respect to the $j$-th variable. For $F : [0,1] \times \mathbb{R}^d \to \mathbb{R}^d$ regular enough, we denote by $D_x F$, $\mathrm{div}_x F$ and $\Delta_x F$ respectively, the Jacobian matrix, the divergence and the vectorial laplacian of $F$, defined for $t, x \in [0,1] \times \mathbb{R}^d$ by $D_x F(t,x) = (\partial_{x_j} F_i(t,x))_{i,j}$, $\mathrm{div}_x F(t,x) := \sum_{j=1}^d \partial_{x_j} F_j(t,x)$, $\Delta_x F(t,x) = (\Delta_x F_1(t,x), ..., \Delta_x F_d(t,x))$.

We remark that the stochastic interpolant admits the simple closed form

$$X_t^{\mathrm{I}} \stackrel{\mathrm{dist}}{=} (1-t)X_0^{\mathrm{I}} + tX_1^{\mathrm{I}} + \sqrt{2t(1-t)}Z , \quad Z \sim \mathrm{N}(0, \mathrm{Id}) , \tag{9}$$

Denote by $(p_t^{\mathrm{I}})_{t \in [0,1]}$ the time marginal densities of $(X_t^{\mathrm{I}})_{t \in [0,1]}$ with respect to the Lebesgue measure. Note that, as a consequence of the very definition (1) of the stochastic interpolant, they write as

$$p_t^{\mathrm{I}}(x) = \int_{\mathbb{R}^{2d}} p_t(x|x_0)p_{1-t}(x_1|x)\tilde{\pi}(\mathrm{d}x_0, \mathrm{d}x_1) ,$$

with

$$\tilde{\pi}(\mathrm{d}x_0, \mathrm{d}x_1) = \frac{\pi(\mathrm{d}x_0, \mathrm{d}x_1)}{p_1(x_1|x_0)} . \tag{10}$$

Also, denote by $\mathrm{K}_t$ the regular kernel associated with the conditional distribution of $(X_0^{\mathrm{I}}, X_1^{\mathrm{I}})$ given $X_t^{\mathrm{I}}$, *i.e.*, the map $\mathrm{K}_t : \mathbb{R}^d \times \mathcal{B}(\mathbb{R}^{2d}) \to [0,1]$ such that

(i) $y \mapsto \mathrm{K}_t(y, \mathsf{A})$ is measurable for any $\mathsf{A} \in \mathcal{B}(\mathbb{R}^{2d})$;

(ii) $\mathsf{A} \mapsto \mathrm{K}_t(y, \mathsf{A})$ is a probability measure for any $y \in \mathbb{R}^d$;

(iii) almost surely it holds

$$\mathrm{K}_t(X_t^{\mathrm{I}}, \mathsf{A}) := \mathbb{E}\left[\mathbb{1}_{\mathsf{A}}(X_0^{\mathrm{I}}, X_T^{\mathrm{I}})|X_t^{\mathrm{I}}\right] , \quad \mathsf{A} \in \mathcal{B}(\mathbb{R}^{2d}) .$$

It follows again from (1) that $\mathrm{K}_t$ admits a transition density with respect to $\mathrm{Leb}^{2d}$ given by

$$\mathrm{k}_t(x_0, x_1|x_t) = p_{t|(0,1)}(x_t|x_0, x_1)(p_t^{\mathrm{I}}(x_t))^{-1}\pi(x_0, x_1) , \tag{11}$$

with $p_{t|(0,1)}$ denoting the conditional density of $B_t$ given $(B_0, B_1)$, *i.e.*,

$$p_{t|(0,1)}(x_0, x_1|x_t) = \frac{p_{1-t}(x_1|x_t)p_t(x_t|x_0)}{p_1(x_1|x_0)} = \exp\left(-\frac{\|x_t - (1-t)x_0 - tx_1\|^2}{4t(1-t)}\right) .$$

Denote also by $\mathrm{K}_t^{\#0}$ and $\mathrm{K}_t^{\#1}$ its first and second marginal respectively and by $\mathrm{k}_t^{\#0} := \mathrm{d}\mathrm{K}_t^{\#0}/\mathrm{d}\mathrm{Leb}^d$, and $\mathrm{k}_t^{\#1} := \mathrm{d}\mathrm{K}_t^{\#1}/\mathrm{d}\mathrm{Leb}^d$ the associated densities. We observe that, with the introduced notation, for any $t \in [0,1)$ and $y_t \in \mathbb{R}^d$, we have that

$$\tilde{\beta}_t(x_t) = \int \frac{x_1 - x_t}{1-t}\mathrm{k}_t(\mathrm{d}x_0, \mathrm{d}x_1|x_t) = \int \frac{x_1 - x_t}{1-t}\mathrm{k}_t^{\#1}(\mathrm{d}x_1|x_t) .$$

## A.1. On the heat kernel

It is well-known that $(s, x, y) \mapsto p_s(y|x)$ defined in (2) is twice continuously differentiable in the space variables $x$ and $y$ and satisfies for $x, y \in \mathbb{R}^d, s \in (0, 1]$,

$$\nabla_x p_s(y|x) = -\frac{x-y}{2s}p_s(y|x) = -\nabla_y p_s(y|x) , \tag{12}$$

$$\nabla_x^2 p_s(y|x) = -\frac{1}{2s} p_s(y|x) \operatorname{Id} + \frac{(x-y)(x-y)^{\mathrm{T}}}{4s^2} p_s(y|x) = \nabla_y^2 p_s(y|x) \ ,$$

$$\Delta_x p_s(y|x) = -\frac{d}{2s} p_s(y|x) + \left\| \frac{x-y}{2s} \right\|^2 p_s(y|x) = \Delta_y p_s(y|x) \ . \tag{13}$$

Moreover (2) satisfies the heat equation, *i.e.*,

$$\partial_s p_s(y|x) = \Delta_x p_s(y|x) \ , \quad s \in (0,1] \ , \ x, y \in \mathbb{R}^d \ . \tag{14}$$

Thus, in particular, $(s, x, y) \mapsto p_s(y|x)$ is continuously differentiable in the time variable $s$.

### A.2. On score's integrability

**Lemma 2.** *Under H3, for any $p \in \{2, 4, 8\}$, it holds*

$$\|\nabla \log \mu\|_{\mathrm{L}^p(\mu)}^p \ , \|\nabla \log \nu^\star\|_{\mathrm{L}^p(\nu^\star)}^p \lesssim \|\nabla \log \pi\|_{\mathrm{L}^p(\pi)}^p \ . \tag{15}$$

*Moreover, if we assume also H2, it holds*

$$\|\nabla \log \tilde{\pi}\|_{\mathrm{L}^p(\pi)} \le \|\nabla \log \pi\|_{\mathrm{L}^p(\pi)} + \sqrt[p]{\mathbf{m}_p[\mu]} + \sqrt[p]{\mathbf{m}_p[\nu^\star]} \ . \tag{16}$$

*Proof of Lemma 2:* We start with (15). We only deal with $\mu$, as the argument for $\nu^\star$ is the very same. We have that

$$\nabla \log \mu(x_0) = \frac{\nabla \mu(x_0)}{\mu(x_0)} = \frac{\int \nabla_{x_0} \pi(x_0, x_1) \mathrm{d}x_1}{\int \pi(x_0, x_1) \mathrm{d}x_1} = \frac{\int \pi(x_0, x_1) \nabla_{x_0} \log \pi(x_0, x_1) \mathrm{d}x_1}{\int \pi(x_0, x_1) \mathrm{d}x_1}$$

$$= \mathbb{E}\left[ \nabla_{x_0} \log \pi(X_0^{\mathrm{I}}, X_1^{\mathrm{I}}) | X_0^{\mathrm{I}} = x_0 \right] \ .$$

Jensen inequality yields that

$$\|\nabla \log \mu(x_0)\|^p \le \mathbb{E}\left[ \left\| \nabla_{x_0} \log \pi(X_0^{\mathrm{I}}, X_1^{\mathrm{I}}) \right\|^p | X_0^{\mathrm{I}} = x_0 \right] \ .$$

Taking expectation and using the properties of conditional expectation, we get that

$$\|\nabla \log \mu\|_{\mathrm{L}^p(\mu)}^p = \mathbb{E}\left[ \|\nabla \log \mu(X_0^{\mathrm{I}})\|^p \right] \le \mathbb{E}\left[ \mathbb{E}\left[ \left\| \nabla_{x_0} \log \pi(X_0^{\mathrm{I}}, X_1^{\mathrm{I}}) \right\|^p | X_0^{\mathrm{I}} \right] \right]$$

$$= \mathbb{E}\left[ \left\| \nabla_{x_0} \log \pi(X_0^{\mathrm{I}}, X_1^{\mathrm{I}}) \right\|^p \right] = \|\nabla_{x_0} \log \pi\|_{\mathrm{L}^p(\pi)}^p \le \|\nabla \log \pi\|_{\mathrm{L}^p(\pi)}^p \ .$$

The bound in (16) is a direct consequence of the definitions (10) of $\tilde{\pi}$ and (2) of $p_1(x_1|x_0)$. $\qquad\square$

### A.3. On the Stochastic Interpolant

We collect here several results on Stochastic Interpolants that will be used later. Lemma 3 reports the result of Lemma 1 in (Gentiloni Silveri et al., 2024) and is a direct consequence of (9). Propositions 1 and 2 are taken from Lemma 2 in (Gentiloni Silveri et al., 2024), while Lemmas 4 and 5 follow from Lemma 3 in (Gentiloni Silveri et al., 2024) and our Lemma 2. We refer to (Gentiloni Silveri et al., 2024) for proofs.

**Lemma 3.** *For any $p \ge 1$, they hold*

$$\mathbb{E}[\| X_s^{\mathrm{I}} - X_0^{\mathrm{I}} \|^{2p}] \lesssim s^{2p} \mathbf{m}_{2p}[\mu] + s^{2p} \mathbf{m}_{2p}[\nu^\star] + d^p s^p (1-s)^p \ ,$$

*and*

$$\mathbb{E}[\| X_1^{\mathrm{I}} - X_s^{\mathrm{I}} \|^{2p}] \mathrm{d}s \lesssim (1-s)^{2p} \mathbf{m}_{2p}[\mu] + (1-s)^{2p} \mathbf{m}_{2p}[\nu^\star] + d^p s^p (1-s)^p \ .$$

**Proposition 1.** *The time reversal of the stochastic interpolant $((X_t^{\mathrm{I}})_{t \in [0,1]})^{\mathrm{R}}$ solves weakly*

$$\mathrm{d}\overleftarrow{X}_t = 2 \overleftarrow{b}_t(\overleftarrow{X}_0, \overleftarrow{X}_t) \mathrm{d}t + \sqrt{2} \mathrm{d}\overleftarrow{B}_t \ , \quad t \in [0,1] \ , \quad \overleftarrow{X}_0 \sim \nu^\star \ .$$

with $(\overleftarrow{B}_t)_{t \in [0,1]}$ *d-dimensional Brownian motion independent of* $\overleftarrow{X}_0$ *and* $(t, x) \mapsto \overleftarrow{b}_t(x)$ *as in (Gentiloni Silveri et al., 2024), Lemma 2. In particular, they hold*

$$(X_t^{\mathrm{I}})_{t \in [0,1]} \stackrel{dist}{=} ((\overleftarrow{X}_t)_{t \in [0,1]})^{\mathrm{R}} ,$$

*and, for any* $u \in [0, 1]$ *and* $t \in [0, 1 - u]$,

$$X_{1-(t+u)}^{\mathrm{I}} - X_{1-u}^{\mathrm{I}} \stackrel{dist}{=} \overleftarrow{f}_u^t + \overleftarrow{g}_u^t , \quad \overleftarrow{f}_u^t = 2 \int_u^{t+u} \overleftarrow{b}_r(\overleftarrow{X}_0, \overleftarrow{X}_r) \mathrm{d}r , \quad \overleftarrow{g}_u^t = \overleftarrow{B}_{t+u} - \overleftarrow{B}_u .$$

*Moreover,* $(\overleftarrow{X}_t)_{t \in [0,1]}$ *is* $(\overleftarrow{\mathcal{F}}_t)_{t \in [0,1]}$*-adapted, with* $\overleftarrow{\mathcal{F}}_t = \sigma(\overleftarrow{X}_0, (\overleftarrow{B}_u)_{u \leq t})$.

**Lemma 4.** *Assume* **H**2 *and* 3. *For any* $u \in [0, 1]$, $t \in [0, 1 - u]$, $p \in \{2, 4, 8\}$, *we have that*

$$\mathbb{E}\left[ \left\| \overleftarrow{f}_u^t \right\|^p \right] \lesssim t^p \left( \| \nabla \log \pi \|_{\mathrm{L}^p(\pi)}^p + \mathbf{m}_p[\mu] + \mathbf{m}_p[\nu^\star] \right) .$$

**Proposition 2.** *The stochastic interpolant* $(X_t^{\mathrm{I}})_{t \in [0,1]}$ *solves weakly*

$$\mathrm{d}\overrightarrow{X}_t = 2 \overrightarrow{b}_t(\overrightarrow{X}_0, \overrightarrow{X}_t) \mathrm{d}t + \sqrt{2} \mathrm{d}\overrightarrow{B}_t , \quad t \in [0, 1] , \quad \overrightarrow{X}_0 \sim \mu .$$

*with* $(\overrightarrow{B}_t)_{t \in [0,1]}$ *d-dimensional Brownian motion independent of* $\overrightarrow{X}_0$ *and* $(t, x) \mapsto \overrightarrow{b}_t(x)$ *as in (Gentiloni Silveri et al., 2024), Lemma 2. In particular, they hold*

$$(X_t^{\mathrm{I}})_{t \in [0,1]} \stackrel{dist}{=} (\overrightarrow{X}_t)_{t \in [0,1]} ,$$

*and for any* $u \in [0, 1]$ *and* $t \in [0, 1 - u]$,

$$X_{t+u}^{\mathrm{I}} - X_u^{\mathrm{I}} \stackrel{dist}{=} \overrightarrow{f}_u^t + \overrightarrow{g}_u^t , \quad \overrightarrow{f}_u^t = 2 \int_u^{t+u} \overrightarrow{b}_r(\overrightarrow{X}_0, \overrightarrow{X}_r) \mathrm{d}r , \quad \overrightarrow{g}_u^t = \overrightarrow{B}_{t+u} - \overrightarrow{B}_u .$$

*Moreover,* $(\overrightarrow{X}_t)_{t \in [0,1]}$ *is* $(\overrightarrow{\mathcal{F}}_t)_{t \in [0,1]}$*-adapted, with* $\overrightarrow{\mathcal{F}}_t = \sigma(\overrightarrow{X}_0, (\overrightarrow{B}_u)_{u \leq t})$.

**Lemma 5.** *Assume* **H**2 *and* 3. *For any* $u \in [0, 1]$, $t \in [0, 1 - u]$, $p \in \{2, 4, 8\}$, *we have that*

$$\mathbb{E}\left[ \left\| \overrightarrow{f}_u^t \right\|^p \right] \lesssim t^p \left( \| \nabla \log \pi \|_{\mathrm{L}^p(\pi)}^p + \mathbf{m}_p[\mu] + \mathbf{m}_p[\nu^\star] \right) ,$$

### A.4. On the conditional Markov kernel

A.4.1. STRONGLY LOG-CONCAVE CASE

Strongly log-concave couplings satisfy **H**6. More specifically, let

**H**9 (Strong log-concavity of the coupling). $\pi \in \mathcal{P}(\mathbb{R}^{2d})$ *is absolutely continuous with respect to* $\mathrm{Leb}^{2d}$, $\log \pi$ *is* $\mathrm{C}^2$, *and satisfies*

$$\nabla^2(-\log \pi) \succeq \alpha_\pi \, \mathrm{Id} ,$$

then $\pi$ satisfies **H**6 with $M_\pi = 0$. For sake of clarity, we first consider the case where $\pi \in \Pi(\mu, \nu^\star)$ satisfies **H**9.

**Lemma 6.** *Assume that* $\pi \in \Pi(\mu, \nu^\star)$ *satisfies* **H**9. *Fix* $t \in (0, 1)$ *and* $y_t \in \mathbb{R}^d$. *Then,* $\mathrm{K}_t(y_t, \cdot)$ *satisfies* **H**9 *with the same parameter* $\alpha_\pi$. *Moreover,* $\mathrm{K}_t^{\#0}(y_t, \cdot)$ *and* $\mathrm{K}_t^{\#1}(y_t, \cdot)$ *satisfy* **H**9 *as well with better parameters. Specifically,*

$$\nabla^2(-\log \mathrm{k}_t^{\#0}(y_t|\cdot)) \succeq \left( \alpha_\pi + \frac{1-t}{2t} \right) \mathrm{Id} , \quad \nabla^2(-\log \mathrm{k}_t^{\#1}(y_t|\cdot)) \succeq \left( \alpha_\pi + \frac{t}{2(1-t)} \right) \mathrm{Id} . \tag{17}$$

*Proof of Lemma* 6: Fix $t \in [0, 1)$ and $y_t \in \mathbb{R}^d$ and denote by

$$V(y_0, y_1) := -\log \pi(y_0, y_1) .$$

Then, because of (11), it holds

$$\mathrm{k}_t(y_0, y_1 | y_t) \propto \exp\left(-\frac{\|y_t - (1-t)y_0 - ty_1\|^2}{4t(1-t)} - V(y_0, y_1)\right) \,,$$

hence

$$-\log \mathrm{k}_t(y_0, y_1 | y_t) = \frac{\|y_t - (1-t)y_0 - ty_1\|^2}{4t(1-t)} + V(y_0, y_1) \,, \tag{18}$$

and

$$\nabla_{y_0}^2(-\log \mathrm{k}_t(y_0, y_1|y_t)) = \frac{1-t}{2t} + \nabla_{y_0}^2 V(y_0, y_1) \,, \quad \nabla_{y_1}^2(-\log \mathrm{k}_t(y_0, y_1|y_t)) = \frac{t}{2(1-t)} + \nabla_{y_1}^2 V(y_0, y_1) \,.$$

At this pont, (17) follows directly from **H**9.

$\square$

### A.4.2. WEAKLY LOG-CONCAVE CASE

We now consider the general case where $\pi \in \Pi(\mu, \nu^\star)$ satisfies **H**6.

**Lemma 7.** *Assume that $\pi \in \Pi(\mu, \nu^\star)$ satisfies **H**6. Fix $t \in (0,1)$ and $y_t \in \mathbb{R}^d$. Then, $\mathrm{K}_t(y_t, \cdot)$ satisfies **H**6 with the same parameters $\alpha_\pi, M_\pi$. Moreover, $\mathrm{K}_t^{\#0}(y_t, \cdot)$ and $\mathrm{K}_t^{\#1}(y_t, \cdot)$ satisfy **H**6 with better parameters. Specifically,*

$$k_{-\log \mathrm{k}_t^{\#0}}(r) \geq \left(\alpha_\pi + \frac{1-t}{2t}\right) - r^{-1} f_{M_\pi}(r) \,, \quad k_{-\log \mathrm{k}_t^{\#1}}(r) \geq \left(\alpha_\pi + \frac{t}{2(1-t)}\right) - r^{-1} f_{M_\pi}(r) \,. \tag{19}$$

*Proof of Lemma 7:* Fix $t \in (0,1)$ and $y_t \in \mathbb{R}^d$. Proceed as in the proof of Lemma 6 to get (18). Using **H**6 and (18), we get that $\mathrm{K}_t(y_t, \cdot)$ satisfies **H**6 with the same parameters $\alpha_\pi, M_\pi$ and that the marginals of $\mathrm{K}_t(y_t, \cdot)$ satisfy **H**6 with parameters given by (19).

$\square$

## A.5. On the mimicking drift's regularity

### A.5.1. STRONGLY LOG CONCAVE CASE

We preface this section with an auxiliary technical lemma.

**Lemma 8.** *Let*

$$Y = \begin{pmatrix} Y^{(1)} \\ Y^{(2)} \end{pmatrix} \,, \quad Y^{(1)}, Y^{(2)} \in \mathbb{R}^d \,,$$

*be a random vector with finite second order moments. Then*

$$\left\|\mathrm{Cov}(Y^{(1)}, Y^{(2)})\right\|_{\mathrm{op}} \leq \sqrt{\left\|\mathrm{Cov}(Y^{(1)})\right\|_{\mathrm{op}} \left\|\mathrm{Cov}(Y^{(2)})\right\|_{\mathrm{op}}} \,.$$

*Proof of Lemma 8:* Let $\Sigma := \mathrm{Cov}(Y)$. Then $\Sigma$ can be written in block form as

$$\Sigma = \begin{pmatrix} \Sigma_{11} & \Sigma_{12} \\ \Sigma_{21} & \Sigma_{22} \end{pmatrix} \,,$$

where

$$\Sigma_{11} = \mathrm{Cov}(Y^{(1)}) \,, \quad \Sigma_{22} = \mathrm{Cov}(Y^{(2)}) \,, \quad \Sigma_{12} = \mathrm{Cov}(Y^{(1)}, Y^{(2)}) \,, \quad \Sigma_{21} = \Sigma_{12}^T \,.$$

Recall that, for any matrix A or order $d$, we have

$$\|A\|_{\mathrm{op}} = \sup_{u,v\in\mathbb{R}^d, \|u\|, \|v\|=1} |u^\top A v| . \tag{20}$$

Let $u, v \in \mathbb{R}^d$ be arbitrary unit vectors. Being $\Sigma$ positive semi-definite, for any $t \in \mathbb{R}$, we have that

$$\begin{pmatrix} u \\ tv \end{pmatrix}^T \Sigma \begin{pmatrix} u \\ tv \end{pmatrix} = u^T \Sigma_{11} u + \left( 2 u^T \Sigma_{12} v \right) t + \left( v^T \Sigma_{22} v \right) t^2 \geq 0 .$$

This implies that the discriminant of the above second order polynomial in $t$ is always non-positive, *i.e.*,

$$\left( 2\, u^T \Sigma_{12} v \right)^2 - 4 \left( u^T \Sigma_{11} u \right) (v^T \Sigma_{22} v) \leq 0 ,$$

or equivalently,

$$(u^T \Sigma_{12} v)^2 \leq (u^T \Sigma_{11} u)(v^T \Sigma_{22} v) .$$

It follows from (20) that

$$|u^T \Sigma_{12} v| \leq \sqrt{\|\Sigma_{11}\|_{\mathrm{op}} \|\Sigma_{22}\|_{\mathrm{op}}} .$$

The thesis then follows from the arbitrary of the unit vectors $u, v \in \mathbb{R}^d$. $\qquad\square$

**Proposition 3.** *Assume that $\pi \in \Pi(\mu, \nu^\star)$ satisfies **H**7 and 9. Then, for any $s \in (0,1)$ and $x, y \in \mathbb{R}^d$, it holds*

$$\|\tilde{\beta}_s(x) - \tilde{\beta}_s(y)\| \leq \frac{4\sqrt{2}}{\sqrt{\alpha_\pi}} \|\nabla^2 \log \pi\|_{\mathrm{L}^2(\pi)} \frac{1}{\sqrt{s(1-s)}} \|x - y\| .$$

*Proof of Proposition 3:* Few computations lead to

$$
\begin{aligned}
&D_x \tilde{\beta}_s(x) \\
&= 2 \frac{\int_{\mathbb{R}^{2d}} \nabla_x p_{1-s}(x_1|x)(\nabla_x p_s(x|x_0))^{\mathrm{T}} \tilde{\pi}(x_0, x_1) \mathrm{d}x_0 \mathrm{d}x_1}{p_s^{\mathrm{I}}(x)} \\
&+ 2 \frac{\int_{\mathbb{R}^{2d}} p_s(x|x_0) \nabla_x^2 p_{1-s}(x_1|x) \tilde{\pi}(x_0, x_1) \mathrm{d}x_0 \mathrm{d}x_1}{p_s^{\mathrm{I}}(x)} \\
&- 2 \Big( \frac{\int_{\mathbb{R}^{2d}} p_s(x|x_0) \nabla_x p_{1-s}(x_1|x) \tilde{\pi}(x_0, x_1) \mathrm{d}x_0 \mathrm{d}x_1}{p_s^{\mathrm{I}}(x)} \Big) \Big( \frac{\int_{\mathbb{R}^{2d}} \nabla_x p_s(x|x_0) p_{1-s}(x_1|x) \tilde{\pi}(x_0, x_1) \mathrm{d}x_0 \mathrm{d}x_1}{p_s^{\mathrm{I}}(x)} \Big)^{\mathrm{T}} \\
&- 2 \Big( \frac{\int_{\mathbb{R}^{2d}} p_s(x|x_0) \nabla_x p_{1-s}(x_1|x) \tilde{\pi}(x_0, x_1) \mathrm{d}x_0 \mathrm{d}x_1}{p_s^{\mathrm{I}}(x)} \Big) \Big( \frac{\int_{\mathbb{R}^{2d}} p_s(x|x_0) \nabla_x p_{1-s}(x_1|x) \tilde{\pi}(x_0, x_1) \mathrm{d}x_0 \mathrm{d}x_1}{p_s^{\mathrm{I}}(x)} \Big)^{\mathrm{T}} .
\end{aligned}
$$

Using (12) and integration by parts formula, we get

$$
\begin{aligned}
&D_x \tilde{\beta}_s(x) \\
&= \frac{\int_{\mathbb{R}^{2d}} \frac{\nabla_{x_1}\tilde{\pi}}{\tilde{\pi}}(x_0,x_1)(\frac{x_0-x}{s})^{\mathrm{T}} p_s(x|x_0) p_{1-s}(x_1|x)\tilde{\pi}(x_0,x_1)\mathrm{d}x_0\mathrm{d}x_1}{p_s^{\mathrm{I}}(x)} \\
&+ \frac{\int_{\mathbb{R}^{2d}} \frac{x_1-x}{1-s}(\frac{\nabla_{x_1}\tilde{\pi}}{\tilde{\pi}}(x_0,x_1))^{\mathrm{T}} p_s(x|x_0) p_{1-s}(x_1|x)\tilde{\pi}(x_0,x_1)\mathrm{d}x_0\mathrm{d}x_1}{p_s^{\mathrm{I}}(x)} \\
&- \left( \frac{\int_{\mathbb{R}^{2d}} \frac{\nabla_{x_1}\tilde{\pi}}{\tilde{\pi}}(x_0,x_1) p_s(x|x_0) p_{1-s}(x_1|x)\tilde{\pi}(x_0,x_1)\mathrm{d}x_0\mathrm{d}x_1}{p_s^{\mathrm{I}}(x)} \right) \\
&\quad \cdot \left( \frac{\int_{\mathbb{R}^{2d}} \frac{x_0-x}{s} p_s(x|x_0) p_{1-s}(x_1|x)\tilde{\pi}(x_0,x_1)\mathrm{d}x_0\mathrm{d}x_1}{p_s^{\mathrm{I}}(x)} \right)^{\mathrm{T}} \\
&- \left( \frac{\int_{\mathbb{R}^{2d}} \frac{x_1-x}{1-s} p_s(x|x_0) p_{1-s}(x_1|x)\tilde{\pi}(x_0,x_1)\mathrm{d}x_0\mathrm{d}x_1}{p_s^{\mathrm{I}}(x)} \right) \\
&\quad \cdot \left( \frac{\int_{\mathbb{R}^{2d}} \frac{\nabla_{x_1}\tilde{\pi}}{\tilde{\pi}}(x_0,x_1) p_s(x|x_0) p_{1-s}(x_1|x)\tilde{\pi}(x_0,x_1)\mathrm{d}x_0\mathrm{d}x_1}{p_s^{\mathrm{I}}(x)} \right)^{\mathrm{T}} \\
&= \mathrm{Cov}_{(X_0^{\mathrm{I}},X_1^{\mathrm{I}})\sim \mathrm{K}_s(x,\cdot)}\left( \frac{X_0^{\mathrm{I}}-x}{s}, \frac{\nabla_{x_1}\tilde{\pi}}{\tilde{\pi}}(X_0^{\mathrm{I}},X_1^{\mathrm{I}})\right) + \mathrm{Cov}_{(X_0^{\mathrm{I}},X_1^{\mathrm{I}})\sim \mathrm{K}_s(x,\cdot)}\left( \frac{X_1^{\mathrm{I}}-x}{1-s}, \frac{\nabla_{x_1}\tilde{\pi}}{\tilde{\pi}}(X_0^{\mathrm{I}},X_1^{\mathrm{I}})\right). \quad (21)
\end{aligned}
$$

Therefore, using Lemma 8, **H**7 and 9, Lemma 6, Brascamp-Lieb inequality (see Lemma 2 in (Chewi & Pooladian, 2023)) and Lemma 2, we get

$$
\begin{aligned}
\|D_x \tilde{\beta}_s(x)\|_{\mathrm{op}} &\leq \left( \sqrt{\left\| \mathrm{Cov}_{X_0^{\mathrm{I}}\sim \mathrm{K}_s^{\#0}(x,\cdot)}\left( \frac{X_0^{\mathrm{I}}-x}{s}\right)\right\|_{\mathrm{op}}} + \sqrt{\left\| \mathrm{Cov}_{X_1^{\mathrm{I}}\sim \mathrm{K}_s^{\#1}(x,\cdot)}\left( \frac{X_1^{\mathrm{I}}-x}{1-s}\right)\right\|_{\mathrm{op}}} \right) \\
&\quad \cdot \sqrt{\left\| \mathrm{Cov}_{(X_0^{\mathrm{I}},X_1^{\mathrm{I}})\sim \mathrm{K}_s(x,\cdot)}\left( \nabla\log\tilde{\pi}(X_0^{\mathrm{I}},X_1^{\mathrm{I}})\right)\right\|_{\mathrm{op}}} \\
&\preceq \left( \frac{1}{s}\sqrt{\frac{2s}{2\alpha_\pi s+1-s}} + \frac{1}{1-s}\sqrt{\frac{2(1-s)}{s+2\alpha_\pi(1-s)}} \right) \sqrt{\frac{\|\nabla\log\tilde{\pi}\|_{\mathrm{L}^2(\pi)}^2}{\alpha_\pi}} \\
&\leq \frac{4\sqrt{2}}{\sqrt{\alpha_\pi}}\|\nabla^2\log\pi\|_{\mathrm{L}^2(\pi)}\frac{1}{\sqrt{s(1-s)}}.
\end{aligned}
$$

Hence, we have that

$$
\|D_x \tilde{\beta}_s(x)\|_{\mathrm{op},\infty} \leq \frac{4\sqrt{2}}{\sqrt{\alpha_\pi}}\|\nabla^2\log\pi\|_{\mathrm{L}^2(\pi)}\frac{1}{\sqrt{s(1-s)}}.
$$

The thesis follows directly. $\qquad\square$

### A.5.2. WEAKLY LOG CONCAVE CASE

**Proposition 4.** *Assume that $\pi \in \Pi(\mu,\nu^\star)$ satisfies **H**6 and 7. Then, for any $s \in (0,1)$ and $x,y \in \mathbb{R}^d$, it holds*

$$
\|\tilde{\beta}_s(x) - \tilde{\beta}_s(y)\| \leq \frac{4\sqrt{2}}{\sqrt{\alpha_\pi}}\exp\left(\frac{M_\pi}{\alpha_\pi}\right)\|\nabla^2\log\pi\|_{\mathrm{L}^2(\pi)}\frac{1}{\sqrt{s(1-s)}}.
$$

*Proof of Proposition 4:* Proceed as in the proof of Proposition 3, to get (21) and therefore

$$
\begin{aligned}
\|D_x \tilde{\beta}_s(x)\|_{\mathrm{op}} &\leq \left( \sqrt{\left\| \mathrm{Cov}_{X_0^{\mathrm{I}}\sim \mathrm{K}_s^{\#0}(x,\cdot)}\left( \frac{X_0^{\mathrm{I}}-x}{s}\right)\right\|_{\mathrm{op}}} + \sqrt{\left\| \mathrm{Cov}_{X_1^{\mathrm{I}}\sim \mathrm{K}_s^{\#1}(x,\cdot)}\left( \frac{X_1^{\mathrm{I}}-x}{1-s}\right)\right\|_{\mathrm{op}}} \right) \\
&\quad \cdot \sqrt{\left\| \mathrm{Cov}_{(X_0^{\mathrm{I}},X_1^{\mathrm{I}})\sim \mathrm{K}_s(x,\cdot)}\left( \nabla\log\tilde{\pi}(X_0^{\mathrm{I}},X_1^{\mathrm{I}})\right)\right\|_{\mathrm{op}}}.
\end{aligned}
$$

For any $\alpha > 0, M \geq 0$, denote by

$$\xi(\alpha, M) := \frac{\alpha}{\exp\left(M/\alpha\right)} \ . \tag{22}$$

Also, fix $s \in (0,1)$ and $y_s \in \mathbb{R}^d$. The byproduct of Lemma 7 and Theorem 5.7 in (Conforti et al., 2023) implies that $\mathrm{K}_s(y_s, \cdot)$, $\mathrm{K}_s^{\#0}(y_s, \cdot)$ and $\mathrm{K}_s^{\#1}(y_s, \cdot)$ satisfy log-Sobolev inequality with parameters $\xi^{-1}(\alpha_\pi, M_\pi)$, $\xi^{-1}(\alpha_\pi + (1-s)/(2s), M_\pi)$ and $\xi^{-1}(\alpha_\pi + s/(2(1-s)), M_\pi)$ respectively, with $\xi(\alpha, M)$ for $\alpha > 0, M \geq 0$ defined in (22). Consequently, as shown in (Villani et al., 2008), $\mathrm{K}_s(y_t, \cdot)$, $\mathrm{K}_s^{\#0}(y_s, \cdot)$ and $\mathrm{K}_s^{\#1}(y_s, \cdot)$ satisfy Poincaré inequality with constants $2\xi(\alpha_\pi, M_\pi)$, $2\xi^{-1}(\alpha_\pi + (1-s)/(2s), M_\pi)$ and $2\xi^{-1}(\alpha_\pi + s/(2(1-s)), M_\pi)$ respectively. We therefore get that

$$\|D_x \tilde{\beta}_s(x)\|_{\mathrm{op}}$$
$$\leq \left(\frac{1}{s}\sqrt{2\xi^{-1}\left(\alpha_\pi + \frac{1-s}{2s}, M_\pi\right)} + \frac{1}{1-s}\sqrt{2\xi^{-1}\left(\alpha_\pi + \frac{s}{2(1-s)}, M_\pi\right)}\right)\|\nabla^2 \log \pi\|_{\mathrm{L}^2(\pi)}\sqrt{2\xi^{-1}(\alpha_\pi, M_\pi)}$$
$$= \left(\frac{2\sqrt{2}}{\sqrt{s}}\frac{1}{\sqrt{2s\alpha_\pi + 1 - s}}\exp\left(\frac{M_\pi}{\alpha_\pi + (1-s)/(2s)}\right) + \frac{2\sqrt{2}}{\sqrt{1-s}}\frac{1}{\sqrt{2(1-s)\alpha_\pi + s}}\exp\left(\frac{M_\pi}{\alpha_\pi + s/(2(1-s))}\right)\right)\|\nabla^2 \log \pi\|_{\mathrm{L}^2(\pi)}$$
$$\leq \frac{4\sqrt{2}}{\sqrt{\alpha_\pi}}\exp\left(\frac{M_\pi}{\alpha_\pi}\right)\|\nabla^2 \log \pi\|_{\mathrm{L}^2(\pi)}\frac{1}{\sqrt{s(1-s)}} \ .$$

$\square$

# B. Convergence Bounds in Kullback–Leibler Divergence

## B.1. Non-early-stopping regime with constant step-size

*Proof of Theorem 1:* For any $t \in [0,1]$, we denote by $\nu_t^\star = \mathrm{Law}(X_t^{\mathrm{M}})$ and $\nu_t^{\theta^\star} = \mathrm{Law}(X_t^{\theta^\star})$. Also, we denote by $\mathcal{L}^{\mathrm{M}}$ the generator of $(X_t^{\mathrm{M}})_{t \in [0,1]}$, which is defined for any $t \in [0,1]$ and $\rho \in C^2(\mathbb{R}^d)$ as

$$\mathcal{L}_t^{\mathrm{M}}\rho := \langle \nabla_x \rho, \tilde{\beta}_t \rangle + \Delta_x \rho \ .$$

We fix $0 < \epsilon_1 < \min\{h, 1/2\}$. First, using the data processing inequality (see, *e.g.*, Lemma 1.6 in (Nutz, 2021)), the standard decomposition of the KL divergence (Chen et al., 2023c;a; Conforti et al., 2025) based on Girsanov theorem, triangle inequality and **H**1, we bound the KL divergence between $\nu_{1-\epsilon_1}^\star$ and $\nu_{1-\epsilon_1}^{\theta^\star}$ as follows

$$\mathrm{KL}(\nu_{1-\epsilon_1}^\star | \nu_{1-\epsilon_1}^{\theta^\star}) \tag{23}$$
$$\leq \mathrm{KL}(\mathrm{Law}((X_t^{\mathrm{M}})_{t \in [0, 1-\epsilon_1]}) | \mathrm{Law}((X_t^{\theta^\star})_{t \in [0, 1-\epsilon_1]}))$$
$$\lesssim \sum_{k=0}^{N-2}\int_{t_k}^{t_{k+1}}\mathbb{E}\left[\left\|s_{\theta^\star}(t_k, X_{t_k}^{\mathrm{M}}) - \tilde{\beta}_t(X_t^{\mathrm{M}})\right\|^2\right]\mathrm{d}t + \int_{1-h}^{1-\epsilon_1}\mathbb{E}\left[\left\|s_{\theta^\star}(1-h, X_{1-h}^{\mathrm{M}}) - \tilde{\beta}_t(X_t^{\mathrm{M}})\right\|^2\right]\mathrm{d}t$$
$$\lesssim \varepsilon^2 + \sum_{k=0}^{N-2}\int_{t_k}^{t_{k+1}}\mathbb{E}\left[\left\|\tilde{\beta}_{t_k}(X_{t_k}^{\mathrm{M}}) - \tilde{\beta}_t(X_t^{\mathrm{M}})\right\|^2\right]\mathrm{d}t + \int_{1-h}^{1-\epsilon_1}\mathbb{E}\left[\left\|\tilde{\beta}_{1-h}(X_{1-h}^{\mathrm{M}}) - \tilde{\beta}_t(X_t^{\mathrm{M}})\right\|^2\right]\mathrm{d}t \ .$$

Second, we aim at bounding the RHS of (23) uniformly in $\epsilon_1$. Indeed, if we assume to be able to bound it with a constant $A$ independent of $\epsilon_1$, then, using the weak convergence of $X_{1-\epsilon_1}^{\mathrm{M}}$ to $X_1^{\mathrm{I}}$ (whose law is given by $\nu^\star$) as $\epsilon_1 \to 0$, the continuity of $(X_t^{\theta^\star})_{t \in [0,1]}$, hence the weak convergence of $X_{1-\epsilon_1}^{\theta^\star}$ to $X^{\theta^\star}$ (whose law is given by $\nu_1^{\theta^\star}$) as $\epsilon_1 \to 0$, and the lower semi-continuity of the KL-divergence with respect to the weak convergence (see, *e.g.*, Theorem 19 in (Van Erven & Harremos, 2014)), we will get

$$\mathrm{KL}(\nu^\star | \nu_1^{\theta^\star}) \leq \liminf_{\epsilon_1 \to 0}\mathrm{KL}(\nu_{1-\epsilon_1}^\star | \nu_{1-\epsilon_1}^{\theta^\star}) \lesssim \liminf_{\epsilon_1 \to 0}A = A \ . \tag{24}$$

Let us therefore bound the RHS of (23). We will do so by using stochastic calculus tools, and, more precisely, Ito's formula. This formula yields

$$\mathrm{d}\tilde{\beta}_t(X_t^{\mathrm{M}}) = (\partial_t + \mathcal{L}_t^{\mathrm{M}})\tilde{\beta}_t(X_t^{\mathrm{M}})\mathrm{d}t + \sqrt{2}D_x\tilde{\beta}_t(X_t^{\mathrm{M}})\mathrm{d}W_t , \quad t \in [0, 1 - \epsilon_1] .$$

So, applying Young inequality and Ito's isometry, we have that, for any $k = 0, \cdots, N - 1$

$$\mathbb{E}\Big[ \big\| \tilde{\beta}_{t_k}(X_{t_k}^{\mathrm{M}}) - \tilde{\beta}_t(X_t^{\mathrm{M}}) \big\|^2 \Big]$$

$$= \mathbb{E}\bigg[ \bigg\| \int_{t_k}^t (\partial_s + \mathcal{L}_s^{\mathrm{M}})\tilde{\beta}_s(X_s^{\mathrm{M}})\mathrm{d}s + \sqrt{2} \int_{t_k}^t D_x\tilde{\beta}_s(X_s^{\mathrm{M}})\mathrm{d}W_s \bigg\|^2 \bigg]$$

$$\lesssim \mathbb{E}\bigg[ \bigg\| \int_{t_k}^t (\partial_s + \mathcal{L}_s^{\mathrm{M}})\tilde{\beta}_s(X_s^{\mathrm{M}})\mathrm{d}s \bigg\|^2 \bigg] + 2 \int_{t_k}^{t_{k+1}} \mathbb{E}\Big[ \big\| D_x\tilde{\beta}_s(X_s^{\mathrm{M}}) \big\|^2 \Big]\mathrm{d}s .$$

We now bound separately the two upper addends. To do so, we introduce the auxiliary measures $\lambda_k^h(\mathrm{d}s) \in \mathcal{P}([t_k, t_{k+1}])$ for $k = 0, ..., N - 2$ and $\lambda_{N-1}^h(\mathrm{d}s) \in \mathcal{P}([1 - h, 1 - \epsilon_1])$ which will help us, via a double change of measure argument, to mitigate the bad behaviour at $t = 0$ and $t = 1$ of the reciprocal characteristic of the mimicking drift (*i.e.*, $\partial_s + \mathcal{L}_s^{\mathrm{M}}$), which is the trickiest addend. Namely, for $k = 0, ..., N - 2$ we consider the measures $\lambda_k^h(\mathrm{d}s) \in \mathcal{P}([t_k, t_{k+1}])$ defined as

$$\lambda_k^h(\mathrm{d}s) = \frac{\rho(s)^{-1}}{Z_k} \mathbb{1}_{[t_k, t_{k+1}]}\mathrm{d}s ,$$

with

$$\rho(s)^{-1} = s^{-7/8}\mathbb{1}_{\{s \le 1/2\}} + (1 - s)^{-7/8}\mathbb{1}_{\{s > 1/2\}} , \tag{25}$$

and

$$Z_k = \int_{\min\{t_k, 1/2\}}^{\min\{t_{k+1}, 1/2\}} r^{-7/8}\mathrm{d}r + \int_{\max\{t_k, 1/2\}}^{\max\{t_{k+1}, 1/2\}} (1 - r)^{-7/8}\mathrm{d}r .$$

For $k = N - 1$, we consider the measure $\lambda_{N-1}^h(\mathrm{d}s) \in \mathcal{P}([1 - h, 1 - \epsilon_1])$ defined as

$$\lambda_{N-1}^h(\mathrm{d}s) = \frac{\rho(s)^{-1}}{Z_{N-1}} \mathbb{1}_{[1-h, 1-\epsilon_1]}\mathrm{d}s ,$$

with $\rho(s)^{-1}$ as in (25) and

$$Z_{N-1} = \int_{\min\{1-h, 1/2\}}^{\min\{1-\epsilon_1, 1/2\}} r^{-7/8}\mathrm{d}r + \int_{\max\{1-h, 1/2\}}^{\max\{1-\epsilon_1, 1/2\}} (1 - r)^{-7/8}\mathrm{d}r .$$

Note that, for any $s \in [0, 1 - \epsilon_1]$ and for any $k = 0, ..., N - 1$, they hold

$$\rho(s) \lesssim 1 , \quad Z_k \lesssim h^{1/8} , \quad \int_0^{1-\epsilon_1} \rho^{-1}(s)\mathrm{d}s \lesssim 1 . \tag{26}$$

We start by bounding the first addend, that is the one that involves the reciprocal characteristic of the mimicking drift. With a first change of measure argument, we get for any $k = 0, ..., N - 1$,

$$\mathbb{E}\bigg[ \bigg\| \int_{t_k}^t (\partial_s + \mathcal{L}_s^{\mathrm{M}})\tilde{\beta}_s(X_s^{\mathrm{M}})\mathrm{d}s \bigg\|^2 \bigg] = Z_k^2\mathbb{E}\bigg[ \bigg\| \int_{t_k}^t (\partial_s + \mathcal{L}_s^{\mathrm{M}})\tilde{\beta}_s(X_s^{\mathrm{M}})\rho(s)\lambda_k^h(\mathrm{d}s) \bigg\|^2 \bigg] ,$$

where, in the last inequality, we used (26). But then, if we apply Jensen inequality and use an other change of measure argument, we get

$$\mathbb{E}\bigg[ \bigg\| \int_{t_k}^t (\partial_s + \mathcal{L}_s^{\mathrm{M}})\tilde{\beta}_s(X_s^{\mathrm{M}})\mathrm{d}s \bigg\|^2 \bigg] \le Z_k^2\mathbb{E}\bigg[ \int_{t_k}^t \big\| (\partial_s + \mathcal{L}_s^{\mathrm{M}})\tilde{\beta}_s(X_s^{\mathrm{M}}) \big\|^2 \rho(s)^2\lambda_k^h(\mathrm{d}s) \bigg] \tag{27}$$

$$\le Z_k \int_{t_k}^t \mathbb{E}\Big[ \big\| (\partial_s + \mathcal{L}_s^{\mathrm{M}})\tilde{\beta}_s(X_s^{\mathrm{M}}) \big\|^2 \Big]\rho(s)\mathrm{d}s$$

$$\lesssim h^{1/8} \int_{t_k}^{t_{k+1}} \mathbb{E}\Big[ \big\| (\partial_s + \mathcal{L}_s^{\mathrm{M}})\tilde{\beta}_s(X_s^{\mathrm{M}}) \big\|^2 \Big]\rho(s)\mathrm{d}s .$$

Let us now focus on the second addend. Remarkably, this addend can be bounded via the reciprocal characteristic of $\tilde{\beta}$: because of Ito's formula, for $t \in [0, 1 - \epsilon_1]$, it holds true

$$\mathrm{d}\left\|\tilde{\beta}_t(X_t^{\mathrm{M}})\right\|^2 = \left\{2\langle\tilde{\beta}_t, (\partial_t + \mathcal{L}_t^{\mathrm{M}})\tilde{\beta}_t\rangle + 2\left\|D_x\tilde{\beta}_t\right\|^2\right\}(X_t^{\mathrm{M}})\mathrm{d}t + 2\sqrt{2}\langle\tilde{\beta}_t, D_x\tilde{\beta}_t\rangle(X_t^{\mathrm{M}})\mathrm{d}W_t .$$

Note that, under **H**2 for $\mu$ and $\nu^\star$ and **H**3 for $\tilde{\pi}$, Lemma 4 in (Gentiloni Silveri et al., 2024) ensures that the process $(\int_0^s \langle\tilde{\beta}_t, D_x\tilde{\beta}_t\rangle(X_t^{\mathrm{M}})\mathrm{d}W_t)_{s\in[0,1-\epsilon_1]}$ is a true martingale. Consequently, we have that

$$2\int_{t_k}^{t_{k+1}} \mathbb{E}\Big[\left\|D_x\tilde{\beta}_s(X_s^{\mathrm{M}})\right\|^2\Big]\mathrm{d}s$$
$$\leq \mathbb{E}\Big[\left\|\tilde{\beta}_{t_{k+1}}(X_{t_{k+1}}^{\mathrm{M}})\right\|^2\Big] - \mathbb{E}\Big[\left\|\tilde{\beta}_{t_k}(X_{t_k}^{\mathrm{M}})\right\|^2\Big] + 2\Big|\int_{t_k}^{t_{k+1}} \mathbb{E}[\langle\tilde{\beta}_s(X_s^{\mathrm{M}}), (\partial_s + \mathcal{L}_s^{\mathrm{M}})\tilde{\beta}_s(X_s^{\mathrm{M}})\rangle]\mathrm{d}s\Big| .$$

But then, using, as before, a double change of measure argument and applying Cauchy-Schwartz inequality, we can bound the above expression as follows

$$2\int_{t_k}^{t_{k+1}} \mathbb{E}\Big[\left\|D_x\tilde{\beta}_s(X_s^{\mathrm{M}})\right\|^2\Big]\mathrm{d}s$$
$$\leq \mathbb{E}\Big[\left\|\tilde{\beta}_{t_{k+1}}(X_{t_{k+1}}^{\mathrm{M}})\right\|^2\Big] - \mathbb{E}\Big[\left\|\tilde{\beta}_{t_k}(X_{t_k}^{\mathrm{M}})\right\|^2\Big] + 2Z_k\Big|\int_{t_k}^{t_{k+1}} \mathbb{E}[\langle\tilde{\beta}_s(X_s^{\mathrm{M}}), (\partial_s + \mathcal{L}_s^{\mathrm{M}})\tilde{\beta}_s(X_s^{\mathrm{M}})\rangle]\rho(s)\lambda_k^h(\mathrm{d}s)\Big|$$
$$\leq \mathbb{E}\Big[\left\|\tilde{\beta}_{t_{k+1}}(X_{t_{k+1}}^{\mathrm{M}})\right\|^2\Big] - \mathbb{E}\Big[\left\|\tilde{\beta}_{t_k}(X_{t_k}^{\mathrm{M}})\right\|^2\Big] + Z_k\int_{t_k}^{t_{k+1}} \mathbb{E}\Big[\left\|\tilde{\beta}_s(X_s^{\mathrm{M}})\right\|^2\Big]\lambda_k^h(\mathrm{d}s)$$
$$+ Z_k\int_{t_k}^{t_{k+1}} \mathbb{E}\Big[\left\|(\partial_s + \mathcal{L}_s^{\mathrm{M}})\tilde{\beta}_s(X_s^{\mathrm{M}})\right\|^2\Big]\rho(s)^2\lambda_k^h(\mathrm{d}s)$$
$$= \mathbb{E}\Big[\left\|\tilde{\beta}_{t_{k+1}}(X_{t_{k+1}}^{\mathrm{M}})\right\|^2\Big] - \mathbb{E}\Big[\left\|\tilde{\beta}_{t_k}(X_{t_k}^{\mathrm{M}})\right\|^2\Big] + \int_{t_k}^{t_{k+1}} \mathbb{E}\Big[\left\|\tilde{\beta}_s(X_s^{\mathrm{M}})\right\|^2\Big]\rho(s)^{-1}\mathrm{d}s$$
$$+ \int_{t_k}^{t_{k+1}} \mathbb{E}\Big[\left\|(\partial_s + \mathcal{L}_s^{\mathrm{M}})\tilde{\beta}_s(X_s^{\mathrm{M}})\right\|^2\Big]\rho(s)\mathrm{d}s .$$

Plugging this bound and (27) in (23), we get

$$\mathrm{KL}(\nu_{1-\epsilon_1}^\star|\nu_{1-\epsilon_1}^{\theta^\star}) \tag{28}$$
$$\lesssim \varepsilon^2 + h\mathbb{E}\Big[\left\|\tilde{\beta}_{1-\epsilon_1}(X_{1-\epsilon_1}^{\mathrm{M}})\right\|^2\Big] + h\int_0^{1-\epsilon_1} \mathbb{E}\Big[\left\|\tilde{\beta}_s(X_s^{\mathrm{M}})\right\|^2\Big]\rho(s)^{-1}\mathrm{d}s$$
$$+ h(h^{1/8} + 1)\int_0^{1-\epsilon_1} \mathbb{E}\Big[\left\|(\partial_s + \mathcal{L}_s^{\mathrm{M}})\tilde{\beta}_s(X_s^{\mathrm{M}})\right\|^2\Big]\rho(s)\mathrm{d}s .$$

We now compute explicitly and upper bound each term appearing in the RHS of (28), recalling that, because of Theorem 1 in (Gentiloni Silveri et al., 2024), for any $s \in [0, 1]$, $\nu_s^\star = \mathrm{Law}(X_s^{\mathrm{I}})$. We start with the second term, that is

$$h\mathbb{E}\Big[\left\|\tilde{\beta}_{1-\epsilon_1}(X_{1-\epsilon_1}^{\mathrm{M}})\right\|^2\Big] .$$

Using (12), integration by part, Jensen inequality, Lemma 2 and $\mathbf{m}_8[\mu], \mathbf{m}_8[\nu^\star] \lesssim d^4$, we get

$$
\begin{aligned}
&\mathbb{E}\left[\left\|\tilde{\beta}_{1-\epsilon_1}(X_{1-\epsilon_1}^{\mathrm{M}})\right\|^2\right] \\
&\int_{\mathbb{R}^d}\left\|\frac{\int_{\mathbb{R}^{2d}} p_{1-\epsilon_1}(x|x_0)\nabla_x p_{\epsilon_1}(x_1|x)\tilde{\pi}(x_0,x_1)\mathrm{d}x_1\mathrm{d}x_0}{p_{1-\epsilon_1}^{\mathrm{I}}(x)}\right\|^2 p_{1-\epsilon_1}^{\mathrm{I}}(x)\mathrm{d}x \\
&\lesssim \int_{\mathbb{R}^d}\left\|\frac{\int_{\mathbb{R}^{2d}} p_{1-\epsilon_1}(x|x_0)p_{\epsilon_1}(x_1|x)(\nabla_{x_1}\tilde{\pi}(x_0,x_1)/\tilde{\pi}(x_0,x_1))\tilde{\pi}(x_0,x_1)\mathrm{d}x_0\mathrm{d}x_1}{p_{1-\epsilon_1}^{\mathrm{I}}(x)}\right\|^2 p_{1-\epsilon_1}^{\mathrm{I}}(x)\mathrm{d}x \\
&= \mathbb{E}\left[\left\|\mathbb{E}\left[\frac{\nabla_{x_1}\tilde{\pi}}{\tilde{\pi}}(X_0^{\mathrm{I}}, X_1^{\mathrm{I}})\Big| X_{1-\epsilon_1}^{\mathrm{I}}\right]\right\|^2\right] \leq \mathbb{E}\left[\mathbb{E}\left[\left\|\frac{\nabla_{x_1}\tilde{\pi}}{\tilde{\pi}}(X_0^{\mathrm{I}}, X_1^{\mathrm{I}})\right\|^2\Big| X_{1-\epsilon_1}^{\mathrm{I}}\right]\right] \\
&= \mathbb{E}\left[\left\|\frac{\nabla_{x_1}\tilde{\pi}}{\tilde{\pi}}(X_0^{\mathrm{I}}, X_1^{\mathrm{I}})\right\|^2\right] = \left\|\frac{\nabla_{x_1}\tilde{\pi}}{\tilde{\pi}}\right\|_{\mathrm{L}^2(\pi)}^2 \leq \|\nabla\log\tilde{\pi}\|_{\mathrm{L}^2(\pi)}^2 \lesssim \|\nabla\log\pi\|_{\mathrm{L}^2(\pi)}^2 + \mathbf{m}_2[\mu] + \mathbf{m}_2[\nu^\star] \\
&\lesssim \|\nabla\log\pi\|_{\mathrm{L}^2(\pi)}^2 + d\,.
\end{aligned}
\tag{29}
$$

Hence, we can bound the second term in (28) as follows

$$
h\mathbb{E}\left[\left\|\tilde{\beta}_{1-\epsilon_1}(X_{1-\epsilon_1}^{\mathrm{M}})\right\|^2\right] \lesssim h\left(\|\nabla\log\pi\|_{\mathrm{L}^2(\pi)}^2 + d\right)\,.
\tag{30}
$$

We now deal with the third term of the RHS of (28), *i.e.*,

$$
h\int_0^{1-\epsilon_1}\mathbb{E}\left[\left\|\tilde{\beta}_s(X_s^{\mathrm{M}})\right\|^2\right]\rho(s)^{-1}\mathrm{d}s\,.
$$

Note that, proceeding as before and using (26), we have that

$$
\begin{aligned}
&\int_0^{1-\epsilon_1}\mathbb{E}\left[\left\|\tilde{\beta}_s(X_s^{\mathrm{M}})\right\|^2\right]\rho(s)^{-1}\mathrm{d}s \\
&\lesssim \int_0^{1-\epsilon_1}\rho(s)^{-1}\int_{\mathbb{R}^d}\left\|\frac{\int_{\mathbb{R}^{2d}} p_s(x|x_0)\nabla_x p_{1-s}(x_1|x)\tilde{\pi}(x_0,x_1)\mathrm{d}x_0\mathrm{d}x_1}{p_s^{\mathrm{I}}(x)}\right\|^2 p_s^{\mathrm{I}}(x)\mathrm{d}x\mathrm{d}s \\
&= \int_0^{1-\epsilon_1}\rho(s)^{-1}\int_{\mathbb{R}^d}\left\|\frac{1}{p_s^{\mathrm{I}}(x)}\int_{\mathbb{R}^{2d}} p_s(x|x_0)p_{1-s}(x_1|x)\frac{\nabla_{x_1}\tilde{\pi}}{\tilde{\pi}}(x_0,x_1)\tilde{\pi}(x_0,x_1)\mathrm{d}x_0,\mathrm{d}x_1\right\|^2 p_s^{\mathrm{I}}(x)\mathrm{d}x \\
&= \int_0^{1-\epsilon_1}\rho(s)^{-1}\mathbb{E}\left[\left\|\mathbb{E}\left[\frac{\nabla_{x_1}\tilde{\pi}}{\tilde{\pi}}(X_0^{\mathrm{I}}, X_1^{\mathrm{I}})\Big| X_s^{\mathrm{I}}\right]\right\|^2\right]\mathrm{d}s \\
&\leq \int_0^{1-\epsilon_1}\rho(s)^{-1}\mathbb{E}\left[\mathbb{E}\left[\left\|\frac{\nabla_{x_1}\tilde{\pi}}{\tilde{\pi}}(X_0^{\mathrm{I}}, X_1^{\mathrm{I}})\right\|^2\Big| X_s^{\mathrm{I}}\right]\right]\mathrm{d}s \\
&= \int_0^{1-\epsilon_1}\rho(s)^{-1}\mathbb{E}\left[\left\|\frac{\nabla_{x_1}\tilde{\pi}}{\tilde{\pi}}(X_0^{\mathrm{I}}, X_1^{\mathrm{I}})\right\|^2\right]\mathrm{d}s \\
&= \int_0^{1-\epsilon_1}\rho(s)^{-1}\mathrm{d}s\,\|\nabla\log\tilde{\pi}\|_{\mathrm{L}^2(\pi)}^2 \lesssim \|\nabla\log\tilde{\pi}\|_{\mathrm{L}^2(\pi)}^2 \lesssim \|\nabla\log\pi\|_{\mathrm{L}^2(\pi)}^2 + \mathbf{m}_2[\mu] + \mathbf{m}_2[\nu^\star] \\
&\lesssim \|\nabla\log\pi\|_{\mathrm{L}^2(\pi)}^2 + d\,.
\end{aligned}
\tag{31}
$$

Therefore, we can bound the third term in (28) as follows

$$
h\int_0^{1-\epsilon_1}\mathbb{E}\left[\left\|\tilde{\beta}_s(X_s^{\mathrm{M}})\right\|^2\right]\rho(s)^{-1}\mathrm{d}s \lesssim h\left(\|\nabla\log\pi\|_{\mathrm{L}^2(\pi)}^2 + d\right)\,.
\tag{32}
$$

We now turn to the last term appearing in the r.h.s. of (28), *i.e.*,

$$h(h^{1/8} + 1) \int_0^{1-\epsilon_1} \mathbb{E}\Big[ \Big\| (\partial_s + \mathcal{L}_s^{\mathrm{M}}) \tilde{\beta}_s(X_s^{\mathrm{M}}) \Big\|^2 \Big] \rho(s)\mathrm{d}s$$

$$= h(h^{1/8} + 1) \int_0^{1-\epsilon_1} \int_{\mathbb{R}^d} \Big\| (\partial_s + \mathcal{L}_s^{\mathrm{M}}) \tilde{\beta}_s(x) \Big\|^2 \rho(s) p_s^{\mathrm{I}}(x)\mathrm{d}x \ \mathrm{d}s \ .$$

Some computations and (14) (we refer to Appendix A4 in (Gentiloni Silveri et al., 2024) for more details) lead to

$$(\partial_s + \mathcal{L}_s^{\mathrm{M}}) \tilde{\beta}_s(x) = \sum_{k=1}^6 A_s^k(x) \ ,$$

where we have defined

$$A_s^1(x) = 4 \frac{\int_{\mathbb{R}^{2d}} \Delta_x p_s(x|x_0) \nabla_x p_{1-s}(x_1|x) \tilde{\pi}(x_0, x_1)\mathrm{d}x_0\mathrm{d}x_1}{p_s^{\mathrm{I}}(x)}$$
$$- 4 \frac{\int_{\mathbb{R}^{2d}} \Delta_x p_s(x|x_0) p_{1-s}(x_1|x) \tilde{\pi}(x_0, x_1)\mathrm{d}x_0\mathrm{d}x_1 \int_{\mathbb{R}^{2d}} p_s(x|x_0) \nabla_x p_{1-s}(x_1|x) \tilde{\pi}(x_0, x_1)\mathrm{d}x_0\mathrm{d}x_1}{(p_s^{\mathrm{I}}(x))^2} \ ,$$

$$A_s^2(x) = 4 \frac{\int_{\mathbb{R}^{2d}} \nabla_x^2 p_{1-s}(x_1|x) \nabla_x p_s(x|x_0) \tilde{\pi}(x_0, x_1)\mathrm{d}x_0\mathrm{d}x_1}{p_s^{\mathrm{I}}(x)}$$
$$- 4 \frac{\int_{\mathbb{R}^{2d}} p_s(x|x_0) \nabla_x^2 p_{1-s}(x_1|x) \tilde{\pi}(x_0, x_1)\mathrm{d}x_0\mathrm{d}x_1 \int_{\mathbb{R}^{2d}} \nabla_x p_s(x|x_0) p_{1-s}(x_1|x) \tilde{\pi}(x_0, x_1)\mathrm{d}x_0\mathrm{d}x_1}{(p_s^{\mathrm{I}}(x))^2} \ .$$

$$A_s^3(x) = -4 \frac{\int_{\mathbb{R}^{2d}} \nabla_x p_{1-s}(x_1|x) (\nabla_x p_s(x|x_0))^{\mathrm{T}} \tilde{\pi}(x_0, x_1)\mathrm{d}x_0\mathrm{d}x_1}{p_s^{\mathrm{I}}(x)}$$
$$\cdot \frac{\int_{\mathbb{R}^{2d}} \nabla_x p_s(x|x_0) p_{1-s}(x_1|x) \tilde{\pi}(x_0, x_1)\mathrm{d}x_0\mathrm{d}x_1}{p_s^{\mathrm{I}}(x)} \ ,$$

$$A_s^4(x) = -4 \frac{\int_{\mathbb{R}^{2d}} \langle \nabla_x p_{1-s}(x_1|x), \nabla_x p_s(x|x_0) \rangle \tilde{\pi}(x_0, x_1)\mathrm{d}x_0\mathrm{d}x_1}{p_s^{\mathrm{I}}(x)}$$
$$\cdot \frac{\int_{\mathbb{R}^{2d}} p_s(x|x_0) \nabla_x p_{1-s}(x_1|x) \tilde{\pi}(x_0, x_1)\mathrm{d}x_0\mathrm{d}x_1}{p_s^{\mathrm{I}}(x)} \ ,$$

$$A_s^5(x) = 4 \frac{\Big\| \int_{\mathbb{R}^{2d}} \nabla_x p_s(x|x_0) p_{1-s}(x_1|x) \tilde{\pi}(x_0, x_1)\mathrm{d}x_0\mathrm{d}x_1 \Big\|^2}{(p_s^{\mathrm{I}}(x))^2}$$
$$\cdot \frac{\int_{\mathbb{R}^{2d}} p_s(x|x_0) \nabla_x p_{1-s}(x_1|x) \tilde{\pi}(x_0, x_1)\mathrm{d}x_0\mathrm{d}x_1}{p_s^{\mathrm{I}}(x)} \ ,$$

$$A_s^6(x) = 4 \frac{\int_{\mathbb{R}^{2d}} p_s(x|x_0) \nabla_x p_{1-s}(x_1|x) \tilde{\pi}(x_0, x_1)\mathrm{d}x_0\mathrm{d}x_1}{p_s^{\mathrm{I}}(x)}$$
$$\cdot \left( \frac{\int_{\mathbb{R}^{2d}} \nabla_x p_s(x|x_0) p_{1-s}(x_1|x) \tilde{\pi}(x_0, x_1)\mathrm{d}x_0\mathrm{d}x_1}{p_s^{\mathrm{I}}(x)} \right)^{\mathrm{T}} \frac{\int_{\mathbb{R}^{2d}} p_s(x|x_0) \nabla_x p_{1-s}(x_1|x) \tilde{\pi}(x_0, x_1)\mathrm{d}x_0\mathrm{d}x_1}{p_s^{\mathrm{I}}(x)} \ .$$

Therefore

$$h(h^{1/8} + 1) \int_0^{1-\epsilon_1} \mathbb{E}\Big[ \Big\| (\partial_s + \mathcal{L}_s^{\mathrm{M}}) \tilde{\beta}_s(X_s^{\mathrm{M}}) \Big\|^2 \Big] \rho(s)\mathrm{d}s \lesssim h(h^{1/8} + 1) \sum_{k=1}^6 \int_0^{1-\epsilon_1} \int_{\mathbb{R}^d} \Big\| A_s^k(x) \Big\|^2 \rho(s) p_s^{\mathrm{I}}(x)\mathrm{d}x\mathrm{d}s \ .$$

Plugging the above inequality, (30) and (32) in (28), we obtain

$$
\mathrm{KL}(\nu_{1-\epsilon_1}^{\star}|\nu_{1-\epsilon_1}^{\theta^{\star}}) \lesssim \varepsilon^2 + h\left\|\nabla \log \tilde{\pi}\right\|_{\mathrm{L}^2(\pi)}^2 + h(h^{1/8}+1)\sum_{k=1}^{6}\int_0^{1-\epsilon_1}\int_{\mathbb{R}^d}\left\|A_s^k(x)\right\|^2\rho(s)p_s^{\mathrm{I}}(x)\mathrm{d}x\mathrm{d}s \, .
$$

We now bound each term $A_s^k$ in the sum, starting with $A_s^1$. Integrating by parts and using (12) and (13), we get

$$
\begin{aligned}
&\int_0^{1-\epsilon_1}\int_{\mathbb{R}^d}\left\|A_s^1(x)\right\|^2\rho(s)p_s^{\mathrm{I}}(x)\mathrm{d}x\mathrm{d}s \\
&\lesssim \int_0^{1-\epsilon_1}\int_{\mathbb{R}^d}\left\|\frac{\int_{\mathbb{R}^{2d}}\Delta_x p_s(x|x_0)\nabla_x p_{1-s}(x_1|x)\tilde{\pi}(x_0,x_1)\mathrm{d}x_0\mathrm{d}x_1}{p_s^{\mathrm{I}}(x)}\right. \\
&\quad -\frac{\int_{\mathbb{R}^{2d}}\Delta_x p_s(x|x_0)p_{1-s}(x_1|x)\tilde{\pi}(x_0,x_1)\mathrm{d}x_0\mathrm{d}x_1}{p_s^{\mathrm{I}}(x)} \\
&\quad \left.\cdot\frac{\int_{\mathbb{R}^{2d}}p_s(x|x_0)\nabla_x p_{1-s}(x_1|x)\tilde{\pi}(x_0,x_1)\mathrm{d}x_0\mathrm{d}x_1}{p_s^{\mathrm{I}}(x)}\right\|^2\rho(s)p_s^{\mathrm{I}}(x)\mathrm{d}x\mathrm{d}s \\
&= \int_0^{1-\epsilon_1}\int_{\mathbb{R}^d}\left\|\frac{\int_{\mathbb{R}^{2d}}\langle\nabla_x p_s(x|x_0),\nabla_{x_0}\tilde{\pi}(x_0,x_1)\rangle\nabla_x p_{1-s}(x_1|x)\mathrm{d}x_0\mathrm{d}x_1}{p_s^{\mathrm{I}}(x)}\right. \\
&\quad -\frac{\int_{\mathbb{R}^{2d}}\langle\nabla_x p_s(x|x_0),\nabla_{x_0}\tilde{\pi}(x_0,x_1)\rangle p_{1-s}(x_1|x)\mathrm{d}x_0\mathrm{d}x_1}{p_s^{\mathrm{I}}(x)} \\
&\quad \left.\cdot\frac{\int_{\mathbb{R}^{2d}}p_s(x|x_0)\nabla_x p_{1-s}(x_1|x)\tilde{\pi}(x_0,x_1)\mathrm{d}x_0\mathrm{d}x_1}{p_s^{\mathrm{I}}(x)}\right\|^2\rho(s)p_s^{\mathrm{I}}(x)\mathrm{d}x\mathrm{d}s \\
&= \int_0^{1-\epsilon_1}\int_{\mathbb{R}^d}\left\|\frac{\int_{\mathbb{R}^{2d}}\left\langle\frac{x-x_0}{s},\frac{\nabla_{x_0}\tilde{\pi}}{\tilde{\pi}}\right\rangle(x_0,x_1)\frac{x_1-x}{1-s}p_s(x|x_0)p_{1-s}(x_1|x)\tilde{\pi}(x_0,x_1)\mathrm{d}x_0\mathrm{d}x_1}{p_s^{\mathrm{I}}(x)}\right. \\
&\quad -\frac{\int_{\mathbb{R}^{2d}}\left\langle\frac{x-x_0}{s},\frac{\nabla_{x_0}\tilde{\pi}}{\tilde{\pi}}(x_0,x_1)\right\rangle p_s(x|x_0)p_{1-s}(x_1|x)\tilde{\pi}(x_0,x_1)\mathrm{d}x_0\mathrm{d}x_1}{p_s^{\mathrm{I}}(x)} \\
&\quad \left.\cdot\frac{\int_{\mathbb{R}^{2d}}\frac{x_1-x}{1-s}p_s(x|x_0)p_{1-s}(x_1|x)\tilde{\pi}(x_0,x_1)\mathrm{d}x_0\mathrm{d}x_1}{p_s^{\mathrm{I}}(x)}\right\|^2\rho(s)p_s^{\mathrm{I}}(x)\mathrm{d}x\mathrm{d}s \\
&= \int_0^{1-\epsilon_1}\mathbb{E}\left[\left\|\mathbb{E}\left[\left\langle\frac{X_s^{\mathrm{I}}-X_0^{\mathrm{I}}}{s},\frac{\nabla_{x_0}\tilde{\pi}}{\tilde{\pi}}(X_0^{\mathrm{I}},X_1^{\mathrm{I}})\right\rangle\frac{X_1^{\mathrm{I}}-X_s^{\mathrm{I}}}{1-s}\Big|X_s^{\mathrm{I}}\right]\right.\right. \\
&\quad \left.\left.-\mathbb{E}\left[\left\langle\frac{X_s^{\mathrm{I}}-X_0^{\mathrm{I}}}{s},\frac{\nabla_{x_0}\tilde{\pi}}{\tilde{\pi}}(X_0^{\mathrm{I}},X_1^{\mathrm{I}})\right\rangle\Big|X_s^{\mathrm{I}}\right]\mathbb{E}\left[\frac{X_1^{\mathrm{I}}-X_s^{\mathrm{I}}}{1-s}\Big|X_s^{\mathrm{I}}\right]\right\|^2\right]\rho(s)\mathrm{d}s \\
&\lesssim \int_0^{1-\epsilon_1}\left\{\mathbb{E}\left[\left\|\mathbb{E}\left[\left\langle\frac{X_s^{\mathrm{I}}-X_0^{\mathrm{I}}}{s},\frac{\nabla_{x_0}\tilde{\pi}}{\tilde{\pi}}(X_0^{\mathrm{I}},X_1^{\mathrm{I}})\right\rangle\frac{X_1^{\mathrm{I}}-X_s^{\mathrm{I}}}{1-s}\Big|X_s^{\mathrm{I}}\right]\right\|^2\right]\right. \\
&\quad \left.+\mathbb{E}\left[\left\|\mathbb{E}\left[\left\langle\frac{X_s^{\mathrm{I}}-X_0^{\mathrm{I}}}{s},\frac{\nabla_{x_0}\tilde{\pi}}{\tilde{\pi}}(X_0^{\mathrm{I}},X_1^{\mathrm{I}})\right\rangle\Big|X_s^{\mathrm{I}}\right]\mathbb{E}\left[\frac{X_1^{\mathrm{I}}-X_s^{\mathrm{I}}}{1-s}\Big|X_s^{\mathrm{I}}\right]\right\|^2\right]\right\}\rho(s)\mathrm{d}s \, .
\end{aligned}
$$

At this point, we split the time interval $[0, 1 - \epsilon_1]$ in two, $[0, 1/2]$ and $[1/2, 1 - \epsilon_1]$ and we rewrite the RHS as

$$
\int_0^{1-\epsilon_1} \int_{\mathbb{R}^d} \left\| A_s^{\mathrm{I}}(x) \right\|^2 \rho(s) p_s^{\mathrm{I}}(x) \mathrm{d}x \mathrm{d}s
$$
$$
\lesssim \int_0^{1/2} \left\{ \mathbb{E}\left[ \left\| \left\langle \frac{X_s^{\mathrm{I}} - X_0^{\mathrm{I}}}{s}, \frac{\nabla_{x_0}\tilde{\pi}}{\tilde{\pi}}(X_0^{\mathrm{I}}, X_1^{\mathrm{I}}) \right\rangle X_1^{\mathrm{I}} - X_s^{\mathrm{I}} \right\|^2 \right] \right.
$$
$$
\left. + \mathbb{E}\left[ \left\| \mathbb{E}\left[ \left\langle \frac{X_s^{\mathrm{I}} - X_0^{\mathrm{I}}}{s}, \frac{\nabla_{x_0}\tilde{\pi}}{\tilde{\pi}}(X_0^{\mathrm{I}}, X_1^{\mathrm{I}}) \right\rangle \Big| X_s^{\mathrm{I}} \right] \mathbb{E}\left[ X_1^{\mathrm{I}} - X_s^{\mathrm{I}} \Big| X_s^{\mathrm{I}} \right] \right\|^2 \right] \right\} \rho(s)\mathrm{d}s
$$
$$
+ \int_{1/2}^{1-\epsilon_1} \left\{ \mathbb{E}\left[ \left\| \left\langle X_s^{\mathrm{I}} - X_0^{\mathrm{I}}, \frac{\nabla_{x_0}\tilde{\pi}}{\tilde{\pi}}(X_0^{\mathrm{I}}, X_1^{\mathrm{I}}) \right\rangle \frac{X_1^{\mathrm{I}} - X_s^{\mathrm{I}}}{1-s} \right\|^2 \right] \right.
$$
$$
\left. + \mathbb{E}\left[ \left\| \mathbb{E}\left[ \left\langle X_s^{\mathrm{I}} - X_0^{\mathrm{I}}, \frac{\nabla_{x_0}\tilde{\pi}}{\tilde{\pi}}(X_0^{\mathrm{I}}, X_1^{\mathrm{I}}) \right\rangle \Big| X_s^{\mathrm{I}} \right] \mathbb{E}\left[ \frac{X_1^{\mathrm{I}} - X_s^{\mathrm{I}}}{1-s} \Big| X_s^{\mathrm{I}} \right] \right\|^2 \right] \right\} \rho(s)\mathrm{d}s \,.
$$

We first focus on the time sub-interval $s \in [0, 1/2]$. To this aim, we fix $s \in [0, 1/2]$. Holder and Jensen inequalities yield

$$
\mathbb{E}\left[ \left\| \left\langle \frac{X_s^{\mathrm{I}} - X_0^{\mathrm{I}}}{s}, \frac{\nabla_{x_0}\tilde{\pi}}{\tilde{\pi}}(X_0^{\mathrm{I}}, X_1^{\mathrm{I}}) \right\rangle X_1^{\mathrm{I}} - X_s^{\mathrm{I}} \right\|^2 \right]
$$
$$
+ \mathbb{E}\left[ \left\| \mathbb{E}\left[ \left\langle \frac{X_s^{\mathrm{I}} - X_0^{\mathrm{I}}}{s}, \frac{\nabla_{x_0}\tilde{\pi}}{\tilde{\pi}}(X_0^{\mathrm{I}}, X_1^{\mathrm{I}}) \right\rangle \Big| X_s^{\mathrm{I}} \right] \mathbb{E}\left[ X_1^{\mathrm{I}} - X_s^{\mathrm{I}} \Big| X_s^{\mathrm{I}} \right] \right\|^2 \right]
$$
$$
\leq \mathbb{E}\left[ \left\| \left\langle \frac{X_s^{\mathrm{I}} - X_0^{\mathrm{I}}}{s}, \frac{\nabla_{x_0}\tilde{\pi}}{\tilde{\pi}}(X_0^{\mathrm{I}}, X_1^{\mathrm{I}}) \right\rangle \right\|^4 \right]^{\frac{1}{2}} \mathbb{E}\left[ \left\| X_1^{\mathrm{I}} - X_s^{\mathrm{I}} \right\|^4 \right]^{\frac{1}{2}}
$$
$$
+ \mathbb{E}\left[ \left\| \mathbb{E}\left[ \left\langle \frac{X_s^{\mathrm{I}} - X_0^{\mathrm{I}}}{s}, \frac{\nabla_{x_0}\tilde{\pi}}{\tilde{\pi}}(X_0^{\mathrm{I}}, X_1^{\mathrm{I}}) \right\rangle \Big| X_s^{\mathrm{I}} \right] \right\|^4 \right]^{\frac{1}{2}} \mathbb{E}\left[ \left\| \mathbb{E}\left[ X_1^{\mathrm{I}} - X_s^{\mathrm{I}} \Big| X_s^{\mathrm{I}} \right] \right\|^4 \right]^{\frac{1}{2}}
$$
$$
\lesssim \mathbb{E}\left[ \left\| \left\langle \frac{X_s^{\mathrm{I}} - X_0^{\mathrm{I}}}{s}, \frac{\nabla_{x_0}\tilde{\pi}}{\tilde{\pi}}(X_0^{\mathrm{I}}, X_1^{\mathrm{I}}) \right\rangle \right\|^4 \right]^{\frac{1}{2}} \mathbb{E}\left[ \left\| X_1^{\mathrm{I}} - X_s^{\mathrm{I}} \right\|^4 \right]^{\frac{1}{2}}
$$
$$
\leq \mathbb{E}\left[ \left\| \frac{X_s^{\mathrm{I}} - X_0^{\mathrm{I}}}{s} \right\|^8 \right]^{\frac{1}{4}} \mathbb{E}\left[ \left\| \frac{\nabla_{x_0}\tilde{\pi}}{\tilde{\pi}}(X_0^{\mathrm{I}}, X_1^{\mathrm{I}}) \right\|^8 \right]^{\frac{1}{4}} \mathbb{E}\left[ \left\| X_1^{\mathrm{I}} - X_s^{\mathrm{I}} \right\|^4 \right]^{\frac{1}{2}} \,.
$$

Therefore, we have that

$$
\int_0^{1/2} \int_{\mathbb{R}^d} \left\| A_s^{\mathrm{I}}(x) \right\|^2 \rho(s) p_s^{\mathrm{I}}(x) \mathrm{d}x \mathrm{d}s \lesssim \int_0^{1/2} \mathbb{E}\left[ \left\| \frac{X_s^{\mathrm{I}} - X_0^{\mathrm{I}}}{s} \right\|^8 \right]^{\frac{1}{4}} \mathbb{E}\left[ \left\| \frac{\nabla_{x_0}\tilde{\pi}}{\tilde{\pi}}(X_0^{\mathrm{I}}, X_1^{\mathrm{I}}) \right\|^8 \right]^{\frac{1}{4}} \mathbb{E}\left[ \left\| X_1^{\mathrm{I}} - X_s^{\mathrm{I}} \right\|^4 \right]^{\frac{1}{2}} \rho(s)\mathrm{d}s \,.
$$

Using Lemma 3, Proposition 1, Lemma 4, Equation (26), Lemma 2 and the fact that $\mathbf{m}_8[\mu], \mathbf{m}_8[\nu^\star] \lesssim d^4$, we get that

$$
\int_0^{1/2} \int_{\mathbb{R}^d} \left\| A_s^1(x) \right\|^2 \rho(s) p_s^{\mathrm{I}}(x) \mathrm{d}x \mathrm{d}s
$$

$$
\lesssim \int_0^{1/2} \mathbb{E}\left[ \left\| \frac{\overleftarrow{X}_1 - \overleftarrow{X}_{1-s}}{s} \right\|^8 \right]^{\frac{1}{4}} \mathbb{E}\left[ \left\| \frac{\nabla_{x_0} \tilde{\pi}}{\tilde{\pi}}(X_0^{\mathrm{I}}, X_1^{\mathrm{I}}) \right\|^8 \right]^{\frac{1}{4}} \mathbb{E}\left[ \left\| X_1^{\mathrm{I}} - X_s^{\mathrm{I}} \right\|^4 \right]^{\frac{1}{2}} \rho(s) \mathrm{d}s
$$

$$
\lesssim \|\nabla \log \tilde{\pi}\|_{\mathrm{L}^8(\pi)}^2 (d^2 + \mathbf{m}_4[\mu] + \mathbf{m}_4[\nu^\star])^{\frac{1}{2}} \int_0^{1/2} \mathbb{E}\left[ \left\| \frac{\overleftarrow{f}_{1-s}^s + \overleftarrow{g}_{1-s}^s}{s} \right\|^8 \right]^{\frac{1}{4}} \rho(s) \mathrm{d}s
$$

$$
\lesssim \|\nabla \log \tilde{\pi}\|_{\mathrm{L}^8(\pi)}^2 \left( d + \sqrt{\mathbf{m}_4[\mu]} + \sqrt{\mathbf{m}_4[\nu^\star]} \right) \int_0^{1/2} \left\{ \mathbb{E}\left[ \left\| \frac{\overleftarrow{f}_{1-s}^s}{s} \right\|^8 \right]^{\frac{1}{4}} + \mathbb{E}\left[ \left\| \frac{\overleftarrow{g}_{1-s}^s}{s} \right\|^8 \right]^{\frac{1}{4}} \right\} \rho(s) \mathrm{d}s
$$

$$
\lesssim \|\nabla \log \tilde{\pi}\|_{\mathrm{L}^8(\pi)}^2 \left( d + \sqrt{\mathbf{m}_4[\mu]} + \sqrt{\mathbf{m}_4[\nu^\star]} \right) \left( \|\nabla \log \pi\|_{\mathrm{L}^8(\pi)}^2 + \sqrt[4]{\mathbf{m}_8[\mu]} + \sqrt[4]{\mathbf{m}_8[\nu^\star]} + \int_0^{1/2} \{d^4 s^{4-8}\}^{\frac{1}{4}} s^{\frac{7}{8}} \mathrm{d}s \right)
$$

$$
= \|\nabla \log \tilde{\pi}\|_{\mathrm{L}^8(\pi)}^2 \left( d + \sqrt{\mathbf{m}_4[\mu]} + \sqrt{\mathbf{m}_4[\nu^\star]} \right) \left( \|\nabla \log \pi\|_{\mathrm{L}^8(\pi)}^2 + \sqrt[4]{\mathbf{m}_8[\mu]} + \sqrt[4]{\mathbf{m}_8[\nu^\star]} + d \int_0^{1/2} s^{-\frac{1}{8}} \mathrm{d}s \right)
$$

$$
\lesssim \|\nabla \log \tilde{\pi}\|_{\mathrm{L}^8(\pi)}^2 \left( d + \sqrt{\mathbf{m}_4[\mu]} + \sqrt{\mathbf{m}_4[\nu^\star]} \right) \left( \|\nabla \log \pi\|_{\mathrm{L}^8(\pi)}^2 + \sqrt[4]{\mathbf{m}_8[\mu]} + \sqrt[4]{\mathbf{m}_8[\nu^\star]} + d \right)
$$

$$
\lesssim \left( \|\nabla \log \pi\|_{\mathrm{L}^8(\pi)}^2 + \sqrt[4]{\mathbf{m}_8[\mu]} + \sqrt[4]{\mathbf{m}_8[\nu^\star]} \right) \left( d + \sqrt{\mathbf{m}_4[\mu]} + \sqrt{\mathbf{m}_4[\nu^\star]} \right) \left( \|\nabla \log \pi\|_{\mathrm{L}^8(\pi)}^2 + \sqrt[4]{\mathbf{m}_8[\mu]} + \sqrt[4]{\mathbf{m}_8[\nu^\star]} + d \right)
$$

$$
\lesssim \left( \|\nabla \log \pi\|_{\mathrm{L}^8(\pi)}^2 + d \right) d \left( \|\nabla \log \pi\|_{\mathrm{L}^8(\pi)}^2 + d \right) \lesssim d \left( d^2 + \|\nabla \log \pi\|_{\mathrm{L}^8(\pi)}^4 \right) .
$$

To handle with the time sub-interval $s \in [1/2, 1 - \epsilon_1]$, we proceed in a similar way, but this time using Proposition 2 and lemma 5 rather than Proposition 1 and lemma 4. By doing so we obtain that

$$
\int_{1/2}^{1-\epsilon_1} \left\| A_s^1(x) \right\|^2 \rho(s) p_s^{\mathrm{I}}(x) \mathrm{d}x \mathrm{d}s \lesssim d \left( d^2 + \|\nabla \log \pi\|_{\mathrm{L}^8(\pi)}^4 \right) .
$$

Putting together the two bounds we get

$$
\int_0^{1-\epsilon_1} \int_{\mathbb{R}^d} \left\| A_s^1(x) \right\|^2 \rho(s) p_s^{\mathrm{I}}(x) \mathrm{d}x \mathrm{d}s \lesssim d \left( d^2 + \|\nabla \log \pi\|_{\mathrm{L}^8(\pi)}^4 \right) .
$$

Proceeding in a similar way (we omit the argument, as it is almost a duplication of the previous one), we get

$$
\int_0^{1-\epsilon_1} \int_{\mathbb{R}^d} \left\| A_s^2(x) \right\|^2 \rho(s) p_s^{\mathrm{I}}(x) \mathrm{d}x \, \mathrm{d}s \lesssim d \left( d^2 + \|\nabla \log \pi\|_{\mathrm{L}^8(\pi)}^4 \right) .
$$

We now switch to $A_s^3$. Using (13), integration by parts, we have that

$$\int_0^{1-\epsilon_1} \int_{\mathbb{R}^d} \left\| A_s^3(x) \right\|^2 \rho(s) p_s^{\mathrm{I}}(x) \mathrm{d}x \mathrm{d}s$$

$$\lesssim \int_0^{1-\epsilon_1} \int_{\mathbb{R}^d} \left\| \frac{\int_{\mathbb{R}^{2d}} \nabla_x p_{1-s}(x_1|x)(\nabla_x p_s(x|x_0))^{\mathrm{T}} \tilde{\pi}(x_0, x_1) \mathrm{d}x_0 \mathrm{d}x_1}{p_s^{\mathrm{I}}(x)} \right.$$
$$\left. \cdot \frac{\int_{\mathbb{R}^{2d}} \nabla_x p_s(x|x_0) p_{1-s}(x_1|x) \tilde{\pi}(x_0, x_1) \mathrm{d}x_0 \mathrm{d}x_1}{p_s^{\mathrm{I}}(x)} \right\|^2 \rho(s) p_s^{\mathrm{I}}(x) \mathrm{d}x \mathrm{d}s$$

$$\lesssim \int_0^{1-\epsilon_1} \int_{\mathbb{R}^d} \left\| \frac{\int_{\mathbb{R}^{2d}} \frac{x_1-x}{1-s} \left( \frac{\nabla_{x_0} \tilde{\pi}}{\tilde{\pi}}(x_0, x_1) \right)^{\mathrm{T}} p_{1-s}(x_1|x) p_s(x|x_0) \tilde{\pi}(x_0, x_1) \mathrm{d}x_0 \mathrm{d}x_1}{p_s^{\mathrm{I}}(x)} \right.$$
$$\left. \cdot \frac{\int_{\mathbb{R}^{2d}} \frac{x-x_0}{s} p_s(x|x_0) p_{1-s}(x_1|x) \tilde{\pi}(x_0, x_1) \mathrm{d}x_0 \mathrm{d}x_1}{p_s^{\mathrm{I}}(x)} \right\|^2 \rho(s) p_s^{\mathrm{I}}(x) \mathrm{d}x \mathrm{d}s$$

$$= \int_0^{1-\epsilon_1} \mathbb{E}\left[ \left\| \mathbb{E}\left[ \frac{X_1^{\mathrm{I}} - X_s^{\mathrm{I}}}{1-s} \left( \frac{\nabla_{x_0} \tilde{\pi}}{\tilde{\pi}} \left( X_0^{\mathrm{I}}, X_1^{\mathrm{I}} \right) \right)^{\mathrm{T}} \middle| X_s^{\mathrm{I}} \right] \mathbb{E}\left[ \frac{X_s^{\mathrm{I}} - X_0^{\mathrm{I}}}{s} \middle| X_s^{\mathrm{I}} \right] \right\|^2 \right] \rho(s) \mathrm{d}s$$

Mimicking the argument used to bound $A_s^1$ (we omit the details, as they are almost a duplication of the previous one), we eventually get that

$$\int_0^{1-\epsilon_1} \int_{\mathbb{R}^d} \left\| A_s^3(x) \right\|^2 \rho(s) p_s^{\mathrm{I}}(x) \mathrm{d}x \mathrm{d}s \lesssim d \left( d^2 + \|\nabla \log \pi\|_{\mathrm{L}^8(\pi)}^4 \right).$$

In the very same way, we get

$$\int_0^{1-\epsilon_1} \int_{\mathbb{R}^d} \left\| A_s^4(x) \right\|^2 \rho(s) p_s^{\mathrm{I}}(x) \mathrm{d}x \mathrm{d}s \lesssim d \left( d^2 + \|\nabla \log \pi\|_{\mathrm{L}^8(\pi)}^4 \right).$$

We now turn to $A_s^5$. Using (13) and integration by parts, we obtain

$$\int_0^{1-\epsilon_1} \int_{\mathbb{R}^d} \left\| A_s^5(x) \right\|^2 \rho(s) p_s^{\mathrm{I}}(x) \mathrm{d}x \mathrm{d}s$$

$$\lesssim \int_0^{1-\epsilon_1} \int_{\mathbb{R}^d} \left\| \frac{\left\| \int_{\mathbb{R}^{2d}} \nabla_x p_s(x|x_0) p_{1-s}(x_1|x) \tilde{\pi}(x_0, x_1) \mathrm{d}x_0 \mathrm{d}x_1 \right\|^2}{(p_s^{\mathrm{I}}(x))^2} \right.$$
$$\left. \cdot \frac{\int_{\mathbb{R}^{2d}} p_s(x|x_0) \nabla_x p_{1-s}(x_1|x) \tilde{\pi}(x_0, x_1) \mathrm{d}x_0 \mathrm{d}x_1}{p_s^{\mathrm{I}}(x)} \right\|^2 \rho(s) p_s^{\mathrm{I}}(x) \mathrm{d}x \mathrm{d}s$$

$$= \int_0^{1-\epsilon_1} \int_{\mathbb{R}^d} \left\| \left\langle \frac{\int_{\mathbb{R}^{2d}} \frac{x-x_0}{s} p_s(x|x_0) p_{1-s}(x_1|x) \tilde{\pi}(x_0, x_1) \mathrm{d}x_0 \mathrm{d}x_1}{p_s^{\mathrm{I}}(x)}, \right. \right.$$
$$\left. \left. \frac{\int_{\mathbb{R}^{2d}} \frac{\nabla_{x_0} \tilde{\pi}}{\tilde{\pi}}(x_0, x_1) p_s(x|x_0) p_{1-s}(x_1|x) \tilde{\pi}(x_0, x_1) \mathrm{d}x_0 \mathrm{d}x_1}{p_s^{\mathrm{I}}(x)} \right\rangle \right.$$
$$\left. \cdot \frac{\int_{\mathbb{R}^{2d}} \frac{x_1-x}{1-s} p_s(x|x_0) p_{1-s}(x_1|x) \tilde{\pi}(x_0, x_1) \mathrm{d}x_0 \mathrm{d}x_1}{p_s^{\mathrm{I}}(x)} \right\|^2 \rho(s) p_s^{\mathrm{I}}(x) \mathrm{d}x \mathrm{d}s$$

$$= \int_0^{1-\epsilon_1} \mathbb{E}\left[ \left\| \left\langle \mathbb{E}\left[ \frac{X_s^{\mathrm{I}} - X_0^{\mathrm{I}}}{s} \middle| X_s^{\mathrm{I}} \right], \mathbb{E}\left[ \frac{\nabla_{x_0} \tilde{\pi}}{\tilde{\pi}} \left( X_0^{\mathrm{I}}, X_1^{\mathrm{I}} \right) \middle| X_s^{\mathrm{I}} \right] \right\rangle \mathbb{E}\left[ \frac{X_1^{\mathrm{I}} - X_s^{\mathrm{I}}}{1-s} \middle| X_s^{\mathrm{I}} \right] \right\|^2 \right] \rho(s) \mathrm{d}s.$$

Proceeding as for $A_s^1$, we eventually get

$$\int_0^{1-\epsilon_1} \int_{\mathbb{R}^d} \left\| A_s^5(x) \right\|^2 \rho(s) p_s^{\mathrm{I}}(x) \mathrm{d}x \mathrm{d}s \lesssim d \left( d^2 + \|\nabla \log \pi\|_{\mathrm{L}^8(\pi)}^4 \right) .$$

The argument to bound $A_s^6$ is almost the same (hence omitted) and leads to

$$\int_0^{1-\epsilon_1} \int_{\mathbb{R}^d} \left\| A_s^6(x) \right\|^2 \rho(s) p_s^{\mathrm{I}}(x) \mathrm{d}x \mathrm{d}s \lesssim d \left( d^2 + \|\nabla \log \pi\|_{\mathrm{L}^8(\pi)}^4 \right) .$$

Putting together the bounds on the $\{A_s^k\}_{k=1}^6$ derived so far, we eventually obtain

$$h(h^{1/8}+1) \int_0^{1-\epsilon_1} \mathbb{E}\left[ \left\| (\partial_s + \mathcal{L}_s^{\mathrm{M}}) \tilde{\beta}_s(\overrightarrow{X}_s) \right\|^2 \right] \rho(s) \mathrm{d}s \lesssim h(h^{1/8}+1) d \left( d^2 + \|\nabla \log \pi\|_{\mathrm{L}^8(\pi)}^4 \right) . \tag{33}$$

Plugging (33), (29) and (31) into (28), we get

$$\mathrm{KL}(\nu_{1-\epsilon_1}^\star | \nu_{1-\epsilon_1}^{\theta^\star}) \lesssim \varepsilon^2 + h(h^{1/8}+1) d \left( d^2 + \|\nabla \log \pi\|_{\mathrm{L}^8(\pi)}^4 \right) .$$

The byproduct of the above estimate and (24) leads to

$$\mathrm{KL}(\nu^\star | \nu_1^{\theta^\star}) \lesssim \varepsilon^2 + h(h^{1/8}+1) d \left( d^2 + \|\nabla \log \pi\|_{\mathrm{L}^8(\pi)}^4 \right) .$$

$$\square$$

## B.2. Early-stopping regime with constant step-size

*Proof of Theorem 2:* Fix $0 < \delta < 1/2$. We want to apply Theorem 1 to $\mu$, $\nu_{1-\delta}^\star$ and the coupling $\pi_{1-\delta} \in \Pi(\mu, \nu_{1-\delta}^\star)$ defined as

$$\pi_{1-\delta}(x_0, x_{1-\delta}) = \int_{\mathbb{R}^d} p_{1-\delta|0,1}^{\mathrm{I}}(x_{1-\delta}|x_0, x_1) \pi_{0|1}(x_0|x_1) \nu^\star(\mathrm{d}x_1) ,$$

where $(x_0, x_1, x_{1-\delta}) \mapsto p_{1-\delta|0,1}^{\mathrm{I}}(x_{1-\delta}|x_0, x_1)$ denotes the density of $X_{1-\delta}^{\mathrm{I}}$ given $(X_0^{\mathrm{I}}, X_1^{\mathrm{I}})$ with respect to the Lebesgue measure. To this aim, we need to prove that $\nu_{1-\delta}^\star$ has finite 8th order moment and that $\pi_{1-\delta}$ satisfies

$$\|\nabla \log \pi_{1-\delta}\|_{\mathrm{L}^8(\pi_{1-\delta})} < +\infty .$$

We start with $\nu_{1-\delta}^\star$. Note that, as a very consequence of (9), we have that

$$\mathbf{m}_8[\nu_{1-\delta}^\star] = \mathbb{E}\left[ \left\| X_{1-\delta}^{\mathrm{I}} \right\|^8 \right] \lesssim \delta^8 \mathbf{m}_8[\mu] + (1-\delta)^8 \mathbf{m}_8[\nu^\star] + d^4 \delta^4 (1-\delta)^4 < +\infty . \tag{34}$$

We now switch to $\pi_{1-\delta}$. It follows from the very definition of the stochastic interpolant that,

$$p_{1-\delta|0,1}^{\mathrm{I}}(x_{1-\delta}|x_0, x_1) = \frac{1}{(4\pi\delta(1-\delta))^{d/2}} \exp\left( -\frac{\|x_{1-\delta} - \delta x_0 - (1-\delta)x_1\|^2}{4\delta(1-\delta)} \right) .$$

Therefore,

$$\nabla_{x_0} p_{1-\delta|0,1}^{\mathrm{I}}(x_{1-\delta}|x_0, x_1) = \frac{x_{1-\delta} - \delta x_0 - (1-\delta)x_1}{2(1-\delta)} p_{1-\delta|0,1}^{\mathrm{I}}(x_{1-\delta}|x_0, x_1) ,$$

and

$$\nabla_{x_{1-\delta}} p_{1-\delta|0,1}^{\mathrm{I}}(x_{1-\delta}|x_0, x_1) = -\frac{x_{1-\delta} - \delta x_0 - (1-\delta)x_1}{2\delta(1-\delta)} p_{1-\delta|0,1}^{\mathrm{I}}(x_{1-\delta}|x_0, x_1) .$$

Furthermore,

$$\frac{p_{1-\delta|0,1}^{\mathrm{I}}(x_{1-\delta}|x_0,x_1)\pi_{0|1}(x_0|x_1)\nu^\star(\mathrm{d}x_1)}{\int_{\mathbb{R}^d} p_{1-\delta|0,1}^{\mathrm{I}}(x_{1-\delta}|x_0,\tilde{x}_1)\pi_{0|1}(x_0|\tilde{x}_1)\nu^\star(\mathrm{d}\tilde{x}_1)} = p_{1|0,1-\delta}^{\mathrm{I}}(\mathrm{d}x_1|x_0,x_{1-\delta}) \ .$$

Consequently, we have that

$$\frac{\nabla_{x_0}\pi_{1-\delta}}{\pi_{1-\delta}}(x_0,x_{1-\delta})$$

$$= \frac{\int_{\mathbb{R}^d} p_{1-\delta|0,1}^{\mathrm{I}}(x_{1-\delta}|x_0,x_1)\nabla_{x_0}\pi_{0|1}(x_0|x_1)\nu^\star(\mathrm{d}x_1) + \int_{\mathbb{R}^d}\nabla_{x_0}p_{1-\delta|0,1}^{\mathrm{I}}(x_{1-\delta}|x_0,x_1)\pi_{0|1}(x_0|x_1)\nu^\star(\mathrm{d}x_1)}{\int_{\mathbb{R}^d} p_{1-\delta|0,1}^{\mathrm{I}}(x_{1-\delta}|x_0,\tilde{x}_1)\pi_{0|1}(x_0|\tilde{x}_1)\nu^\star(\mathrm{d}\tilde{x}_1)}$$

$$= \int_{\mathbb{R}^d}\nabla_{x_0}\log\pi_{0|1}(x_0|x_1)p_{1|0,1-\delta}^{\mathrm{I}}(\mathrm{d}x_1|x_0,x_{1-\delta}) + \int_{\mathbb{R}^d}\frac{x_{1-\delta}-\delta x_0-(1-\delta)x_1}{2(1-\delta)}p_{1|0,1-\delta}^{\mathrm{I}}(\mathrm{d}x_1|x_0,x_{1-\delta}) \ ,$$

and that

$$\frac{\nabla_{x_{1-\delta}}\pi_{1-\delta}}{\pi_{1-\delta}}(x_0,x_{1-\delta}) = \frac{\int_{\mathbb{R}^d}\nabla_{x_{1-\delta}}p_{1-\delta|0,1}^{\mathrm{I}}(x_{1-\delta}|x_0,x_1)\pi_{0|1}(x_0|x_1)\nu^\star(\mathrm{d}x_1)}{\int_{\mathbb{R}^d} p_{1-\delta|0,1}^{\mathrm{I}}(x_{1-\delta}|x_0,\tilde{x}_1)\pi_{0|1}(x_0|\tilde{x}_1)\nu^\star(\mathrm{d}\tilde{x}_1)}$$

$$= -\int_{\mathbb{R}^d}\frac{x_{1-\delta}-\delta x_0-(1-\delta)x_1}{2\delta(1-\delta)}p_{1|0,1-\delta}^{\mathrm{I}}(\mathrm{d}x_1|x_0,x_{1-\delta}) \ .$$

But then, if we use Jensen inequality and (37), we get

$$\int_{\mathbb{R}^{2d}}\left\|\nabla_{x_0}\log\frac{\mathrm{d}\pi_{1-\delta}}{\mathrm{dLeb}^{2d}}\right\|^8\mathrm{d}\pi_{1-\delta} \lesssim \mathbb{E}\left[\left\|\mathbb{E}\left[\nabla_{x_0}\log\pi_{0|1}(X_0^{\mathrm{I}}|X_1^{\mathrm{I}})\Big|(X_0^{\mathrm{I}},X_{1-\delta}^{\mathrm{I}})\right]\right\|^8\right]$$

$$+ \mathbb{E}\left[\left\|\mathbb{E}\left[\frac{X_{1-\delta}^{\mathrm{I}}-\delta X_0^{\mathrm{I}}-(1-\delta)X_1^{\mathrm{I}}}{1-\delta}\Big|(X_0^{\mathrm{I}},X_{1-\delta}^{\mathrm{I}})\right]\right\|^8\right]$$

$$\lesssim \left\|\nabla\log\pi_{0|1}\right\|_{\mathrm{L}^8(\pi_{0|1})}^8 + \mathbb{E}\left[\left\|\frac{X_{1-\delta}^{\mathrm{I}}-\delta X_0^{\mathrm{I}}-(1-\delta)X_1^{\mathrm{I}}}{1-\delta}\right\|^8\right]$$

$$\lesssim \left\|\nabla\log\pi_{0|1}\right\|_{\mathrm{L}^8(\pi_{0|1})}^8 + \mathbf{m}_8[\nu_{1-\delta}^\star]\frac{1}{(1-\delta)^8} + \mathbf{m}_8[\mu]\frac{\delta^8}{(1-\delta)^8} + \mathbf{m}_8[\nu^\star]$$

$$\lesssim \left\|\nabla\log\pi_{0|1}\right\|_{\mathrm{L}^8(\pi_{0|1})}^8 + \mathbf{m}_8[\mu]\frac{\delta^8}{(1-\delta)^8} + \mathbf{m}_8[\nu^\star] + d^4\frac{\delta^4}{(1-\delta)^4}$$

$$\lesssim \left\|\nabla\log\pi_{0|1}\right\|_{\mathrm{L}^8(\pi_{0|1})}^8 + \mathbf{m}_8[\mu]\frac{1}{(1-\delta)^8} + \mathbf{m}_8[\nu^\star]\frac{1}{\delta^8} + d^4\frac{1}{\delta^4(1-\delta)^4} \ .$$

and (similarly)

$$\int_{\mathbb{R}^{2d}}\left\|\nabla_{x_{1-\delta}}\log\frac{\mathrm{d}\pi_{1-\delta}}{\mathrm{dLeb}^{2d}}\right\|^8\mathrm{d}\pi_{1-\delta} \lesssim \mathbb{E}\left[\left\|\mathbb{E}\left[\frac{X_{1-\delta}^{\mathrm{I}}-\delta X_0^{\mathrm{I}}-(1-\delta)X_1^{\mathrm{I}}}{\delta(1-\delta)}\Big|(X_0^{\mathrm{I}},X_{1-\delta}^{\mathrm{I}})\right]\right\|^8\right]$$

$$\lesssim \mathbf{m}_8[\mu]\frac{1}{(1-\delta)^8} + \mathbf{m}_8[\nu^\star]\frac{1}{\delta^8} + d^4\frac{1}{\delta^4(1-\delta)^4} \ .$$

Plugging these estimates in the convergence bound provided in Theorem 1 and using the fact that $\mathbf{m}_8[\mu],\mathbf{m}_8[\nu^\star]\lesssim d^4$ allows to conclude. $\qquad\square$

*Proof of Corollary 1:* It follows from the very definition of Fortet- Mourier distance, triangle inequality and Pinsker's inequality that

$$\mathscr{W}_{2,\mathrm{FM}}^2(\nu^\star,\nu_{1-\delta}^{\theta^\star}) \lesssim \mathscr{W}_2^2(\nu^\star,\nu_{1-\delta}^\star) + \mathrm{TV}^2(\nu_{1-\delta}^\star,\nu_{1-\delta}^{\theta^\star}) \lesssim \mathscr{W}_2^2(\nu^\star,\nu_{1-\delta}^\star) + \mathrm{KL}(\nu_{1-\delta}^\star|\nu_{1-\delta}^{\theta^\star}) \ . \tag{35}$$

with TV denoting the total variation distance. It follows from (9), that

$$\mathscr{W}_2^2(\nu^\star, \nu_{1-\delta}^\star) \lesssim \delta^2 \mathbf{m}_2[\mu] + \delta^2 \mathbf{m}_2[\nu^\star] + \delta(1-\delta)d \ .$$

Plugging this and the bound provided in Theorem 2 into (35) leads to

$$\mathscr{W}_{2,\mathrm{FM}}^2(\nu^\star, \nu_{1-\delta}^{\theta^\star}) \lesssim \delta^2 \mathbf{m}_2[\mu] + \delta^2 \mathbf{m}_2[\nu^\star] + \delta d + \varepsilon^2 + h(h+1)\left(\frac{d^2}{\delta^4} + \left\|\nabla \log \pi_{0|1}\right\|_{\mathrm{L}^8(\pi_{0|1})}^4\right)d \ .$$

Therefore, if $\delta = \mathcal{O}(\varepsilon^2/d)$, when choosing $h = \mathcal{O}(\varepsilon^{10}/d^7)$, we have that

$$\mathscr{W}_{2,\mathrm{FM}}^2(\nu^\star, \nu_{1-\delta}^{\theta^\star}) \lesssim \mathcal{O}(\varepsilon^2) \ ,$$

which proves the desired bound.

$\square$

## B.3. Early-stopping regime with novel step-size schedule

*Proof of Theorem 3:* Our aim is to apply Theorem 1 on the sub-partition $\{t_k\}_{k=0}^{M_h}$ of $[0, 1/2]$ with constant step size $h_k = h$ for $k \leq M_h$, and Proposition 5.1 in (Liu et al., 2025) on the sub-partition $\{t_k\}_{k=M_h}^{M_h+N}$ of $[1/2, 1]$ with exponentially decreasing step sizes $h_k = h \min\{t_k, 1-t_k\}$ for $M_h < k \leq M_h + N$. Since the hypothesis of Proposition 5.1 in (Liu et al., 2025) are satisfied in our setting, we immediately get that

$$\mathrm{KL}(\nu^\star|\nu_1^{\theta^\star}) \lesssim \mathrm{KL}(\nu_{1/2}^\star|\nu_{1/2}^{\theta^\star}) + \varepsilon^2 + h^2 d^3 + h^2 d^3 \log\frac{1}{\delta} + hd^2 + hd^2 \log\frac{1}{\delta} \quad (36)$$

$$= \mathrm{KL}(\nu_{1/2}^\star|\nu_{1/2}^{\theta^\star}) + \varepsilon^2 + h^2 d^3\left(1 + \log\frac{1}{\delta}\right) + hd^2\left(1 + \log\frac{1}{\delta}\right)$$

$$= \mathrm{KL}(\nu_{1/2}^\star|\nu_{1/2}^{\theta^\star}) + \varepsilon^2 + \left(1 + \log\frac{1}{\delta}\right)\left(h^2 d^3 + hd^2\right)$$

$$\lesssim \mathrm{KL}(\nu_{1/2}^\star|\nu_{1/2}^{\theta^\star}) + \varepsilon^2 + hd^3\left(1 + \log\frac{1}{\delta}\right)$$

$$\lesssim \mathrm{KL}(\nu_{1/2}^\star|\nu_{1/2}^{\theta^\star}) + \varepsilon^2 + hd^3 \log\frac{1}{\delta} \ ,$$

with $\nu_{1/2}^{\theta^\star}$ denoting the law of $X_{1/2}^{\theta^\star}$. In order to apply Theorem 1, we first need to prove that the probability distribution $\nu_{1/2}^\star = \mathcal{L}\left(X_{1/2}^{\mathrm{I}}\right)$ and the coupling

$$\pi_{0,1/2}(x_0, x_{1/2}) = \int_{\mathbb{R}^d} p_{1/2|0,1}^{\mathrm{I}}(x_{1/2}|x_0, x_1)\pi_{0|1}(x_0|x_1)\nu^\star(\mathrm{d}x_1) \in \Pi(\mu, \nu_{1/2}^\star) \ ,$$

with $(x_0, x_1, x_{1/2}) \mapsto p_{1/2|0,1}^{\mathrm{I}}(x_{1/2}|x_0, x_1)$ denoting the density of $X_{1/2}^{\mathrm{I}}$ given $(X_0^{\mathrm{I}}, X_1^{\mathrm{I}})$ with respect to the Lebesgue measure, satisfy $\mathbf{m}_8[\nu^\star] < +\infty$, and

$$\|\nabla \log \pi_{0,1/2}\|_{\mathrm{L}^8(\pi_{0,1/2})} < +\infty \ ,$$

respectively. On the one side, as a very consequence of (9), we have that

$$\mathbf{m}_8[\nu_{1/2}^\star] = \mathbb{E}\left[\left\|X_{1/2}^{\mathrm{I}}\right\|^8\right] \lesssim \mathbf{m}_8[\mu] + \mathbf{m}_8[\nu^\star] + d^4 < +\infty \ . \quad (37)$$

On the other side, being

$$\nabla_{x_0}\pi_{0,1/2}(x_0, x_{1/2})$$

$$= \int_{\mathbb{R}^d} \nabla_{x_0}p_{1/2|0,1}^{\mathrm{I}}(x_{1/2}|x_0, x_1)\pi_{0|1}(x_0|x_1)\nu^\star(\mathrm{d}x_1) + \int_{\mathbb{R}^d} p_{1/2|0,1}^{\mathrm{I}}(x_{1/2}|x_0, x_1)\nabla_{x_0}\pi_{0|1}(x_0|x_1)\nu^\star(\mathrm{d}x_1) \ ,$$

we have that

$$
\nabla_{x_0} \log \pi_{0,1/2}(x_0, x_{1/2})
$$
$$
= \frac{\int_{\mathbb{R}^d} \nabla_{x_0} \log p^{\mathrm{I}}_{1/2|0,1}(x_{1/2}|x_0, x_1) p^{\mathrm{I}}_{1/2|0,1}(x_{1/2}|x_0, x_1) \pi_{0|1}(x_0|x_1) \nu^\star(\mathrm{d}x_1)}{\int_{\mathbb{R}^d} p^{\mathrm{I}}_{1/2|0,1}(x_{1/2}|x_0, \tilde{x}_1) \pi_{0|1}(x_0|\tilde{x}_1) \nu^\star(\mathrm{d}\tilde{x}_1)}
$$
$$
+ \frac{\int_{\mathbb{R}^d} \nabla_{x_0} \log \pi_{0|1}(x_0|x_1) p^{\mathrm{I}}_{1/2|0,1}(x_{1/2}|x_0, x_1) \pi_{0|1}(x_0|x_1) \nu^\star(\mathrm{d}x_1)}{\int_{\mathbb{R}^d} p^{\mathrm{I}}_{1/2|0,1}(x_{1/2}|x_0, \tilde{x}_1) \pi_{0|1}(x_0|\tilde{x}_1) \nu^\star(\mathrm{d}\tilde{x}_1)} \ .
$$

Since

$$
\frac{p^{\mathrm{I}}_{1/2|0,1}(x_{1/2}|x_0, x_1) \pi_{0|1}(x_0|x_1) \nu^\star(\mathrm{d}x_1)}{\int_{\mathbb{R}^d} p^{\mathrm{I}}_{1/2|0,1}(x_{1/2}|x_0, \tilde{x}_1) \pi_{0|1}(x_0|\tilde{x}_1) \nu^\star(\mathrm{d}\tilde{x}_1)} = p^{\mathrm{I}}_{1|0,1/2}(\mathrm{d}x_1|x_0, x_{1/2}) \ ,
$$

we can rewrite $\nabla_{x_0} \log \pi_{0,1/2}$ as

$$
\nabla_{x_0} \log \pi_{0,1/2}(x_0, x_{1/2}) = \int_{\mathbb{R}^d} \nabla_{x_0} \log p^{\mathrm{I}}_{1/2|0,1}(x_{1/2}|x_0, x_1) p^{\mathrm{I}}_{1|0,1/2}(\mathrm{d}x_1|x_0, x_{1/2}) \tag{38}
$$
$$
+ \int_{\mathbb{R}^d} \nabla_{x_0} \log \pi_{0|1}(x_0|x_1) p^{\mathrm{I}}_{1|0,1/2}(\mathrm{d}x_1|x_0, x_{1/2}) \ .
$$

It follows from the very definition of the stochastic interpolant that,

$$
p^{\mathrm{I}}_{1/2|0,1}(x_{1/2}|x_0, x_1) = \frac{1}{\pi^{d/2}} \exp\left( -\left\| x_{1/2} - \frac{1}{2} x_0 - \frac{1}{2} x_1 \right\|^2 \right) . \tag{39}
$$

Hence

$$
\nabla_{x_0} \log p^{\mathrm{I}}_{1/2|0,1}(x_{1/2}|x_0, x_1) = -\frac{1}{2} x_0 - \frac{1}{2} x_1 + x_{1/2} \ .
$$

Plugging this equality in (38), we get

$$
\nabla_{x_0} \log \pi_{0,1/2}(x_0, x_{1/2}) = \int_{\mathbb{R}^d} \left( -\frac{1}{2} x_0 - \frac{1}{2} x_1 + x_{1/2} \right) p^{\mathrm{I}}_{1|0,1/2}(\mathrm{d}x_1|x_0, x_{1/2})
$$
$$
+ \int_{\mathbb{R}^d} \nabla_{x_0} \log \pi_{0|1}(x_0|x_1) p^{\mathrm{I}}_{1|0,1/2}(\mathrm{d}x_1|x_0, x_{1/2}) \ .
$$

Therefore, using Jensen inequality we get that

$$
\|\nabla_{x_0} \log \pi_{0,1/2}\|^8_{\mathrm{L}^8(\pi_{0,1/2})}
$$
$$
\lesssim \mathbb{E}\left[ \left\| \mathbb{E}\left[ -\frac{1}{2} X^{\mathrm{I}}_0 - \frac{1}{2} X^{\mathrm{I}}_1 + X^{\mathrm{I}}_{1/2} \middle| (X^{\mathrm{I}}_0, X^{\mathrm{I}}_{1/2}) \right] \right\|^8 \right] + \mathbb{E}\left[ \left\| \mathbb{E}\left[ \nabla_{x_0} \log \pi_{0|1}(X^{\mathrm{I}}_0|X^{\mathrm{I}}_1) \middle| (X^{\mathrm{I}}_0, X^{\mathrm{I}}_{1/2}) \right] \right\|^8 \right]
$$
$$
\lesssim \mathbb{E}\left[ \left\| -\frac{1}{2} X^{\mathrm{I}}_0 - \frac{1}{2} X^{\mathrm{I}}_1 + X^{\mathrm{I}}_{1/2} \right\|^8 \right] + \mathbb{E}\left[ \left\| \nabla_{x_0} \log \pi_{0|1}(X^{\mathrm{I}}_0|X^{\mathrm{I}}_1) \right\|^8 \right] \ .
$$

But then, leveraging (37), we obtain

$$
\|\nabla_{x_0} \log \pi_{0,1/2}\|_{\mathrm{L}^8(\pi_{0,1/2})} \lesssim \sqrt[8]{\mathbf{m}_8[\mu]} + \sqrt[8]{\mathbf{m}_8[\nu^\star]} + \|\nabla \log \pi_{0|1}\|_{\mathrm{L}^8(\pi_{0|1})} \lesssim \sqrt{d} + \|\nabla \log \pi_{0|1}\|_{\mathrm{L}^8(\pi_{0|1})} \ .
$$

Similarly, being

$$
\nabla_{x_{1/2}} \pi_{0,1/2}(x_0, x_{1/2}) = \int_{\mathbb{R}^d} \nabla_{x_{1/2}} p^{\mathrm{I}}_{1/2|0,1}(x_{1/2}|x_0, x_1) \pi_{0|1}(x_0|x_1) \nu^\star(\mathrm{d}x_1) \ ,
$$

we have that

$$
\begin{aligned}
&\nabla_{x_{1/2}} \log \pi_{0,1/2}(x_0, x_{1/2}) \\
&= \frac{\int_{\mathbb{R}^d} \nabla_{x_{1/2}} \log p_{1/2|0,1}^{\mathrm{I}}(x_{1/2}|x_0, x_1) p_{1/2|0,1}^{\mathrm{I}}(x_{1/2}|x_0, x_1) \pi_{0|1}(x_0|x_1) \nu^\star(\mathrm{d}x_1)}{\int_{\mathbb{R}^d} \nabla_{x_{1/2}} p_{1/2|0,1}^{\mathrm{I}}(x_{1/2}|x_0, \tilde{x}_1) \pi_{0|1}(x_0|\tilde{x}_1) \nu^\star(\mathrm{d}\tilde{x}_1)} \\
&= \int_{\mathbb{R}^d} \nabla_{x_{1/2}} \log p_{1/2|0,1}^{\mathrm{I}}(x_{1/2}|x_0, x_1) p_{1|0,1/2}^{\mathrm{I}}(\mathrm{d}x_1|x_0, x_{1/2}) .
\end{aligned}
$$

It follows from (39) that

$$
\nabla_{x_{1/2}} \log p_{1/2|0,1}^{\mathrm{I}}(x_{1/2}|x_0, x_1) = -2x_{1/2} + x_0 + x_1 .
$$

At this point, proceeding as before, we get

$$
\|\nabla_{x_{1/2}} \log \pi_{0,1/2}\|_{\mathrm{L}^8(\pi_{0,1/2})} \lesssim \sqrt[8]{\mathbf{m}_8[\mu]} + \sqrt[8]{\mathbf{m}_8[\nu^\star]} \lesssim \sqrt{d} .
$$

Therefore, we have that

$$
\|\nabla \log \pi_{0,1/2}\|_{\mathrm{L}^8(\pi_{0,1/2})} \lesssim \sqrt{d} + \|\nabla \log \pi_{0|1}\|_{\mathrm{L}^8(\pi_{0|1})} .
$$

Recalling Remark 1 and applying Theorem 1 on the time interval $[0, 1/2]$, with prior $\mu$, target $\nu_{1/2}^\star$ and coupling $\pi_{0,1/2}$, we get

$$
\mathrm{KL}(\nu_{1/2}^\star | \nu_{1/2}^{\theta^\star}) \lesssim \varepsilon^2 + h(h^{1/8} + 1)\Big(d^2 + \|\nabla \log \pi_{0|1}\|_{\mathrm{L}^8(\pi_{0|1})}^4\Big)d .
$$

Combining the above bound with (36) yields

$$
\mathrm{KL}(\nu^\star | \nu_1^{\theta^\star}) \lesssim \varepsilon^2 + h(h^{1/8} + 1)\Big(d^2 + \|\nabla \log \pi_{0|1}\|_{\mathrm{L}^8(\pi_{0|1})}^4\Big)d + hd^3 \log \frac{1}{\delta} .
$$

$\square$

*Proof of Corollary 2:* It follows from the very definition of Fortet-Mourier distance, triangle inequality and Pinsker's inequality that

$$
\mathscr{W}_{2,\mathrm{FM}}^2(\nu^\star, \nu_{1-\delta}^{\theta^\star}) \lesssim \mathscr{W}_2^2(\nu^\star, \nu_{1-\delta}^\star) + \mathrm{TV}^2(\nu_{1-\delta}^\star, \nu_{1-\delta}^{\theta^\star}) \lesssim \mathscr{W}_2^2(\nu^\star, \nu_{1-\delta}^\star) + \mathrm{KL}(\nu_{1-\delta}^\star | \nu_{1-\delta}^{\theta^\star}) . \tag{40}
$$

with TV denoting the total variation distance. It follows from (9), that

$$
\mathscr{W}_2^2(\nu^\star, \nu_{1-\delta}^\star) \lesssim \delta^2 \mathbf{m}_2[\mu] + \delta^2 \mathbf{m}_2[\nu^\star] + \delta(1-\delta)d .
$$

Plugging this and the bound provided in Theorem 3 into (40) leads to

$$
\mathscr{W}_{2,\mathrm{FM}}^2(\nu^\star, \nu_{1-\delta}^{\theta^\star}) \lesssim \delta^2 \mathbf{m}_2[\mu] + \delta^2 \mathbf{m}_2[\nu^\star] + \delta d + \varepsilon^2 + hd^3 \log \frac{1}{\delta} + h(h^{1/8} + 1)\Big(d^2 + \|\nabla \log \pi_{0|1}\|_{\mathrm{L}^8(\pi_{0|1})}^4\Big)d .
$$

Therefore, if $\delta = \mathcal{O}(\varepsilon^2/d)$, when choosing $h = \tilde{\mathcal{O}}(\varepsilon^2/d^3)$, we have that

$$
\mathscr{W}_{2,\mathrm{FM}}^2(\nu^\star, \nu_{1-\delta}^{\theta^\star}) \lesssim \mathcal{O}(\varepsilon^2) ,
$$

which proves the desired bound. $\square$

## C. Convergence Bounds in Wasserstein-2 Distance

### C.1. Strong Log-Concave and Full Log-Lipschitz Distributions

For sake of clarity, we first prove (8) under **H**9, *i.e.*, we prove the following result.

**Theorem 5.** *Let $\{t_k\}_{k=0}^{N_h}$ be a uniform partition of $[0,1]$ with step size $h = 1/N_h > 0$. Under **H**2, 3, 5, 7 and 9, denoting by $\nu_1^{\theta^\star}$ the law of $X_1^{\theta^\star}$, we have that*

$$\mathscr{W}_2(\nu^\star, \nu_1^{\theta^\star}) \lesssim \exp\left(\frac{8\sqrt{2}\|\nabla^2 \log \pi\|_{\mathrm{L}^2(\pi)}}{\sqrt{\alpha_\pi}}\right)\left(\varepsilon + \sqrt{h}(h^{1/16} + 1)\sqrt{\left(d^2 + \|\nabla \log \pi\|_{\mathrm{L}^8(\pi)}^4\right)d}\right). \tag{41}$$

*Proof of Theorem 5:* Consider the synchronous coupling between $(X_t^{\mathrm{M}})_{t \in [0,1]}$ and the continuous time interpolation of $(X_t^{\theta^\star})_{t \in [0,1]}$ with the same initialization, *i.e.*, use the same Brownian motion to drive the two processes and set $X_0^{\theta^\star} = X_0^{\mathrm{M}}$. Then, it holds

$$\mathscr{W}_2(\nu^\star, \nu^{\theta^\star}) \le \left\|X_T^{\mathrm{M}} - X_T^{\theta^\star}\right\|_{\mathrm{L}^2} = \left\|X_{t_N}^{\mathrm{M}} - X_{t_N}^{\theta^\star}\right\|_{\mathrm{L}^2}, \tag{42}$$

where, with abuse of notation, we denoted by $(X_t^{\theta^\star})_{t \in [0,1]}$ either the process (6) and its time continuous interpolation. To upper bound the r.h.s. of the above expression, we estimate $\left\|X_{t_{k+1}}^{\mathrm{M}} - X_{t_{k+1}}^{\theta^\star}\right\|_{\mathrm{L}^2}$ by means of $\left\|X_{t_k}^{\mathrm{M}} - X_{t_k}^{\theta^\star}\right\|_{\mathrm{L}^2}$ and develop the recursion. Fix $k \in \{0, ..., N-1\}$. As we considered the synchronous coupling, we have that

$$
\begin{aligned}
&\left\|X_{t_{k+1}}^{\mathrm{M}} - X_{t_{k+1}}^{\theta^\star}\right\|_{\mathrm{L}^2} \qquad\qquad\qquad\qquad\qquad\qquad\qquad\qquad\qquad\qquad\qquad\qquad\qquad\qquad (43)\\
&= \left\|X_{t_k}^{\mathrm{M}} - X_{t_k}^{\theta^\star} + \int_{t_k}^{t_{k+1}} \left\{\tilde{\beta}_t(X_t^{\mathrm{M}}) - s_{\theta^\star}(t_k, X_{t_k}^{\theta^\star})\right\} \mathrm{d}t\right\|_{\mathrm{L}^2}\\
&\le \left\|X_{t_k}^{\mathrm{M}} - X_{t_k}^{\theta^\star}\right\|_{\mathrm{L}^2} + \sqrt{h}\left(\int_{t_k}^{t_{k+1}} \mathbb{E}\left[\left\|\left\{\tilde{\beta}_t(X_t^{\mathrm{M}}) - \tilde{\beta}_{t_k}(X_{t_k}^{\mathrm{M}})\right\}\right\|^2\right]\mathrm{d}t\right)^{1/2} + h\left\|\tilde{\beta}_{t_k}(X_{t_k}^{\mathrm{M}}) - \tilde{\beta}_{t_k}(X_{t_k}^{\theta^\star})\right\|_{\mathrm{L}^2}\\
&\quad + h\left\|\tilde{\beta}_{t_k}(X_{t_k}^{\theta^\star}) - s_{\theta^\star}(t_k, X_{t_k}^{\theta^\star})\right\|_{\mathrm{L}^2}\\
&\le \left\|X_{t_k}^{\mathrm{M}} - X_{t_k}^{\theta^\star}\right\|_{\mathrm{L}^2} + h\left\|\tilde{\beta}_{t_k}(X_{t_k}^{\mathrm{M}}) - \tilde{\beta}_{t_k}(X_{t_k}^{\theta^\star})\right\|_{\mathrm{L}^2} + h\varepsilon\\
&\quad + h\left(\sum_{k=0}^{N-1}\int_{t_k}^{t_{k+1}} \mathbb{E}\left[\left\|\left\{\tilde{\beta}_t(X_t^{\mathrm{M}}) - \tilde{\beta}_{t_k}(X_{t_k}^{\mathrm{M}})\right\}\right\|^2\right]\mathrm{d}t\right)^{1/2},
\end{aligned}
$$

where, in the last inequality, we have used **H**1. We now focus on the second term appearing in the r.h.s. of the above expression. First note that if $k = 0$ then this term is null. Therefore, we can assume $0 < k < N$, so that $0 < t_k < 1$, use **H**9 and apply Proposition 3. By doing so, we get

$$h\left\|\tilde{\beta}_{t_k}(X_{t_k}^{\mathrm{M}}) - \tilde{\beta}_{t_k}(X_{t_k}^{\theta^\star})\right\|_{\mathrm{L}^2} \le h\frac{4\sqrt{2}}{\sqrt{\alpha_\pi}}\|\nabla^2 \log \pi\|_{\mathrm{L}^2(\pi)}\frac{1}{\sqrt{t_k(1-t_k)}}\left\|X_{t_k}^{\mathrm{M}} - X_{t_k}^{\theta^\star}\right\|_{\mathrm{L}^2}.$$

Plugging this into (43), we get

$$
\begin{aligned}
\left\|X_{t_{k+1}}^{\mathrm{M}} - X_{t_{k+1}}^{\theta^\star}\right\|_{\mathrm{L}^2} &\le \left\|X_{t_k}^{\mathrm{M}} - X_{t_k}^{\theta^\star}\right\|_{\mathrm{L}^2} + h\frac{4\sqrt{2}}{\sqrt{\alpha_\pi}}\|\nabla^2 \log \pi\|_{\mathrm{L}^2(\pi)}\frac{1}{\sqrt{t_k(1-t_k)}}\left\|X_{t_k}^{\mathrm{M}} - X_{t_k}^{\theta^\star}\right\|_{\mathrm{L}^2} \qquad (44)\\
&\quad + h\varepsilon + h\left(\sum_{k=0}^{N-1}\int_{t_k}^{t_{k+1}} \mathbb{E}\left[\left\|\left\{\tilde{\beta}_t(X_t^{\mathrm{M}}) - \tilde{\beta}_{t_k}(X_{t_k}^{\mathrm{M}})\right\}\right\|^2\right]\mathrm{d}t\right)^{1/2}.
\end{aligned}
$$

We are left with bounding the last term appearing in the r.h.s.. To this aim, we proceed as in the proof of Theorem 1 and

obtain

$$h \left( \sum_{k=0}^{N-1} \int_{t_k}^{t_{k+1}} \mathbb{E} \left[ \left\| \left\{ \tilde{\beta}_t(X_t^{\mathrm{M}}) - \tilde{\beta}_{t_k}(X_{t_k}^{\mathrm{M}}) \right\} \right\|^2 \right] \mathrm{d}t \right)^{1/2}$$

$$\leq \mathbf{c} h \sqrt{h(h^{1/8} + 1) \left( d^2 + \|\nabla \log \pi\|_{\mathrm{L}^8(\pi)}^4 \right) d}$$

$$\leq \mathbf{c} h \sqrt{h} (h^{1/16} + 1) \sqrt{\left( d^2 + \|\nabla \log \pi\|_{\mathrm{L}^8(\pi)}^4 \right) d} \,,$$

for some universal constant $\mathbf{c} > 0$, which may change from line to line. Plugging this bound in (44), we get

$$\left\| X_{t_{k+1}}^{\mathrm{M}} - X_{t_{k+1}}^{\theta^\star} \right\|_{\mathrm{L}^2} \leq \gamma_k \left\| X_{t_k}^{\mathrm{M}} - X_{t_k}^{\theta^\star} \right\|_{\mathrm{L}^2} + h\sqrt{h}\mathrm{D} + h\varepsilon \,, \quad k \in \{0, ..., N-2\} \,,$$

with

$$\gamma_k = 1 + h \frac{4\sqrt{2}}{\sqrt{\alpha_\pi}} \|\nabla^2 \log \pi\|_{\mathrm{L}^2(\pi)} \frac{1}{\sqrt{t_k(1 - t_k)}} \,,$$

and

$$\mathrm{D} = \mathbf{c}(h^{1/16} + 1) \sqrt{\left( d^2 + \|\nabla \log \pi\|_{\mathrm{L}^8(\pi)}^4 \right) d} \,. \tag{45}$$

Therefore, if we develop the recursion and use the fact that we set $X_0^{\mathrm{M}} = X_0^{\theta^\star}$, we get that

$$\left\| X_T^{\mathrm{M}} - X_T^{\theta^\star} \right\|_{\mathrm{L}^2} \leq \left\| X_0^{\mathrm{M}} - X_0^{\theta^\star} \right\|_{\mathrm{L}^2} \prod_{l=0}^{N-2} \gamma_l + (h\sqrt{h}\mathrm{D} + h\varepsilon) \sum_{k=0}^{N-1} \prod_{l=k}^{N-2} \gamma_l$$

$$= (\sqrt{h}\mathrm{D} + \varepsilon) \left( h \sum_{k=0}^{N-1} \prod_{l=k}^{N-1} \gamma_l \right) \,.$$

Note that

$$h \sum_{k=0}^{N-1} \prod_{l=k}^{N-1} \gamma_l = h \sum_{k=0}^{\lfloor N/2 \rfloor} \prod_{l=k}^{\lfloor N/2 \rfloor} \gamma_l + h \sum_{k=\lfloor N/2 \rfloor}^{N-1} \prod_{l=k}^{N-1} \gamma_l$$

$$\leq h \sum_{k=0}^{\lfloor N/2 \rfloor} \prod_{l=k}^{\lfloor N/2 \rfloor} \left( 1 + \sqrt{h} \frac{4\sqrt{2} \, \|\nabla^2 \log \pi\|_{\mathrm{L}^2(\pi)}}{\sqrt{\alpha_\pi}} \frac{1}{\sqrt{l}} \right) + h \sum_{k=\lfloor N/2 \rfloor}^{N-1} \prod_{l=k}^{N-1} \left( 1 + \sqrt{h} \frac{4\sqrt{2} \, \|\nabla^2 \log \pi\|_{\mathrm{L}^2(\pi)}}{\sqrt{\alpha_\pi}} \frac{1}{\sqrt{l}} \right)$$

$$= h \sum_{k=0}^{N-1} \prod_{l=k}^{N-1} \left( 1 + \sqrt{h} \frac{4\sqrt{2} \, \|\nabla^2 \log \pi\|_{\mathrm{L}^2(\pi)}}{\sqrt{\alpha_\pi}} \frac{1}{\sqrt{l}} \right) \leq h \sum_{k=0}^{N-1} \prod_{l=k}^{N-1} \exp\left( \sqrt{h} \frac{4\sqrt{2} \, \|\nabla^2 \log \pi\|_{\mathrm{L}^2(\pi)}}{\sqrt{\alpha_\pi}} \frac{1}{\sqrt{l}} \right)$$

$$= h \sum_{k=0}^{N-1} \exp\left( \sqrt{h} \frac{4\sqrt{2} \, \|\nabla^2 \log \pi\|_{\mathrm{L}^2(\pi)}}{\sqrt{\alpha_\pi}} \sum_{l=k}^{N-1} \frac{1}{\sqrt{l}} \right) \leq h \sum_{k=0}^{N-1} \exp\left( \sqrt{hN} \frac{8\sqrt{2} \, \|\nabla^2 \log \pi\|_{\mathrm{L}^2(\pi)}}{\sqrt{\alpha_\pi}} \right)$$

$$\leq Nh \exp\left( \frac{8\sqrt{2} \, \|\nabla^2 \log \pi\|_{\mathrm{L}^2(\pi)}}{\sqrt{\alpha_\pi}} \right) = \exp\left( \frac{8\sqrt{2} \, \|\nabla^2 \log \pi\|_{\mathrm{L}^2(\pi)}}{\sqrt{\alpha_\pi}} \right) \,.$$

Therefore, we have that

$$\left\| X_T^{\mathrm{M}} - X_T^{\theta^\star} \right\|_{\mathrm{L}^2} \leq \exp\left( \frac{8\sqrt{2}}{\sqrt{\alpha_\pi}} \|\nabla^2 \log \pi\|_{\mathrm{L}^2(\pi)} \right) (\sqrt{h}\mathrm{D} + \varepsilon) \,.$$

Recalling the definition (45) of D, and plugging this bound in (42), we obtain (41). $\square$

## C.2. Weakly Log-Concave and One-Sided Log-Lipschitz Distributions

*Proof of Theorem 4:* We proceed as in the proof of Theorem 5, to obtain (43), *i.e.*,

$$
\left\| X_{t_{k+1}}^{\mathrm{M}} - X_{t_{k+1}}^{\theta^\star} \right\|_{\mathrm{L}^2} \leq \left\| X_{t_k}^{\mathrm{M}} - X_{t_k}^{\theta^\star} \right\|_{\mathrm{L}^2} + h \left\| \tilde{\beta}_{t_k}(X_{t_k}^{\mathrm{M}}) - \tilde{\beta}_{t_k}(X_{t_k}^{\theta^\star}) \right\|_{\mathrm{L}^2} + h\varepsilon
$$
$$
+ h \left( \sum_{k=0}^{N-1} \int_{t_k}^{t_{k+1}} \mathbb{E}\left[ \left\| \left\{ \tilde{\beta}_t(X_t^{\mathrm{M}}) - \tilde{\beta}_{t_k}(X_{t_k}^{\mathrm{M}}) \right\} \right\|^2 \right] \mathrm{d}t \right)^{1/2} .
$$

To bound the second term of the r.h.s. of the above expression, we use Proposition 4 instead of Proposition 3. By doing so, we get

$$
h \left\| \tilde{\beta}_{t_k}(X_{t_k}^{\mathrm{M}}) - \tilde{\beta}_{t_k}(X_{t_k}^{\theta^\star}) \right\|_{\mathrm{L}^2} \leq (\gamma_k - 1) \left\| X_{t_k}^{\mathrm{M}} - X_{t_k}^{\theta^\star} \right\|_{\mathrm{L}^2} ,
$$

with

$$
\gamma_k = 1 + h \frac{4\sqrt{2}}{\sqrt{\alpha_\pi}} \exp\left( \frac{M_\pi}{\alpha_\pi} \right) \| \nabla^2 \log \pi \|_{\mathrm{L}^2(\pi)} \frac{1}{\sqrt{t_k(1 - t_k)}} . \tag{46}
$$

Therefore, we have that

$$
\left\| X_{t_{k+1}}^{\mathrm{M}} - X_{t_{k+1}}^{\theta^\star} \right\|_{\mathrm{L}^2} \leq \gamma_k \left\| X_{t_k}^{\mathrm{M}} - X_{t_k}^{\theta^\star} \right\|_{\mathrm{L}^2} + h\varepsilon + h \left( \sum_{k=0}^{N-1} \int_{t_k}^{t_{k+1}} \mathbb{E}\left[ \left\| \left\{ \tilde{\beta}_t(X_t^{\mathrm{M}}) - \tilde{\beta}_{t_k}(X_{t_k}^{\mathrm{M}}) \right\} \right\|^2 \right] \mathrm{d}t \right)^{1/2} .
$$

To bound the last term of the r.h.s. of the above equation, as before, we first proceed as in the proof of Theorem 1 and obtain

$$
h \left( \sum_{k=0}^{N-1} \int_{t_k}^{t_{k+1}} \mathbb{E}\left[ \left\| \left\{ \tilde{\beta}_t(X_t^{\mathrm{M}}) - \tilde{\beta}_{t_k}(X_{t_k}^{\mathrm{M}}) \right\} \right\|^2 \right] \mathrm{d}t \right)^{1/2} \leq \mathbf{c} h \sqrt{h}(h^{1/16} + 1) \sqrt{\left( d^2 + \| \nabla \log \pi \|_{\mathrm{L}^8(\pi)}^4 \right) d} ,
$$

for some universal constant $\mathbf{c} > 0$.

Therefore, this time we get a recursive formula of the type

$$
\left\| X_{t_{k+1}}^{\mathrm{M}} - X_{t_{k+1}}^{\theta^\star} \right\|_{\mathrm{L}^2} \leq \gamma_k \left\| X_{t_k}^{\mathrm{M}} - X_{t_k}^{\theta^\star} \right\|_{\mathrm{L}^2} + h\sqrt{h}\mathrm{D} + h\varepsilon , \quad k \in \{0, ..., N-2\} ,
$$

with $\gamma_k = (\gamma_k - 1) + 1$ and $(\gamma_k - 1)$ as in (46) and

$$
\mathrm{D} = \mathbf{c}(h^{1/16} + 1) \sqrt{\left( d^2 + \| \nabla \log \pi \|_{\mathrm{L}^8(\pi)}^4 \right) d} . \tag{47}
$$

Developing the recursion as before and using (42), we obtain that

$$
\mathscr{W}_2(\nu^\star, \nu^{\theta^\star}) \leq \left\| X_T^{\mathrm{M}} - X_T^{\theta^\star} \right\|_{\mathrm{L}^2} \leq \left\| X_0^{\mathrm{M}} - X_0^{\theta^\star} \right\|_{\mathrm{L}^2} \prod_{l=0}^{N-2} \gamma_l + (h\sqrt{h}\mathrm{D} + h\varepsilon) \sum_{k=0}^{N-1} \prod_{l=k}^{N-2} \gamma_l \tag{48}
$$
$$
= (\sqrt{h}\mathrm{D} + \varepsilon) \left( h \sum_{k=0}^{N-1} \prod_{l=k}^{N-1} \gamma_l \right) .
$$

Proceeding as before, we get

$$
h \sum_{k=0}^{N-1} \prod_{l=k}^{N-1} \gamma_l \leq \exp\left( \frac{8\sqrt{2}}{\sqrt{\alpha_\pi}} \exp\left( \frac{M_\pi}{\alpha_\pi} \right) \| \nabla^2 \log \pi \|_{\mathrm{L}^2(\pi)} \right)
$$

Plugging this bound in (48), and recalling the definition (47) of D respectively, we obtain (8). $\qquad\square$

