# OpenReview forum: "Diffusion Flow Matching: Dimension-Improved KL Bounds and Wasserstein Guarantees"
_ICML.cc/2026/Conference — ICML 2026 spotlight_

### Official Review · Reviewer_FVSb · 2026-03-05

**Soundness:** 4
**Presentation:** 3
**Significance:** 4
**Originality:** 4
**Overall Recommendation:** 5
**Confidence:** 3

**Summary:**

The article explores the convergence of  Diffusion Flow Matching for generative modelling. Under mild integrability, drift approximation and finite-moment assumptions the authors achieve improved bounds in KL divergence: 1. Without early stopping and constant stepsize 2. With early stopping and constant stepsize 3. By proposing a novel stepsize schedule and using early stopping.

By assuming weak-log-concavity and one-sided Lipschitz continuity the authors are able to also provide 2-Wasserstein convergence guarantees.

**Compliance With Llm Reviewing Policy:**

Affirmed.

**Final Justification:**

This is an interesting article, with good presentation, technically sound and original. The authors addressed all my concerns, therefore I give a final score of 5.

**Key Questions For Authors:**

1) When using early stopping the authors are able to relax the integrability assumption H3 to H4. Could the authors give some insight on the difference in proof roadmap?

2) As I am not very accustomed to the practical side of this work, how realistic is assumption H6? Are there any different ways to measure the approximation?

3) Could the authors explain the proof roadmap for the Wasserstein distance and its main differences from related works?

**Limitations:**

Yes

**Strengths And Weaknesses:**

Soundness: Even though I am not very familiar with the related literature, to my understanding the arguments are well-supported.

Presentation: The presentation is very good. The authors present a very nice background for DFLM which is also useful for a general audience. They proceed to state the assumptions, commenting on their standing with respect to related literature.
They present their results in a nice way. I believe it would be useful if the also presented a table for an easier comparison with related literature.
The presentation for proof roadmap, could be a little detailed but it is to be understood due to the page limit.

Significance : Although not very familiar with related literature, the results are novel and enhance the understaning of the field.
The authors provide state of the art results with respect to the dimension, in both KL and Wasserstein under standard or even weaker assumptions than the stated related work. They also produce results with and without early stopping which strengthens the contributions.

Originality: In the proof roadmap, I believe that many of the approaches used are original in the sense that although they are inspired by other works, they have substantial differences which lead to improved bounds.

---

> ### Author Rebuttal · Authors · 2026-03-30
>
> We thank the reviewer for the careful reading and for recognizing the clarity of our presentation, the rigor of our results, and the technical novelty of our bounds under both KL and Wasserstein distances. We address the questions below.
>
> 1. We thank the reviewer for this question. The key idea is that we can essentially apply Theorem 1 on a truncated time interval $[0,1-\delta]$ to obtain the early-stopping bound in Theorem 2. Concretely, we consider the interpolated process restricted to $[0,1-\delta]$, i.e., $X^I_{[0,1-\delta]}$. Denoting by $\pi_{1-\delta}$ the coupling between $\mu$ and $\nu^\star_{1-\delta}$ corresponding to $(X^I_0,X^I_{1-\delta})$, $\pi_{1-\delta}(x_0, x_{1-\delta})=\int_{R^d} p^I_{1-\delta|0,1}(x_{1-\delta}|x_0, x_1) \pi_{0|1}(x_0| x_1)\nu^\star(d x_1)$, where $(x_0,x_1,x_{1-\delta}) \mapsto p^I_{1-\delta|0,1}(x_{1-\delta}|x_0, x_1)$ denotes the density of $X^I_{1-\delta}$ given $(X^I_0, X^I_1)$ with respect to the Lebesgue measure. By the property of the Brownian bridge, $X^I_{[0,1-\delta]}$ is itself a stochastic interpolant generated from $\pi_{1-\delta}$ and a Brownian bridge on $[0,1-\delta]$. This allows us to apply Theorem 1 “as is” on $[0,1-\delta]$. For this application, it is sufficient that $\pi_{1-\delta}$ satisfies H3. Thanks to the construction above, verifying H3 for $\pi_{1-\delta}$ reduces to the milder assumption H4, which is why early stopping enables this relaxation.
>
> 2. We thank the reviewer for this insightful question. We agree that how to measure the drift approximation error is an important and partly open issue. While alternative notions exist, our choice is closely tied to the statistical analysis of DFMs. Moreover, to the best of our knowledge, this type of control is difficult to avoid when deriving Wasserstein convergence guarantees for diffusion-based models under minimal assumptions. In particular, measuring the approximation error along the law of the generative process is standard in this line of work (see e.g. [15],[16],[17],[18],[19]) . If one is willing to impose stronger regularity assumptions on the estimator, alternative formulations are possible (e.g., measuring the error with respect to the ideal process rather than along the learned dynamics).
>
> [15] Bruno Stefano et al. On diffusion-based generative models and their error bounds: The log-concave case with full convergence estimates.
>
> [16] Strasman Stanislas et al. An analysis of the noise schedule for score-based generative models.
>
> [17] Strasman Stanislas et al. Wasserstein convergence of critically damped langevin diffusions.
>
> [18] Gao Xuefeng et al. Wasserstein convergence guarantees for a general class of score-based generative models.
>
> [19] Gentiloni Silveri Marta, and Antonio Ocello. Beyond log-concavity and score regularity: Improved convergence bounds for score-based generative models in W2-distance.
>
> 3. We thank the reviewer for this question. While a sketch is provided in Section 5, we summarize here the main ideas with additional details. Our starting point is the synchronous coupling between the Markovian projection $X^M$ of the stochastic interpolant and the continuous-time interpolation $X^{\theta^\star}$ of the generative model. The proof then proceeds via a recursive argument in time: for each discretization step $k$, we estimate the $L^2$ norm of the difference at time $t_{k+1}$ between $X^M$ and $X^{\theta^\star}$ in terms of that at time $t_k$, and propagate the bound along the time grid. As is standard, such a recursion requires sufficient regularity of the mimicking drift, and establishing this regularity under minimal data assumptions is the main technical challenge. The key step is to obtain a bound on the Lipschitz constant of the mimicking drift. To this end, we exploit the fact that its Jacobian admits an explicit representation in terms of conditional expectations involving logarithmic derivatives of conditional distributions. This representation allows us to leverage the structure of the stochastic interpolant: since it shares the same Brownian bridge, the regularity assumptions imposed on the data distribution can be transported to the associated conditional kernels and their marginals (see Lemma 9). Building on this, we control the resulting conditional expectations via log-Sobolev and Poincaré inequalities, which yields a quantitative bound on the spatial Lipschitz constant of the mimicking drift (see Proposition 4). Once this regularity estimate is established, the recursive scheme closes and leads to the desired Wasserstein-2 bound.Regarding comparison, as noted in Section 4, no prior work provides Wasserstein bounds for DFMs (without Gaussian prior assumptions) that also account for time discretization error. Therefore, a direct comparison is not available, and our result fills this gap both conceptually and technically.

---

> > ### Author Rebuttal · Reviewer_FVSb · 2026-04-01
> >
> > The authors have answered my questions and addressed my concerns.

---

> > > ### Author Response · Authors · 2026-04-08
> > >
> > > We sincerely thank the reviewer for their time and for their valuable feedback.

---

### Official Review · Reviewer_bK8W · 2026-03-10

**Soundness:** 3
**Presentation:** 4
**Significance:** 3
**Originality:** 3
**Overall Recommendation:** 5
**Confidence:** 3

**Summary:**

The paper deals with Diffusion Flow Matching (DFM), which is a framework in generative modeling. DFM aims to learn a stochastic transport from the base (prior) distribution $\mu$ and the target (data) distribution $\nu^{\star}$, thereby making it possible to generate samples approximately distributed according to $\nu^{\star}$. An interpolated process is, in general, not Markovian. Consequently, DFM defines its Markovian projection through a stochastic differential equation (SDE). The drift is not directly accessible and has to be estimated using neural networks. The SDE is then discretized and solved numerically. The goal of the paper is to quantify how close the law of the generated samples is to the target distribution, i.e., to derive an error bound.

The bounds w.r.t. KL divergence can be categorized by the presence or absence of early stopping and by the step-size policy (constant or following a special schedule). The bound without early stopping and with constant step-size $h$ scales as $\mathcal{O}(hd^3)$, where $d$ is the dimension. Thereby, the authors improve upon the result of Silveri et al. (2024) by relaxing the integrability conditions on the scores of the marginals and by reducing the dimension dependence from $d^4$ to $d^3$. When early stopping is employed, i.e., the simulation is run up to time $1-\delta$, the bound is $\mathcal{O}(hd^3\delta^{-4})$. The bound holds, again, under more general conditions than a corresponding bound by Silveri et al. which is worse by an order of $d$. If, additionally, a special step-size schedule is used, the authors derive a bound $\mathcal{O}(hd^3 \log \delta^{-1})$.

The bound w.r.t. 2-Wasserstein distance is derived for the constant step-size and no early stopping; it scales with $h$ as $\sqrt{h}$ and with $d$ as $\sqrt{d^3}$. Although the existing 2-Wasserstein bound by Xiangjun and Zhongjia (2025) scales better, it was derived under much more restrictive assumptions.

**Compliance With Llm Reviewing Policy:**

Affirmed.

**Final Justification:**

The rebuttal addressed my two main concerns, namely, the authors agreed to add a discussion on practical implications of their theoretical results, and they clarified the mistake in the complexity discussion following Corollary 1. The other results remain correct. Because of these improvements, I raised my score from 4 to 5 (accept). The paper is theoretically sound, well written, and provides meaningful improvements over prior work.

**Key Questions For Authors:**

Could you be more specific about what you mean by computational complexity and how you estimate it? Is it of the same order as the number of discretization points $N_h$? If yes, please clarify the following discrepancy: in Corollary 1, you claim that a step-size of order $\varepsilon^{10}/d^7$ results in complexity $\mathcal{O}(\varepsilon^{-2})$, whereas the implied number of discretization points is $N_h \propto d^7/\varepsilon^{10}$.

**Limitations:**

Mostly yes

**Strengths And Weaknesses:**

Strengths:
- The paper looks theoretically sound. Assumptions are stated clearly, results are formulated rigorously, proofs are available (although I haven’t inspected them thoroughly).
- The work is well-written. In particular, Section 2 presents a clear overview of the subject, and Section 4 provides a detailed comparison with SOTA.
- The work deepens understanding of DFM by providing refined bounds under more general conditions.

Weaknesses:
- The work doesn’t discuss potential practical implications of the theoretical results.
- Computational complexity should be clarified, see questions.

As a sidenote, the work doesn’t provide any numerical illustration, but this is acceptable for a purely theoretical paper in my opinion.

---

> ### Author Rebuttal · Authors · 2026-03-30
>
> We thank the reviewer for the careful reading and the positive assessment of the clarity and rigor of our work, in particular the well-written overview  and comparison with prior art . We address the points raised below.
>
> 1. We agree that the practical implications could have been made more explicit. While our contributions are theoretical, they have direct practical consequences. The improvement in dimension dependence from $d^4$ to $d^3$ yields tighter error bounds in high-dimensional regimes, which are highly relevant in modern ML applications. Our assumptions cover realistic, non-convex distributions, such as double-well potentials and Gaussian mixtures, ensuring applicability beyond idealized settings. The Wasserstein-2 bounds account for all sources of error, including time discretization, providing a robust and complete characterization of the generative process. Overall, these results translate into improved robustness guarantees and a more reliable understanding of DFM behavior in realistic, high-dimensional scenarios. We will add a concise discussion of these practical implications in the revised version.
>
>
> 2. We thank the reviewer for pointing out the discrepancy and agree that there is a typo in the discussion following Corollary 1. For a DFM run up to $T=1$, the computational complexity is, as the reviewer correctly noted, simply proportional to the number of discretization points, i.e., to the reciprocal of the step size. We will correct this in the revised version. We emphasize that all the other complexity results reported in this section (namely Theorems 1, 2, and 3, as well as Corollary 2) are correct and unaffected. We apologize for the oversight and will clarify this point to avoid any confusion.

---

> > ### Author Rebuttal · Reviewer_bK8W · 2026-04-01
> >
> > My questions have been addressed and I will raise the score by one point.

---

> > > ### Author Response · Authors · 2026-04-08
> > >
> > > We appreciate the reviewer’s careful consideration of our response and their updated assessment.

---

### Official Review · Reviewer_tNUE · 2026-03-12

**Soundness:** 3
**Presentation:** 3
**Significance:** 3
**Originality:** 3
**Overall Recommendation:** 5
**Confidence:** 3

**Summary:**

This paper analyzes discretization error in Brownian-bridge diffusion flow matching (DFM). It derives non-asymptotic convergence guarantees for the learned Markovian projection and its Euler-Maruyama discretization under both KL divergence and Wasserstein-2 distance. The main technical result is an improved KL bound that reduces dimension dependence from $d^4$ to $d^3$ without early stopping, under finite-moment and score-integrability assumptions. The paper also gives early-stopped KL guarantees under weaker conditional-score assumptions, introduces a nonuniform step-size schedule with better complexity, and proves $W_2$ convergence under weak log-concavity and one-sided log-Lipschitz conditions.

**Compliance With Llm Reviewing Policy:**

Affirmed.

**Final Justification:**

The author clarifies the intended scope of the paper and gives a more persuasive explanation of why assumptions such as H3/H4/H5 are milder than classical global regularity conditions. In particular, the authors explain that the work is meant as a population-level analytical convergence study, and they provide additional intuition/examples for the regularity assumptions used in the KL and Wasserstein analyses. This addresses part of my concern about how to interpret the assumptions. However, the concern in W1 remains: the theory is still entirely population-level, and the main approximation assumptions are not connected to finite-sample estimation, neural network capacity, or optimization error. Overall, the rebuttal does not fully resolve this issue, but it improves the paper’s positioning and makes the assumptions better motivated.

**Key Questions For Authors:**

See weaknesses.

**Limitations:**

yes

**Strengths And Weaknesses:**

Strengths:
1. The paper targets a meaningful gap in the DFM literature: rigorous non-asymptotic guarantees that simultaneously reflect drift approximation and time discretization, especially beyond deterministic FM.
2. The paper provides a genuine technical improvement over the closest KL analysis.

Weaknesses:
1. The paper assumes population-level drift approximation errors H1/H6, but does not connect these assumptions to finite-sample estimation, neural network capacity, etc. This might limit the practical interpretability of the bounds.
2. The assumptions are milder than some earlier work, but they are still strong and somewhat abstract from a practical ML perspective.

---

> ### Author Rebuttal · Authors · 2026-03-30
>
> We thank the reviewer for the careful reading and for recognizing the technical novelty of our work, in particular the non-asymptotic guarantees that account for both drift approximation and discretization, as well as the improvement over prior KL analysis. We address the concerns below.
>
> 1. We agree that connecting population-level assumptions to finite-sample estimation, neural network capacity, and other practical considerations is an important direction. Our current work focuses on analytical convergence bounds under population-level assumptions. As noted in Section 6, it would indeed be highly valuable to complement these results with statistical convergence rates. We emphasize that statistical estimation (finite-sample, model capacity, approximation error) and analytical convergence (KL or Wasserstein guarantees under population-level assumptions) are complementary but distinct problems, each requiring separate techniques. Indeed, to the best of our knowledge, except for [9], existing works on FMs do not address statistical convergence, and no prior work on DFMs provides such guarantees. Extending our results to a full statistical framework is important but nontrivial, and we will clarify this positioning in the revised version to better delineate the scope of our analysis.
>
> [9] Yuan Gao et al. Convergence of continuous normalizing flows for learning probability distributions.
>
> 2. We would like to clarify that our assumptions are significantly milder than they may appear, and are standard in modern ML theory. For the KL bounds, assumption H3 requires only integrability of the score function, which is weak and does not impose any global Lipschitz condition. Moreover, in the early stopping regime, the requirement can be further relaxed: for instance, in the case of independent coupling, the only assumption on the data distribution reduces to a finite moment condition, which is among the mildest assumptions commonly used in ML (see Theorem 2). For the Wasserstein-2 bounds, assumption H5 requires weak log-concavity and one-sided log-Lipschitz continuity, which are strictly weaker than the usual global Lipschitz assumptions on the score and allow for non-convex and multimodal distributions. For example, perturbations of strongly log-concave distributions (including double-well potentials) satisfy weak log-concavity, and Gaussian mixtures satisfy H5. Assumptions of this type are now standard in ML theory, in particular in the analysis of diffusion-based models (e.g., [10], [11], [12], [13], [14]). That said, we acknowledge that our assumptions still do not cover all real-world data distributions, and there remains room for further improvements in this direction.
>
>
> [10] Gitte Kremling et al. Non asymptotic error bounds for probability flow odes under weak log-concavity.
>
> [11] Gentiloni Silveri Marta et al. Exponential Convergence Guarantees for Iterative Markovian Fitting.
>
> [12] Gentiloni Silveri Marta, and Antonio Ocello. Beyond log-concavity and score regularity: Improved convergence bounds for score-based generative models in W2-distance.
>
> [13] Vahan Arsenyan et al. Assessing the Quality of Denoising Diffusion Models in Wasserstein Distance: Noisy Score and Optimal Bounds
>
> [14] Bruno Stefano, and Sotirios Sabanis. Wasserstein convergence of score-based generative models under semiconvexity and discontinuous gradients.

---

> > ### Author Rebuttal · Reviewer_tNUE · 2026-04-01
> >
> > The author clarifies the intended scope of the paper and gives a more persuasive explanation of why assumptions such as H3/H4/H5 are milder than classical global regularity conditions. In particular, the authors explain that the work is meant as a population-level analytical convergence study, and they provide additional intuition/examples for the regularity assumptions used in the KL and Wasserstein analyses. This addresses part of my concern about how to interpret the assumptions. However, the concern in W1 remains: the theory is still entirely population-level, and the main approximation assumptions are not connected to finite-sample estimation, neural network capacity, or optimization error.
> > Overall, the rebuttal does not fully resolve this issue, but it improves the paper’s positioning and makes the assumptions better motivated. I have increased the score to 5.

---

> > > ### Author Response · Authors · 2026-04-08
> > >
> > > We thank the reviewer for considering our response and for increasing their score accordingly.

---

### Official Review · Reviewer_HYB4 · 2026-03-13

**Soundness:** 3
**Presentation:** 3
**Significance:** 3
**Originality:** 3
**Overall Recommendation:** 4
**Confidence:** 2

**Summary:**

This paper provides convergence analysis for diffusion flow matching in both KL divergence and Wasserstein distance. The results improve and extend existing convergence results.

**Compliance With Llm Reviewing Policy:**

Affirmed.

**Final Justification:**

The authors have adequately addressed my concerns in the rebuttal, and I'll maintain my score.

**Key Questions For Authors:**

1. What's the key technique and intuition that enables the improvement in the dependence on dimension?

2. Why is DFM of interest? What is their advantage over commonly used FM and DM? While the paper says DFM has empirical success, no citation is provided, and the reviewer is not aware of such work. It would be good to cite empirical works and discuss the practical relevance of the present work.

3. The results for FM is better than the results here. What makes the analysis of DFM particularly difficult?

**Limitations:**

Yes

**Strengths And Weaknesses:**

The results are presented in a mathematically rigorous manner and seem technically sound, although I didn't check the proof details. The results are novel and improve existing results. The authors only mention that the improvement relies on refined bounds, but provides no summary of the key technique and intuition.

---

> ### Author Rebuttal · Authors · 2026-03-30
>
> We thank the reviewer for the positive assessment of the paper, in particular for highlighting the novelty, technical soundness, and improvement over prior work. We are happy to clarify the key points raised below.
>
> 1. We thank the reviewer for this important question. We clarify both the intuition and the key technical mechanism. Our starting point was the observation that the existing $d^4$ scaling is likely not intrinsic. On one hand, [1] derives KL bounds with $\mathcal{O}(d^3)$ dependence, albeit at the cost of two-sided early stopping. On the other hand, in the Gaussian prior setting, [2] obtain $\mathcal{O}(\sqrt{d})$ scaling in Wasserstein-2 distance. See Section 4 for a detailed comparison. These results suggest that the previously observed dimension dependence is mainly a byproduct of proof techniques. Our approach builds on the control-theoretic viewpoint of [3] and [4]. We reformulate the KL bound as a control problem, reducing it to bounding the $\mathrm{L}^2$ norm of the adjoint process in the Pontryagin system associated with the Markovian projection of the interpolant. This leads to estimating the time-integrated $\mathrm{L}^2$ norm of the reciprocal characteristic of the mimicking drift, which admits an explicit expansion involving up to three logarithmic derivatives of conditional distributions. The main bottleneck is the control of terms involving three logarithmic derivatives. In [4], these are handled via direct expansion, leading to $d^4$ scaling. Our key idea is to avoid this expansion and instead apply an integration-by-parts argument (see p.23), which transfers one derivative onto the coupling. This effectively reduces the order of differentiation from three to two, lowering the combinatorial complexity and improving the scaling from $d^4$ to $d^3$. Due to space constraints, Section 5 only sketches these arguments. We will expand this part in the revision to make the mechanism more transparent.
>
> [1] Yuhao Liu et al. Finite-time analysis of discrete-time stochastic interpolants.
>
> [2] Xiangjun Meng, and Zhongjian Wang. Pathway to $o(\sqrt{d})$ complexity bound under Wasserstein metric of flow-based models.
>
> [3] Conforti Giovanni et al. KL convergence guarantees for score diffusion models under minimal data assumptions.
>
> [4] Gentiloni Silveri Marta et al. Theoretical guarantees in kl for diffusion flow matching.
>
> 2. We thank the reviewer for this comment and agree that our phrasing may have overemphasized the “empirical success” of DFMs. In fact, our primary motivation is theoretical: understanding DFMs rigorously is highly nontrivial due to the substantial technical challenges involved. See our response to the next question for more details on these challenges. We will soften the language regarding empirical success in the revised version. That said, there is indeed empirical evidence supporting DFMs: in [5] (Section 7), stochastic interpolants are shown to outperform deterministic ones and increase sample diversity, highlighting advantages over standard FMs. Furthermore, DFMs can be seen as the first iterative step of Diffusion Schrödinger Bridge Matching (DSBM) [6], which have recently gained popularity in generative modeling. These points reinforce the practical and conceptual relevance of studying DFMs alongside the theoretical analysis we provide.
>
> [5] Michael S. Albergo et al. Stochastic Interpolants: A Unifying Framework for Flows and Diffusions.
>
> [6] Yuyang Shi et al. Diffusion Schrödinger Bridge Matching.
>
> 3. The main reason FM results appear better is that they are obtained under stronger assumptions, in particular Lipschitz assumptions on the approximated velocity field, and typically do not account for discretization error ([7], [8]). Our goal, and what we achieve in this work, is to derive convergence bounds under minimal assumptions that capture all sources of error, including both drift approximation and time discretization. In addition, DFMs introduce intrinsic challenges absent in standard FMs. The core difficulty is that stochastic interpolants are non-Markovian, so they cannot be analyzed directly; one must instead consider their Markovian projection. This adds a significant layer of complexity: the associated mimicking drift is defined via a nontrivial conditional expectation, whose regularity properties are not immediate. These challenges drive the technical core of our analysis: for the KL bound, we control the $\mathrm{L}^2$ norm of the reciprocal characteristic of this drift; for the Wasserstein bound, we carefully propagate regularity assumptions from the data distribution through this implicit object. These difficulties are intrinsic to DFMs and explain the gap with FM results.
>
> [7] Albergo Michael S., and Eric Vanden-Eijnden. Building normalizing flows with stochastic interpolants.
>
> [8] Benton Joe et al. Error bounds for flow matching methods.

---

> > ### Author Rebuttal · Reviewer_HYB4 · 2026-04-02
> >
> > The authors have adequately addressed most of my concerns in the rebuttal. I will maintain my positive score.

---

> > > ### Author Response · Authors · 2026-04-08
> > >
> > > We thank the reviewer for engaging with our response and for their constructive comments.

---

### Decision · Program_Chairs · 2026-04-30

**Decision:**

Accept (spotlight)

**Comment:**

The reviewers agree that this paper makes a technically strong contribution to the theory of Diffusion Flow Matching. In particular, they highlight the paper’s rigorous non-asymptotic analysis of discretization error under both KL divergence and Wasserstein-2 distance, its improved dimensional dependence relative to prior work, and its ability to obtain these guarantees under weaker or more general assumptions than existing results. Several reviewers also noted that the paper is clearly written, provides a useful overview of the area, and offers a careful comparison to prior literature. Importantly, the rebuttal was effective in clarifying the key technical ideas behind the improved bounds, the intended population-level scope of the paper, and the role and motivation of the assumptions, while also addressing presentation issues such as the discussion of practical implications and the complexity typo. Although some concerns remain about the lack of connection to finite-sample estimation, model capacity, and optimization error, the consensus is that these limitations concern scope rather than correctness and do not diminish the significance of the theoretical advances made here. Overall, the paper is viewed as technically sound, original, and likely to be of interest to researchers working on the theory of generative modeling, and I therefore recommend acceptance.